# NKX2-1 drives neuroendocrine transdifferentiation of prostate cancer via epigenetic and 3D chromatin remodeling

Xiaodong Lu [1,7], Viriya Keo[1,2,7], Irina Cheng [2], Wanqing Xie[1],
Galina Gritsina [1,2], Juan Wang[3], Lina Lu[3], Cheng-Kai Shiau [3], Yueying He[3],
Qiushi Jin[3], Peng Jin [4], Martin G. Sanda[1,5], Victor G. Corces[4],
Nicolas Altemose[6], Ruli Gao [3], Jonathan C. Zhao [4,5] ✉ & Jindan Yu [1,4,5] ✉

A substantial amount of castration-resistant prostate cancer (CRPC) progresses into a neuroendocrine (NE) subtype, known as NEPC, which is associated with poor clinical outcomes. Here we report distinct three-dimensional chromatin architectures between NEPC and CRPC tumors, which were recapitulated by isogenic cell lines undergoing NE transformation (NET). Mechanistically, pioneer factors such as FOXA2 initiate binding at NE enhancers to mediate regional DNA demethylation and induce neural transcription factor (TF) *NKX2-1* expression. NKX2-1 preferentially binds gene promoters and interacts with enhancer-bound FOXA2 through chromatin looping. NKX2-1 is highly expressed in NEPC and indispensable for NET of prostate cancer. NKX2-1/FOXA2 further recruits p300/CBP to activate NE enhancers, and pharmacological inhibition of p300/CBP effectively blunts NE gene expression and abolishes NEPC tumor growth. Taken together, our study reports a hierarchical network of TFs governed by NKX2-1 in critically regulating chromatin remodeling and driving luminal-to-NE transformation and suggests promising therapeutic approaches to mitigate NEPC.

Following extensive treatment with androgen receptor (AR) pathway inhibitors, advanced prostate cancer (PCa) frequently develops treatment resistance. Approximately 20% of these castration-resistant prostate cancers (CRPC) transdifferentiate to display, at least partially, neuroendocrine (NE) histological and molecular features, thus called neuroendocrine PCa (NEPC)[1–4]. While genetic alterations of tumor-suppressor genes *TP53* and *RB1* are more frequent in NEPC[5–9], it largely shares genomic changes with CRPC tumors[3]. In contrast, substantial epigenetic differences in DNA methylation[3], chromatin accessibility[10]

and histone modifications[11] have been noted in NEPC and might have causative roles in promoting NE transformation (NET) of PCa. A plethora of transcription factors (TFs) and chromatin modifiers, including SOX2, MYCN, EZH2 and FOXA1, are deregulated in NEPC to suppress AR signaling and/or mediate epigenetic remodeling, yielding lineage plasticity[11–18]. Of high relevance, FOXA2, a nuclear biomarker of NEPC, is a pioneer factor that promotes NET of PCa, in part by driving KIT pathway activations[19,20].

Recent advances in the study of three-dimensional (3D) chromatin folding have suggested that the human genome is hierarchically

[1]Department of Urology, Emory University School of Medicine, Atlanta, GA, USA. [2]Division of Hematology and Oncology, Department of Medicine, Northwestern University Feinberg School of Medicine, Chicago, IL, USA. [3]Department of Biochemistry and Molecular Genetics, Northwestern University Feinberg School of Medicine, Chicago, IL, USA. [4]Department of Human Genetics, Emory University School of Medicine, Atlanta, GA, USA. [5]Winship Cancer Institute, Emory University School of Medicine, Atlanta, GA, USA. [6]Department of Genetics, Stanford University, Stanford, CA, USA. [7]These authors contributed equally: Xiaodong Lu, Viriya Keo. ✉e-mail: changsheng.zhao@emory.edu; jindan.yu@emory.edu

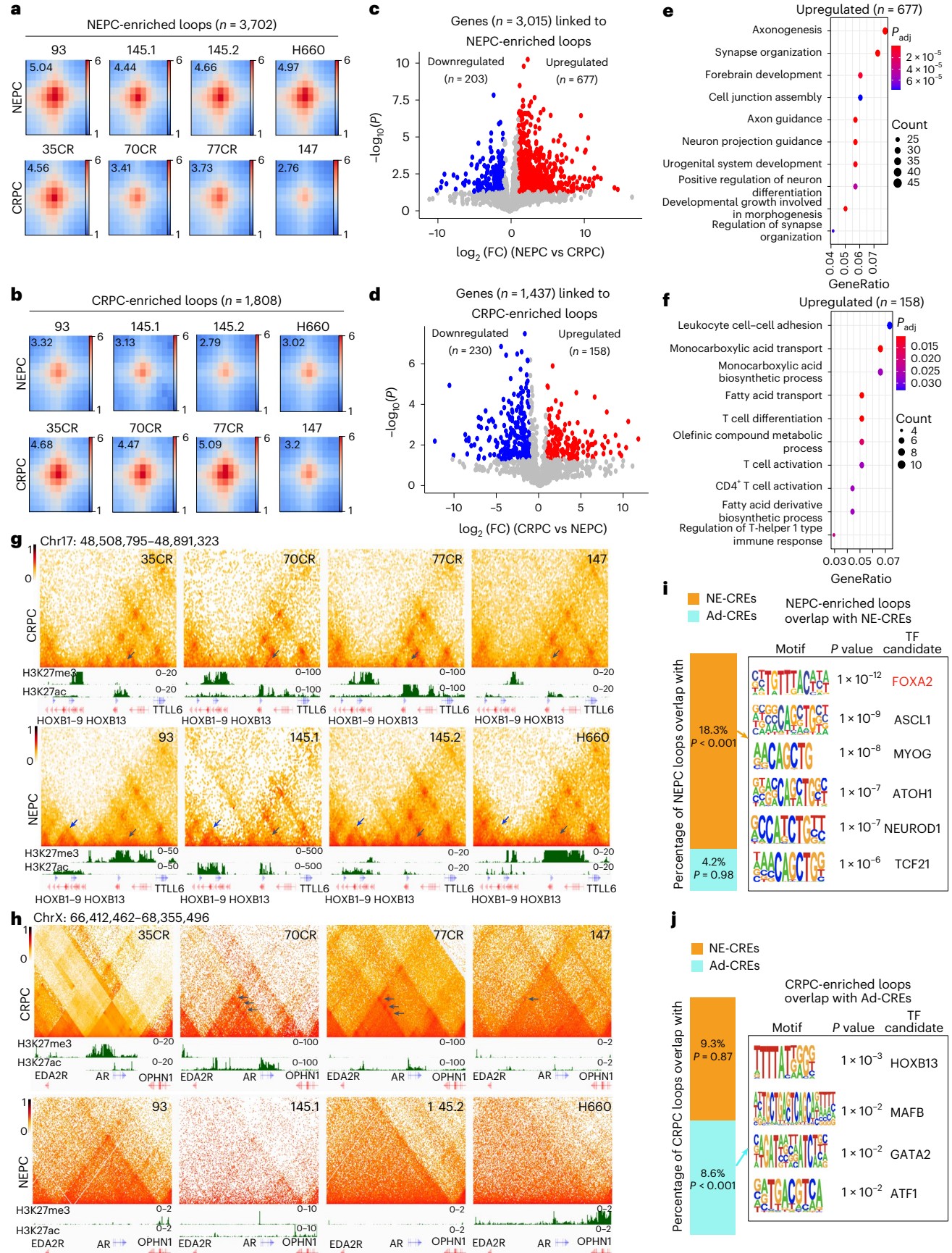

**Fig. 1 | Distinct 3D chromatin architecture in NEPC compared to CRPC tumors.**
**a,b**, APA plots of NEPC-enriched (**a**) and CRPC-enriched (**b**) loops in NEPC (LuCaP93, 145.1, 145.2, and NCI-H660) and CRPC samples (LuCaP35CR, 70CR, 77CR, 147). The APA score is annotated within each plot. Scale bars indicate $\log_2$ of observed over expected enrichment. $P = 0.026$ and $P = 0.022$, respectively, for NEPC- and CRPC-enriched loop APA scores between NEPC and CRPC samples. $P$ values by one-sided Student's $t$ test. **c**, Volcano plot showing genes linked to NEPC-enriched loops that are differentially expressed in NEPC versus CRPC using $P < 0.05$ (two-sided $t$ test) and $\log_2(FC) \geq 1$. **d**, Volcano plot of genes linked to CRPC-enriched loops and are differentially expressed in CRPC versus NEPC using $P < 0.05$ (two-sided $t$ test) and $\log_2(FC) \geq 1$. **e,f**, GO analyses of genes linked to NEPC-enriched (**e**) and CRPC-enriched (**f**) loops and are upregulated in NEPC and CRPC, respectively, as defined in **c** and **d**. **g,h**, Hi-C contact maps (5 kb resolution) of the *HOXB* gene cluster (**g**) and *AR* gene region (**h**) in CRPC (top row) and NEPC (bottom row) models. Gene boxes and 1D ChIP–seq tracks of matched models are aligned at the bottom. Blue arrows in **g** indicate NEPC-enriched loops at the *HOXB1–HOXB9* loci, and black arrows indicate CRPC-enriched loops at the *HOXB13* locus. The scale bars indicate ICE-normalized contact frequency. **i,j**, The bar graphs showing the percentage of NEPC-enriched (**i**) or CRPC-enriched (**j**) loop anchors that overlap with previously reported[11] NE-enriched CREs ($n = 14,985$) and Ad-CREs ($n = 4,338$). The enriched motifs at overlapped anchors are shown on the right. $P$ values by one-sided permutation test (without multiple comparisons) to evaluate the significant differences of the indicated overlap relative to control regions of equal size.

organized into multiple layers, including A/B compartments[21], topologically associating domains (TADs)[22], and enhancer–promoter (E–P) chromatin loops[23]. Such 3D chromatin structure, shaped by lineage-specific TFs, is lineage-specific and critical for cell fate determination[24–27]. A recent Hi-C sequencing of 80 clinical CRPC highlighted major interactions among 3D chromatin structure, epigenetic landscape and gene expression[28]. However, the 3D chromatin architecture in NEPC tumors and its difference from CRPC have not been explored. Furthermore, it remains elusive how lineage-specific pioneer factors cooperate with neuronal TFs to regulate epigenetic remodeling and chromatin reorganization to facilitate the luminal-to-NE cell type switch that underlies NET.

NKX2-1 is a neuronal TF that is detected early in the endodermal thyroid and lung and in restricted neuroblast cells of the brain[29–31]. It regulates the identity of neuronal progenitor cells, mediates interneuron specification and directs postmitotic neuron migration[32,33]. In the lung, it has been shown to recruit FOXA proteins to lung-specific loci to promote cell growth and cellular identity while repressing gastric differentiation[34–36]. NKX2-1 is undetectable in the prostate epithelium but is highly expressed in more than 50% of NE lesions[37]. Ectopic overexpression (OE) of NKX2-1 has been shown to facilitate ASCL1 in redistributing FOXA1 from luminal- to NE-specific regulatory elements, resulting in NET[11]. This could be an adaptive event as FOXA1 is usually downregulated, rather than upregulated, in NEPC[12]. How NKX2-1, as a neuronal TF, actively drives NET and the mechanisms of its induction in NEPC tumors are yet to be characterized.

Here we showed different 3D chromatin architecture in NEPC versus CRPC tumors that were recapitulated in isogenic LNCaP cells undergoing NET following FOXA2 OE. Mechanistically, NKX2-1, predominantly binding the promoters, is induced by FOXA2 and interacts with enhancer-bound FOXA2, which further stabilizes FOXA2 binding at NE enhancers and propels regional epigenetic remodeling. Subsequently, NKX2-1 and FOXA2 induce NE enhancer activation and NE transcriptional program through recruiting p300/CBP. Accordingly, p300/CBP inhibition abolishes NEPC growth in vitro and in vivo. Taken together, our study deciphers a mechanism fundamental to cellular identity switch during NET and identifies potential therapeutic approaches to suppress NEPC.

## Results

### Distinct 3D chromatin architecture in NEPC versus CRPC tumors

To compare the 3D chromatin architecture of PCa cells, we performed Hi-C analyses of a panel of patient-derived xenograft (PDX) tumors with over 600 million valid paired-end reads per sample (Supplementary Table 1). Unsupervised hierarchical clustering of the pairwise correlation coefficients of Hi-C samples based on their first principal component largely separated CRPC and NEPC tumors, except LuCaP147, which has a transcriptome in between[38] (Extended Data Fig. 1a). Next, we identified chromatin loops at 10 kb resolution[39] and defined those preferentially ($P < 0.001$) detected in at least two NEPC samples in pairwise comparisons to CRPC as NEPC-enriched loops ($n = 3,702$) and vice versa for CRPC-enriched loops ($n = 1,808$). Concordantly, NEPC-enriched loops showed significantly ($P = 0.026$) higher aggregate peak analysis (APA) scores in NEPC tumors than CRPC (Fig. 1a), and vice versa for CRPC-enriched loops ($P = 0.022$; Fig. 1b).

To assess the potential targets of these NEPC/CRPC-enriched loops, we identified genes within ±3 kb of the loop anchors and examined their expression in public NEPC versus CRPC RNA-seq data[40,41] (Fig. 1c,d). Gene Ontology (GO) analyses revealed that NEPC-loop-associated genes that were also upregulated in NEPC were enriched for neuron development processes, whereas CRPC-loop genes that were upregulated in CRPC were enriched for epithelial and prostatic functions (Fig. 1e,f). For instance, a unique TAD at the *HOXB1–HOXB9* cluster, encoding genes crucial for embryonic development, was present in NEPC but not CRPC tumors (Fig. 1g). Accordingly, this region was strongly marked by H3K27me3 in CRPC, which was diminished in NEPC with a striking gain of active histone marker H3K27ac, accompanied by their concordant upregulation in NEPC (Extended Data Fig. 1b). By contrast, the adjacent luminal TF *HOXB13* belonged to a different TAD that showed strong chromatin interactions in CRPC tumors and was diminished in NEPC, with a concordant loss of H3K27ac and gain of H3K27me3 (Fig. 1g), in agreement with its downregulation in NEPC[42]. A similar loss in chromatin interactions between a recently reported super-enhancer (SE)[43] and the *AR* promoter was also observed (Fig. 1h and Extended Data Fig. 1c), whereas chromatin interactions at NE regulators, such as *ASCL1* and *INSM1*, were enhanced in NEPC versus CRPC (Extended Data Fig. 1d–g).

**Fig. 2 | 3D chromatin reorganization in isogenic PCa cells undergoing NET.**
**a**, Heatmap shows six clusters of genes differentially expressed over the time course (adjusted $P < 0.0001$). Representative genes and significantly enriched molecular concepts are shown on the right. Bulk RNA-seq was performed in LNCaP cells with FOXA2 OE at the indicated time points. Color bar −$z$ score. Adjusted $P$ values were calculated using the likelihood ratio test, followed by Benjamini–Hochberg correction. **b**, WB analyses of luminal and NE lineage markers in time-course LNCaP+FOXA2 cells. LNCaP cells were infected with control (plv) or FOXA2 virus and followed up for 28 days. Data shown are from one of three ($n = 3$) independent experiments. **c**, Integrative analyses of ARS and NES scores in time-course LNCaP+FOXA2 samples and those of clinical PCa samples (GSE126078). ARS and NES scores were calculated by gene set variation analysis. **d**, APA plots of luminal (top row) and NE (bottom row) loops (D28 versus D0) in time-course LNCaP+FOXA2 samples and NCI-H660. The APA score is annotated within each plot. Scale bars indicate $\log_2$ of observed over expected enrichment. **e,f**, Hi-C contact maps (5 kb resolution) of the *AR* gene region (**e**) and the *HOXB* gene cluster (**f**) in time-course LNCaP+FOXA2 cells. Gene boxes and 1D ChIP–seq tracks of matched models are aligned at the bottom. Black arrows indicate luminal loops at *AR* and *HOXB13* loci, and blue arrows indicate NE loops at *HOXB1–HOXB9* loci. Enhancers are highlighted in light blue, and promoters in light orange color. FOXA2, H3K27ac, H3K4me1, H3K27me3 and CTCF tracks under the D0 contact map used their corresponding ChIP–seq performed in closely related D2 cells. All others were done in conditions exactly matched to Hi-C. The scale bars indicate ICE-normalized contact frequency.

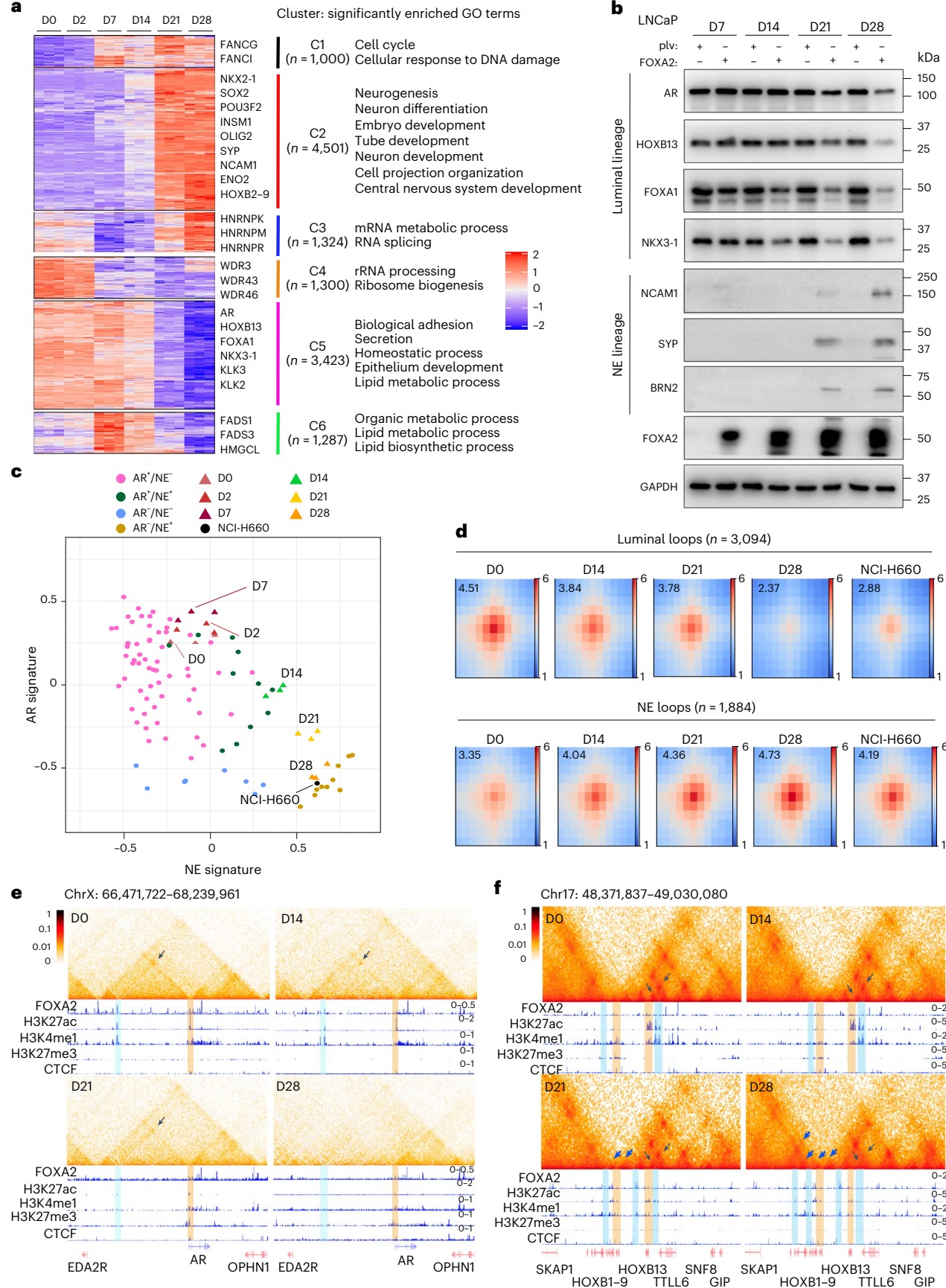

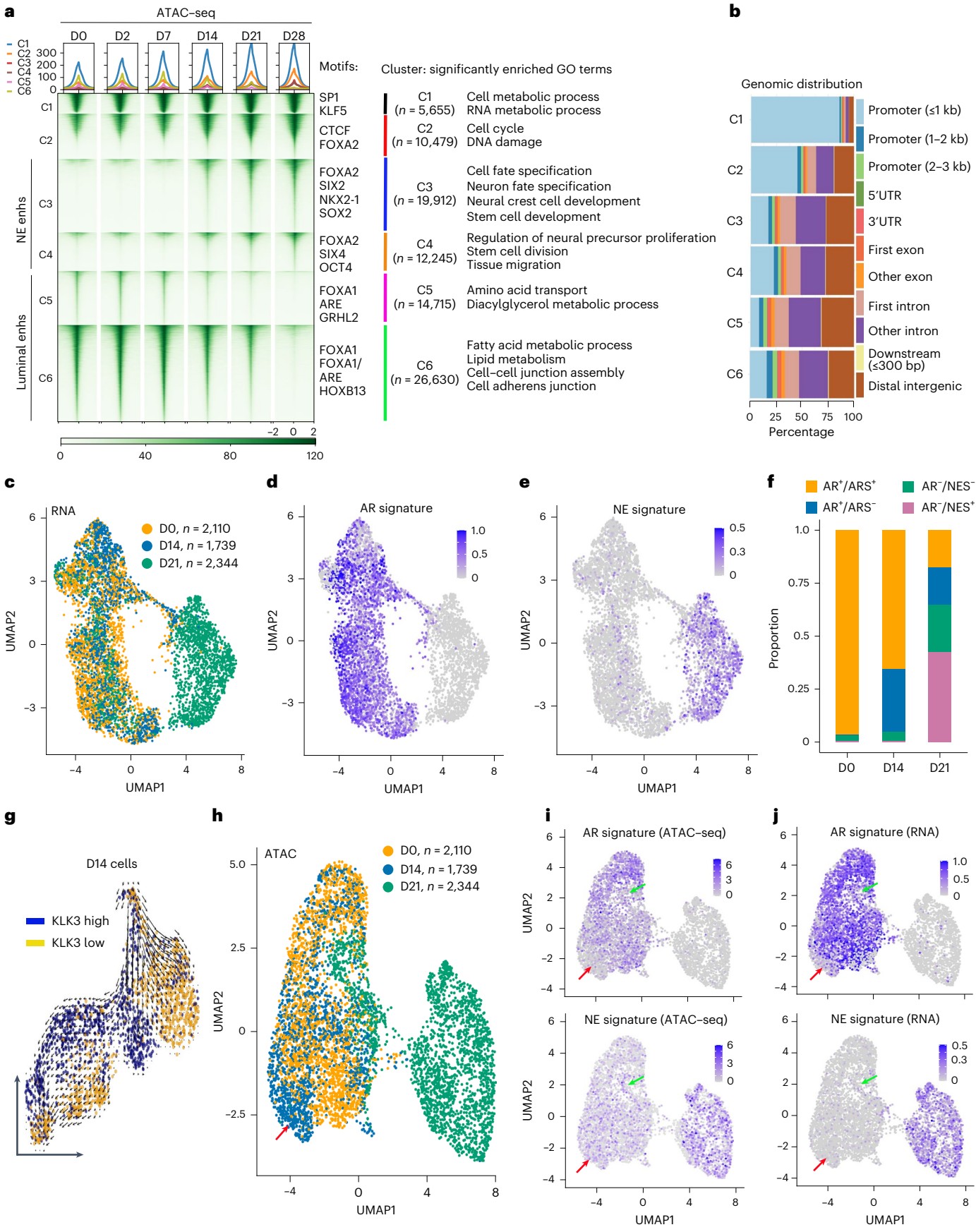

**Fig. 3 | Single-cell multiome identified transitioning individual cells with intermediate transcriptome and chromatin states. a**, *K*-means clustering reveals six clusters of differential ATAC–seq peaks (±2 kb) across time-course LNCaP+FOXA2 samples (adjusted *P* < 0.0001). Heatmaps shown here were scaled across samples. Significantly enriched TF motifs and GO terms for each cluster, along with the number of peaks, are shown on the right. Color bar at the bottom indicates the scale of enrichment intensity. Adjusted *P* values were calculated using the likelihood ratio test followed by Benjamini–Hochberg correction. **b**, Genomic distribution of the six clusters of ATAC–seq peaks in **a**. **c**–**f**, scRNA–seq UMAP visualization of D0, D14 and D21 LNCaP+FOXA2 cells (**c**), AR (**d**) and NE (**e**) signature genes, which are further quantified for the proportion of $AR^+/ARS^+$, $AR^+/ARS^-$, $AR^-/NES^-$ and $AR^-/NES^+$ cells (**f**). **g**, RNA velocity analysis of D14 LNCaP+FOXA2 cells. The data are visualized as streamlines in a UMAP-based embedding. Blue dots indicate *KLK3*-high cells, and the yellow dots indicate *KLK3*-low cells. **h**–**j**, scATAC–seq UMAP visualization of D0, D14 and D21 LNCaP+FOXA2 cells (**h**), chromatin accessibility (**i**) and expression (**j**) of ARS/NES genes. Red and green arrows indicate transitioning D14 and D21 cells, respectively. enhs, enhancers.

As lineage-specific TFs bind at enhancers and might orchestrate CRPC- and NEPC-specific enhancer looping to target promoters, we focused on previously defined NE-enriched candidate regulatory elements (NE-CREs) and adenocarcinoma-enriched CREs (Ad-CREs)[11]. Not surprisingly, significantly more NEPC- and CRPC-enriched loops overlapped with NE-CREs and Ad-CREs, respectively, than with control chromatin regions (*P* < 0.001 by permutation test; Fig. 1i,j). Interestingly, the DNA-binding motif of FOXA2, a biomarker[44,45] and a driver of NEPC[19,20], was significantly enriched in NEPC-enriched loops that anchored at NE-CREs. In contrast, the motifs of luminal TFs were enriched in CRPC-enriched loops overlapping with Ad-CREs (Fig. 1i,j). Altogether, our data demonstrate striking differences in 3D chromatin architecture between NEPC and CRPC tumors and nominate FOXA2 as a major regulator of NE-specific E–P looping.

### FOXA2 drives NET of PCa, showing clinically relevant phenotypes

We infected PCa cell lines with FOXA2 OE lentivirus and observed a gradual transformation of AR-positive LNCaP and 22Rv1 cells to NEPC phenotype over the course of 28 days and of the AR-negative Du145 cells in only 7 days (Extended Data Fig. 2a,b and Supplementary Note 1). On the flip side, FOXA2 was upregulated in the previously reported NEPC model of LNCaP with *TP53* and *RB1* knockdown (KD)[6], and FOXA2 knockout (KO) abolished NE marker gene expression (Extended Data Fig. 2c). Furthermore, LNCaP with FOXA2 OE grew xenograft tumors much more rapidly than the control cells, showing mixed adenocarcinoma to small-cell carcinoma histology and strong synaptophysin (SYP) staining that closely recapitulated treatment-induced NEPC in patients (Extended Data Fig. 2d–f and Supplementary Note 1).

RNA-sequencing (RNA-seq) analyses of time-course LNCaP+FOXA2 cells revealed many differentially expressed genes (*n* = 12,835), indicating major transcriptional reprogramming. *K*-means clustering revealed six clusters, including three clusters each of upregulated (C1–C3, *n* = 6,825) and downregulated genes (C4–C6, *n* = 6,010) that were respectively involved in neuron/embryonic development and luminal cell functions (Fig. 2a and Supplementary Note 1). Western blot (WB) confirmed gradually reduced expression of luminal TFs over time and increased expression of NE markers, such as SYP, NCAM1 and BRN2, manifesting at D21 and peaking at D28 (Fig. 2b). Critically, D0, D2 and D7 LNCaP+FOXA2 cells showed transcriptomes similar to $AR^+/NE^-$ clinical PCa samples[40], while D21 and D28 cells more closely clustered with $AR^-/NE^+$ (NEPC) tumors (Fig. 2c). Notably, D14 cells fell somewhere in between, indicating an intermediate transcriptome. These data suggest that LNCaP+FOXA2 cells faithfully modeled the clinical transition of PCa from adenocarcinoma to NEPC and identified D14 as the tipping point of the transition.

To determine whether such major transcriptional reprogramming may be propelled by 3D chromatin reorganization, we performed Hi-C of time-course LNCaP+FOXA2 cells. Not surprisingly, D0 and D14 cells harbored 3D chromatin architecture similar to CRPC tumors, whereas D28 cells were closely grouped with NEPC tumors (Extended Data Fig. 2g and Supplementary Note 1). Interestingly, despite the NE-like transcriptome, D21 cells remained in the CRPC cluster, suggesting that stable chromatin reorganization lagged transcriptional reprogramming during NET. Similar to findings in clinically relevant PDX tumors (Fig. 1e,f), genes associated with NE (enriched in D28) and luminal (enriched in D0) loops were, respectively, implicated in axonogenesis/developmental processes and epithelial functions (Fig. 2d, Supplementary Note 1 and Supplementary Fig. 1a). A TAD between AR enhancer and promoter was detected at D0, which was gradually eliminated over time, accompanied by a decrease of H3K4me1 and H3K27ac and gain of H3K27me3 (Fig. 2e). Likewise, there were contrasting changes in epigenetic modifications and TADs at the *HOXB1–HOXB9* cluster and *HOXB13* gene over time (Fig. 2f), recapitulating our earlier observations in PDX tumors. Therefore, our LNCaP+FOXA2 system represents an isogenic cell model of NET with high clinical relevance.

### Clonal transformation and expansion during NET of PCa

To understand the epigenetic bases of transcriptional reprogramming and 3D chromatin reorganization during NET, we performed the assay for transposase-accessible chromatin using sequencing (ATAC–seq) in time-course LNCaP+FOXA2 cells. PCA analyses of ATAC–seq peaks showed that D0/D2/D7 cells and D21/D28 cells were, respectively, clustered closer to CRPC and NEPC PDX tumors[10], whereas the D14 cells were positioned somewhere in between, indicating a transitioning stage (Extended Data Fig. 3a). *K*-means clustering revealed two clusters, each of gradually increasing (C3–C4) and gradually decreasing (C5–C6) gene-distal ATAC–seq peaks, hereafter defined as NE and luminal enhancers, respectively (Fig. 3a,b). Accordingly, NE and luminal enhancers, respectively, are enriched for motifs of neural/stem cell TFs and luminal TFs and associated with genes involved in developmental processes and luminal cell functions (Extended Data Fig. 3b,c and

**Fig. 4 | NKX2-1 is induced by FOXA2 and required for FOXA2-driven NET. a**, Genome browser view of *NKX2-1* mRNA (top row), and FOXA2, H3K4me1, H3K27ac ChIP–seq signal around the *NKX2-1* gene in time-course LNCaP+FOXA2 cells. The promoter is highlighted in light blue, and enhancers are highlighted in yellow. **b**, WB of NEPC cell lines (LuNE, NCI-H660) and organoids (LuCaP145.2) with control or *FOXA2* KD. Data shown are from one of two (*n* = 2) independent experiments. LuNE cells—stable LNCaP+FOXA2 cells. **c**, FOXA2 Co-IP showing its interaction with NKX2-1 protein in NEPC cell line NCI-H660. Data shown are from one of three (*n* = 3) independent experiments. **d**, Heatmap showing AR, NKX2-1 and FOXA2 occupancy at previously reported[11] NE-CREs (*n* = 14,985) and Ad-CREs (*n* = 4,338) in LNCaP or NEPC models (NCI-H660, LuCaP93, LuCap145.1, LuCaP145.2). Scale bar—enrichment intensity. **e**, Heatmap showing active (H3K27ac) enhancer (H3K4me1) and promoter (H3K4me3) marks at NKX2-1-only, FOXA2-only and shared binding sites in LuCaP145.1. Scale bar—enrichment intensity. **f**, APA plots of FOXA2- and NKX2-1-anchored E–P loops identified in D28 in time-course LNCaP+FOXA2 Hi-C samples. The APA score is annotated within each plot. Scale bars indicate log₂ of observed over expected enrichment. **g**, *NKX2-1* KD abolished FOXA2-induced NET. LNCaP cells were co-infected with FOXA2 virus along with either sg*NC* or sg*NKX2-1* virus. The infected cells were collected at the indicated time points for WB analyses of luminal and NE markers. The red star indicates a nonspecific band. Data shown are from one of three (*n* = 3) independent experiments. **h**, Heatmap showing FOXA2, NKX2-1, H3K27ac and H3K4me1 ChIP–seq around (±2 kb) the six ATAC–seq peak clusters in LNCaP+FOXA2 cells, with or without *NKX2-1* KD, at indicated time points. Peaks within each cluster were separately sorted by FOXA2 ChIP–seq intensity. Scale bar—enrichment intensity.

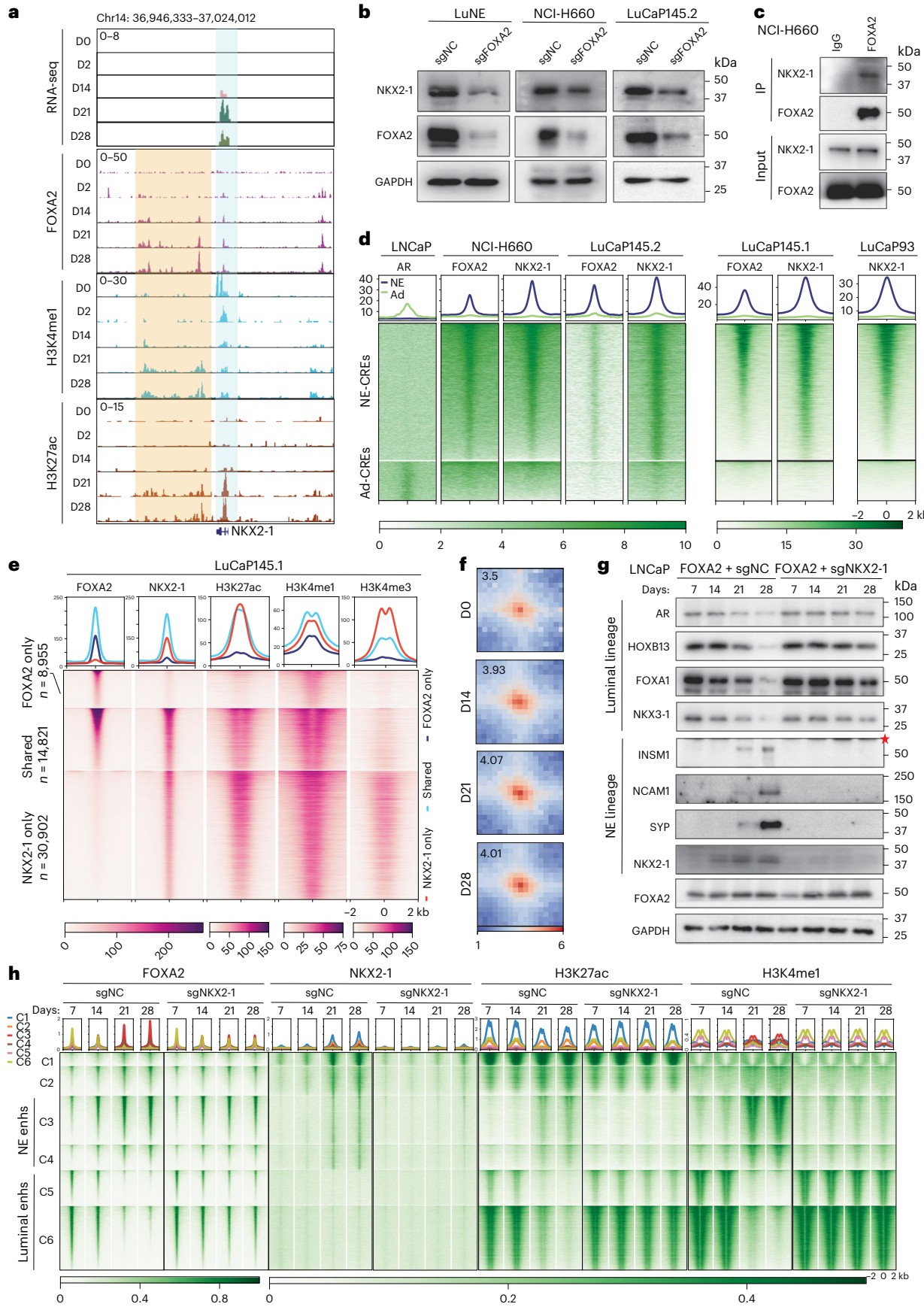

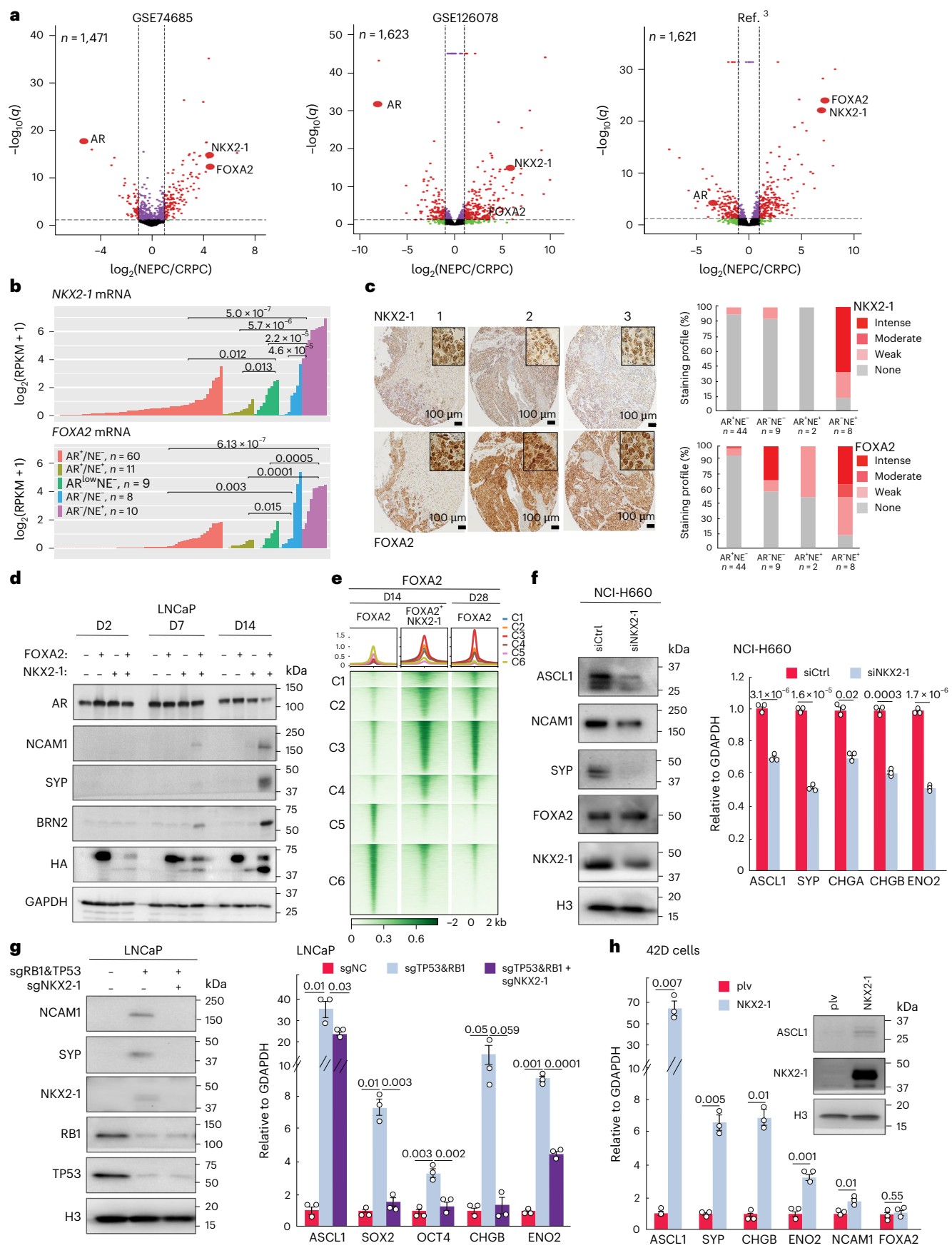

**Fig. 5 | NKX2-1 is highly expressed in NEPC tumors and accelerates NET of PCa.**
**a**, Volcano plots showing human TFs differentially expressed in CRPC versus NEPC. *X* axis represents log$_2$ of FC, while the *y* axis shows −log$_{10}$ of *q* values. Each dot represents a TF, red dots−log$_2$(FC) ≥ 1 and adjusted *P* < 0.05. Adjusted *P* values by Wald test with Benjamini−Hochberg correction. **b**, *NKX2-1* and *FOXA2* gene expression in human PCa samples with distinct expression of AR and NE genes. *P* values by two-sided Wilcoxon test. **c**, IHC of NKX2-1 and FOXA2 proteins in clinical CRPC TMAs. Representative IHC images (*n* = 3) are shown in ×10 with the insets shown in ×40 (left). Scale bar = 100 μm. IHC staining intensities in AR$^+$NE$^−$, AR$^−$NE$^−$, AR$^+$NE$^+$ and AR$^−$NE$^+$ samples are quantified on the right. **d**, Concomitant NKX2-1 OE accelerated FOXA2-driven NET. LNCaP cells were infected with FOXA2, NKX2-1 or both. Cells were collected at the indicated time points and analyzed by WB (*n* = 3). **e**, Heatmap showing FOXA2 ChIP−seq in LNCaP cells with OE of FOXA2 alone, or FOXA2 and NKX2-1 at D14 or D28. Peaks were centered around (±2 kb) the six ATAC−seq clusters and sorted by FOXA2 ChIP−seq intensity. Scale bar−enrichment intensity. **f**, WB (left, *n* = 3) and RT−PCR (right) showing *NKX2-1* KD decreases the expression of NE lineage markers in NCI-H660. NCI-H660 cells were transfected with control or *NKX2-1* siRNAs and collected on day 5 after transfection for WB and RT−PCR analyses. RT−PCR data were normalized to *GAPDH*. Shown are the mean ± s.e.m. of technical replicates from one of three (*n* = 3) independent experiments. *P* values by two-sided *t* test. **g**, WB (left, *n* = 3) and RT−PCR (right) showing *NKX2-1* KD abolishes *RB1*/TP53-KD-induced NET. LNCaP cells were co-infected with sg*RB1* and sg*TPS3* virus along with either sg*NC* or sg*NKX2-1* virus. The infected cells were collected at 4 weeks after infection for analyses of NE and stem cell lineage markers. Data of RT−PCR are shown as in **f**. **h**, RT−PCR (left) and WB (right, *n* = 3) showing NKX2-1 OE promotes NET in ENZ-resistant AR$^+$/PSA$^−$ cell line 42D. The infected cells were collected at 4 weeks after infection for analyses of NE lineage markers. Data of RT−PCR are shown as in **f**.

Supplementary Note 2). There were also some constant peaks, such as C1, that predominantly bound at promoter regions and enriched for motifs of ubiquitous TFs, such as SP1, that regulate housekeeping gene expression. Altogether, these results show a remarkable switch of chromatin accessibility from luminal-to-NE enhancers from D0 to D28 of NET, in agreement with transcriptional reprogramming and 3D chromatin rewiring.

Next, we wondered whether we could capture individual transitioning cells with intermediate transcriptome and chromatin states. Using single-cell multiome (single-cell RNA-seq (scRNA-seq) and scATAC−seq), we analyzed 2,110 D0, 1,739 D14 and 2,344 D21 LNCaP+FOXA2 cells, whose transcriptome, respectively, aligned with clinical primary PCa, CRPC and NEPC, as expected[46] (Extended Data Fig. 3d,e). Uniform manifold approximation and projection (UMAP) analyses revealed two distinct clusters that comprised primarily D0 and D21 cells that expressed high levels of luminal and NE genes, respectively (Fig. 3c–e and Extended Data Fig. 3f). Of note, most of the D14 and some of the D21 cells still belonged to the D0 cluster. These cells remained *AR*-positive but had a reduced AR signature (ARS) score and started to express some NE signature (NES) genes (Supplementary Note 2), indicating an intermediate transcriptome. Furthermore, RNA velocity analyses of D14 cells revealed a strong tendency for the *AR*$^+$/ARS$^+$ (*KLK3*$^+$) cells to become *AR*$^+$/ARS$^−$ (*KLK3*-low; Fig. 3g), supporting the concept of clonal transformation. UMAP analyses of matched scATAC−seq data likewise confirmed the following two clusters of cells: an NE cluster of D21 cells and a luminal cluster with mixed D0, D14 and some D21 cells (Fig. 3h) and validated that most D14 and some D21 cells had an intermediate chromatin state that was consistent with their transitioning transcriptome (Fig. 3i,j and Supplementary Note 2). Moreover, clonal variance analyses using scRNA-seq data verified that LNCaP, like many cancer cell lines[47–49], contained multiple clones and revealed two major clones, clone 1 and clone 2, which, respectively, dominated D0 and D28 cells. Critically, both clones went through NET from D0 to D28, with clone 2 cells becoming fully transformed and predominant after D21 (Extended Data Fig. 4 and Supplementary Note 3), supporting a mixed model of clonal transformation and expansion underlying NET of PCa.

## NKX2-1 induction is required for FOXA2-driven NET
FOXA2 is a pioneering factor that requires the coordination of lineage-specific TF for enhancer activation[50]. Focusing on neural TFs that were turned on during NET, we found multiple FOXA2-binding events at regulatory elements of *NKX2-1*, as early as D2 following FOXA2 OE, followed by an increase of H3K4me1 at D14 and of H3K27ac at D21, resulting in *NKX2-1* upregulation (Fig. 4a and Extended Data Fig. 5a). In concordance, *FOXA2* KD directly reduced NKX2-1 expression in various NEPC models, including LNCaP cells with stable FOXA2 OE (>28 days), termed LuNE cells hereafter (Fig. 4b and Extended Data Fig. 5b). Moreover, Hi-C revealed increased interactions between FOXA2-bound NKX2-1 enhancers and the *NKX2-1* promoter over LNCaP+FOXA2 time course, which were verified in NCI-H660 and NEPC PDXs (Extended Data Fig. 5c,d), suggesting that FOXA2 binds to *NKX2-1* enhancer to directly induce *NKX2-1* transcription via E–P looping.

Next, we attempted to determine whether NKX2-1 induction is required for FOXA2-driven NET. We observed strong interactions between FOXA2 and NKX2-1 proteins and substantial co-occupancy at NE enhancers[11] (Fig. 4c,d). A majority of FOXA2-binding sites were co-occupied by NKX2-1 and marked by H3K27ac and H3K4me1, suggesting active NE enhancers (Fig. 4e and Extended Data Fig. 5e,f), whereas NKX2-1-only binding sites were enriched for promoter and marked by H3K4me3 (Fig. 4e and Extended Data Fig. 5e–g). Hi-C revealed a substantial increase in the intensity and number of chromatin loops anchored at FOXA2 and NKX2-1 binding sites, indicating that NKX2-1, which primarily binds at promoters, interacts with enhancer-bound FOXA2 through E–P looping to propel NE enhancer priming and activation (Fig. 4f and Extended Data Fig. 5h). Indeed, KD of NKX2-1 abolished FOXA2-driven switches of luminal and NE marker gene expression (Fig. 4g). Mechanistically, FOXA2 primarily bound to luminal enhancers that were marked by H3K4me1 and H3K27ac and notably also at future NE enhancers immediately following OE, but gradually shifted from luminal-to-NE enhancers over time (Fig. 4h). By contrast, NKX2-1 formed strong and selective binding at NE enhancers starting at D21, accompanied by an abrupt luminal-to-NE enhancer switch of H3K27ac and H3K4me1. Critically, NKX2-1 KD not only abolished such changes of the enhancer marks but

**Fig. 6 | FOXA2 induces regional DNA demethylation at NE enhancers.**
**a**, IGV view of FOXA2 ChIP−seq (top), mCpG (middle) and single DNA molecules of mCpG and mA (bottom) around the *NKX2-1* gene in D2, D14 and D28 LNCaP+FOXA2 cells. Green box−CpG islands. Yellow box−*NKX2-1* promoter. **b**, FOXA2 DiMeLo-seq was performed in D28 LNCaP+FOXA2 cells, and mA and mCpG were called with a probability ≥0.5. Aggregate mA and mCpG curves (top) for each quartile were created with a 50-bp rolling window centered (±1 kb) at D28-specific FOXA2 peaks, which were sorted into four quartiles with q4 comprising the strongest peaks. Inset−a zoomed-in version of the mCpG curves. Heatmap (bottom) shows mA and mCpG on single DNA molecules. A and CG base density (scale bar at the bottom) across the 2 kb region of FOXA2 peaks within each quartile are shown in the 1D heatmaps. Color scales are shown as in **a**. **c**, Correlation plots of mA and mCpG counts on the same DNA molecules in D2, D14 and D28 LNCaP+FOXA2 cells. **d**, PCA analyses of methylation profiles of CRPC, NEPC PDX and D2, D14, D28 LNCaP+FOXA2 samples. DMRs identified between D2 and D28 were used for PCA analysis. **e**, IGV view of %mCpG in samples named on the left at *HOXB2* (top row) and *AR* promoter (bottom row) region. CpG islands are shown in green. **f**, FOXA2 DiMeLo-seq showing FOXA2 binding (mA/A) at previously reported[11] Ad-CREs and NE-CREs in NEPC PDX (LuCaP145.1 and LuCaP145.2) and cell line (NCI-H660). **g**, Correlation plots of mA and mCpG counts on the same DNA molecules in NEPC PDX (LuCaP145.1 and LuCaP145.2) and cell line (NCI-H660). q1 comprises the weakest peaks, and q4 comprises the strongest peaks of FOXA2 based on FOXA2 ChIP−seq signals in respective samples. IGV, integrative genomics viewer; q1, quartile 1; q4, quartile 4.

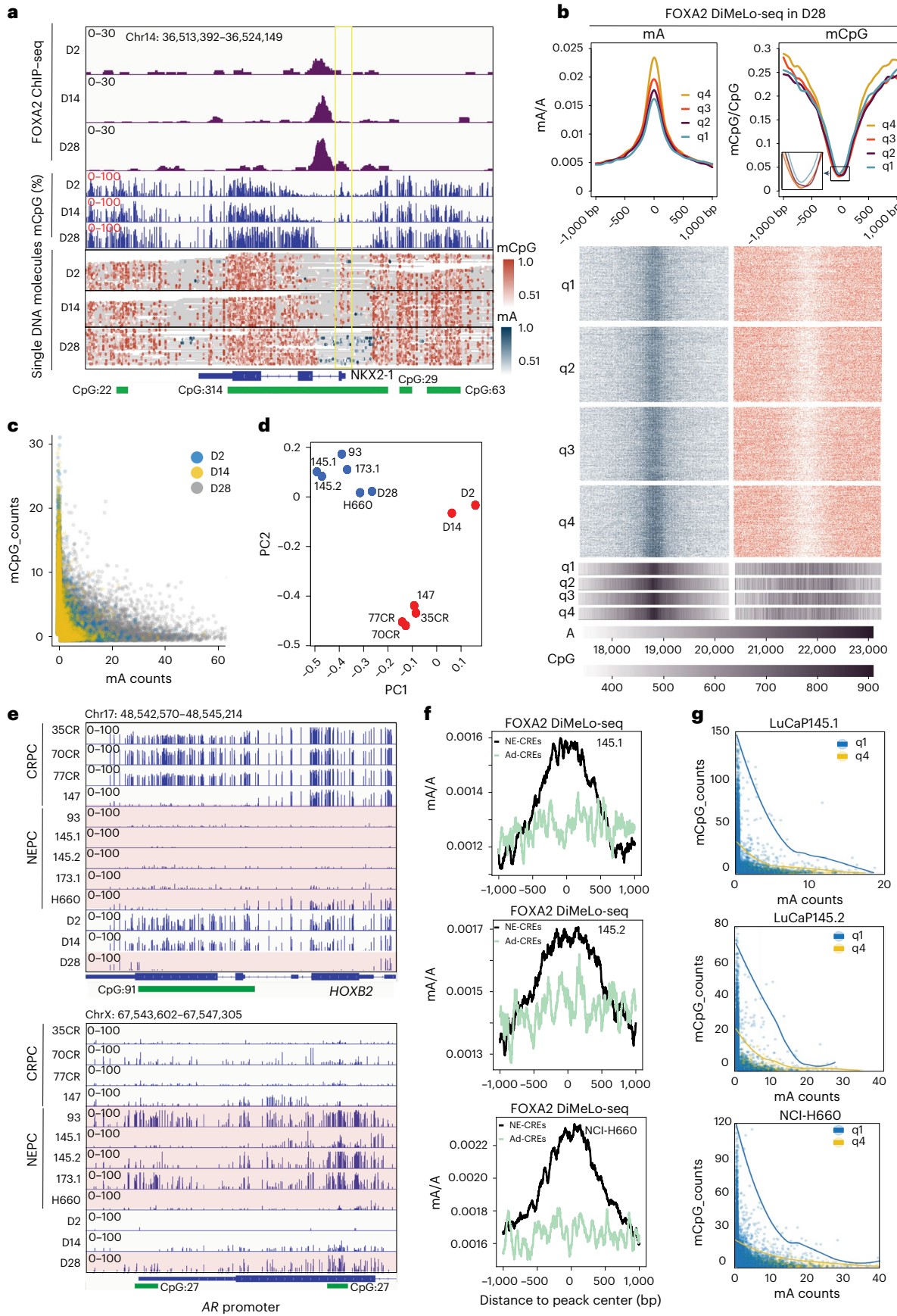

also halted the shift of FOXA2 cistrome from luminal-to-NE enhancers, suggesting a feed-forward loop between FOXA2 and NKX2-1 binding (Fig. 4h). Therefore, FOXA2 initiates NE enhancers but requires the induction of NKX2-1 to complete NET.

## NKX2-1 is upregulated in NEPC and accelerates NET of PCa

To examine the clinical relevance of our findings, we analyzed publicly available PCa datasets[3,40,51] and found *NKX2-1* and *FOXA2* to be significantly co-induced in NEPC tumors (Fig. 5a,b and Extended Data Fig. 6a). Concordantly, H3K27ac was markedly increased while DNA methylation drastically reduced around the *NKX2-1* gene in NEPC versus CRPC tumors, whereas genomic alterations of *NKX2-1* were infrequent (Extended Data Fig. 6b–d). Significantly, tissue microarrays (TMA) analyses validated strongly elevated NKX2-1 and FOXA2 proteins in AR⁻/NE⁺ tumors relative to other CRPC subtypes (Fig. 5c). Moreover, while FOXA2 was also increased in some AR⁻/NE⁻ tumors, strong NKX2-1 staining was restricted to AR⁻/NE⁺ tumors, indicating higher specificity. Of noteworthy, some NKX2-1-high tumors lacked FOXA2 expression, suggesting that NKX2-1 might be induced by and collaborate with other cofactors such as ASCL1 to regulate NEPC[11] (Extended Data Fig. 7a–f and Supplementary Note 4).

We next attempted to investigate directly whether NKX2-1 co-expression facilitates FOXA2 in driving NET. We found that concurrent OE of NKX2-1 and FOXA2 drastically accelerated NET of LNCaP cells compared to FOXA2 OE alone, whereas NKX2-1 OE alone failed to drive NET (Fig. 5d,e and Extended Data Fig. 7g). To further assess if NKX2-1 is a general regulator of NEPC, we performed *NKX2-1* KD in NCI-H660 and observed an ablation of NE marker genes at both protein and mRNA levels (Fig. 5f). Likewise, NKX2-1 was also upregulated in the NE-like LNCaP cells with *RB1/TP53*-KD, and its depletion abolished NE marker expression (Fig. 5g). On the other hand, NKX2-1 was undetectable in 42D, an AR⁺/PSA⁻ cell line derived in vivo upon resistance to AR antagonist enzalutamide (ENZ)[18], and its OE remarkably increased the expression of NE markers despite low FOXA2 in these cells (Fig. 5h). Taken together, our data demonstrated that NKX2-1 is selectively upregulated in NEPC and accelerates NET through collaboration with FOXA2 and other cofactors.

## FOXA2 induces regional DNA demethylation around NE enhancers

We next sought to understand how FOXA2, after initial binding to the enhancers, prepares the chromatin for NET. Using the directed methylation with long-read sequencing (DiMeLo-seq) approach[52], we observed sharp peaks of CpG methylation (mCpG) right at the *NKX2-1* promoter in D2 cells, which were slightly decreased at D14 and completely lost at D28, when abundant N6-methyl-deoxyadenosine (mA), reflecting FOXA2 binding, was detected (Fig. 6a). Globally, strong D28 FOXA2-binding peaks showed much higher mA levels in D28 cells,

as expected, but, critically, much less DNA methylation, indicated by deep and broad DNA methylation valleys (Fig. 6b). As controls, these peaks lacked mA but showed strong DNA methylation in D2 cells (Extended Data Fig. 8a). Furthermore, at the level of single DNA molecules, there was also a mutually exclusive pattern of mA and mCpG at each time point, with D28 cells containing DNA molecules with the highest levels of mA and the lowest mCpG (Fig. 6c), suggesting that FOXA2 binding led to regional DNA demethylation at NE enhancers over the time course.

To evaluate the clinical relevance of these findings, we first identified differentially methylated regions (DMRs) between D28 and D2 LNCaP+FOXA2 cells. As expected, genes with hyper-DMR in D28 were involved in epithelial cell function, whereas hypo-DMR genes were significantly enriched in embryo development and neurogenesis, and they clustered D2 and D28 cells with CRPC and NEPC tumors, respectively (Extended Data Fig. 8b,c). To correlate DNA methylation with FOXA2 binding, we performed DiMeLo-seq in a number of PDX tumors and observed that DNA methylation was significantly lower at Ad-CREs and NE-CREs, respectively, in CRPC and NEPC PDXs, being consistent with their respective luminal and NE enhancer activation (Extended Data Fig. 8d,e). NEPC tumors shared methylation profiles with D28 cells, which were clearly separated from D2 and D14 cells and CRPC tumors (Fig. 6d). For instance, the *HOXB2* gene, induced in D28 cells, was marked by dense mCpG in D2 and CRPC PDXs but became completely demethylated in D28 and NEPC PDXs, whereas an opposite DNA methylation remodeling was observed at the AR promoter (Fig. 6e). Significantly, mA, reflecting FOXA2 binding, was substantially more enriched at NE-CREs than Ad-CREs in NEPC PDX tumors and was mutually exclusive to mCpG (Fig. 6f, g).

## p300/CBP are critical therapeutic targets in NEPC

To investigate how NE enhancers are ultimately activated, we performed mass spectrometry (MS) and found that FOXA2 interacts with histone acetyltransferases CREBBP (CBP) and EP300 (p300), which was further confirmed by co-immunoprecipitation (Co-IP; Fig. 7a–c). Moreover, chromatin immunoprecipitation followed by sequencing (ChIP–seq) demonstrated that p300 and consequently H3K27ac were strongly enriched at FOXA2 and NKX2-1 co-occupied NE enhancers, which were abolished by either *FOXA2* or *NKX2-1* KD (Fig. 7d and Extended Data Fig. 9a), suggesting that both FOXA2 and NKX2-1 are required for p300 recruitment to NE enhancers to catalyze H3K27ac. Not surprisingly, the depletion of p300 and, to some extent, CBP greatly reduced H3K27ac at these enhancer elements. Significantly, CCS1477, an orally active and selective inhibitor of p300/CBP bromodomain[53], completely eliminated H3K27ac at NE enhancers, many of which were SEs that regulate lineage-specific TFs and pro-oncogenes (Extended Data Fig. 9b–f and Supplementary Note 5). To determine whether p300/CBP are required for FOXA2/NKX2-1-mediated transcriptional program, we performed RNA-seq in LuNE cells with KD of control, *p300*, *CBP* or both

---

**Fig. 7 | NKX2-1 and FOXA2 recruit p300/CBP to activate NE enhancers and induce NEPC tumor growth, which can be abolished by p300/CBP inhibition. a**, Volcano plot showing FOXA2-interacting proteins in LuNE cells by MS (pooled data from three Co-IP replicates). The *x* axis represents log₂(FC), and the *y* axis represents −log₁₀*P* by two-sided *t* test. Each dot represents a protein. **b**, Co-IP showing that FOXA2 and NKX2-1 interact with CBP and p300 in LuNE cells. Data shown are from one of three (*n* = 3) independent experiments. **c**, Co-IP showing that NKX2-1 interacts with FOXA2 and p300 in NCI-H660 cells. Data are shown as in **b** (*n* = 3). **d**, Heatmaps showing indicated ChIP–seq intensity centered (±2 kb) around the five clusters of FOXA2-binding sites identified in the time-course LNCaP+FOXA2 cells. Scale bar −enrichment intensity. **e**, Heatmap showing that genes induced by FOXA2 and NKX2-1 (FC ≥ 2 and adjusted *P* < 0.05) are regulated by p300/CBP in LuNE cells. Data shown is the log₂(FC) of sg*NKX2-1*, sg*FOXA2* or sg*FOXA2*+sg*NKX2-1* relative to sg*NC*, as well as of sh*p300*, sh*CBP* or sh*p300*+sh*CBP* to sh*Ctrl*. Adjusted *P* values by Wald test with Benjamini–Hochberg correction. **f**, LuNE cells were treated with DMSO or CCS1477 (250 nM)

for 72 h. Differentially expressed genes were identified by DESeq2 with FC ≥ 2, adjusted *P* < 0.05. Color bar−*z* score. Adjusted *P* values are calculated as in **e**. **g**, GO analysis of CCS1477-repressed genes identified in **f**. Top enriched molecular concepts are shown on the *y* axis, while the *x* axis indicates enrichment. *P* values by one-sided hypergeometric test. **h**, Gene set enrichment analysis (GSEA) showing enrichment of SE-associated genes in LuNE cells treated with DMSO or CCS1477. *P* value by two-sided permutation test with Benjamini–Hochberg correction. **i,j**, Tumor growth curve (**i**) and weight (**j**) of LuNE xenograft tumors treated with vehicle or CCS1477 (*n* = 6 mice per group). Data are mean ± s.e.m, and *P* values by two-sided *t* test. **k,l**, Tumor growth curve (**k**) and weight (**l**) of LuCaP145.2 xenograft tumors treated with vehicle or CCS1477 (*n* = 6 mice per group in **k**, *n* = 7 tumors per group in **l**). Data are mean ± s.d, and *P* values by two-sided *t* test. **m**, Representative IHC images (*n* = 6 mice per group) of H3K27ac, SOX2, NKX2-1 and Ki-67 in LuCaP145.2 xenograft tumors treated with vehicle or CCS1477. Scale bar = 30 µm. Insets are shown at higher magnification (×40).

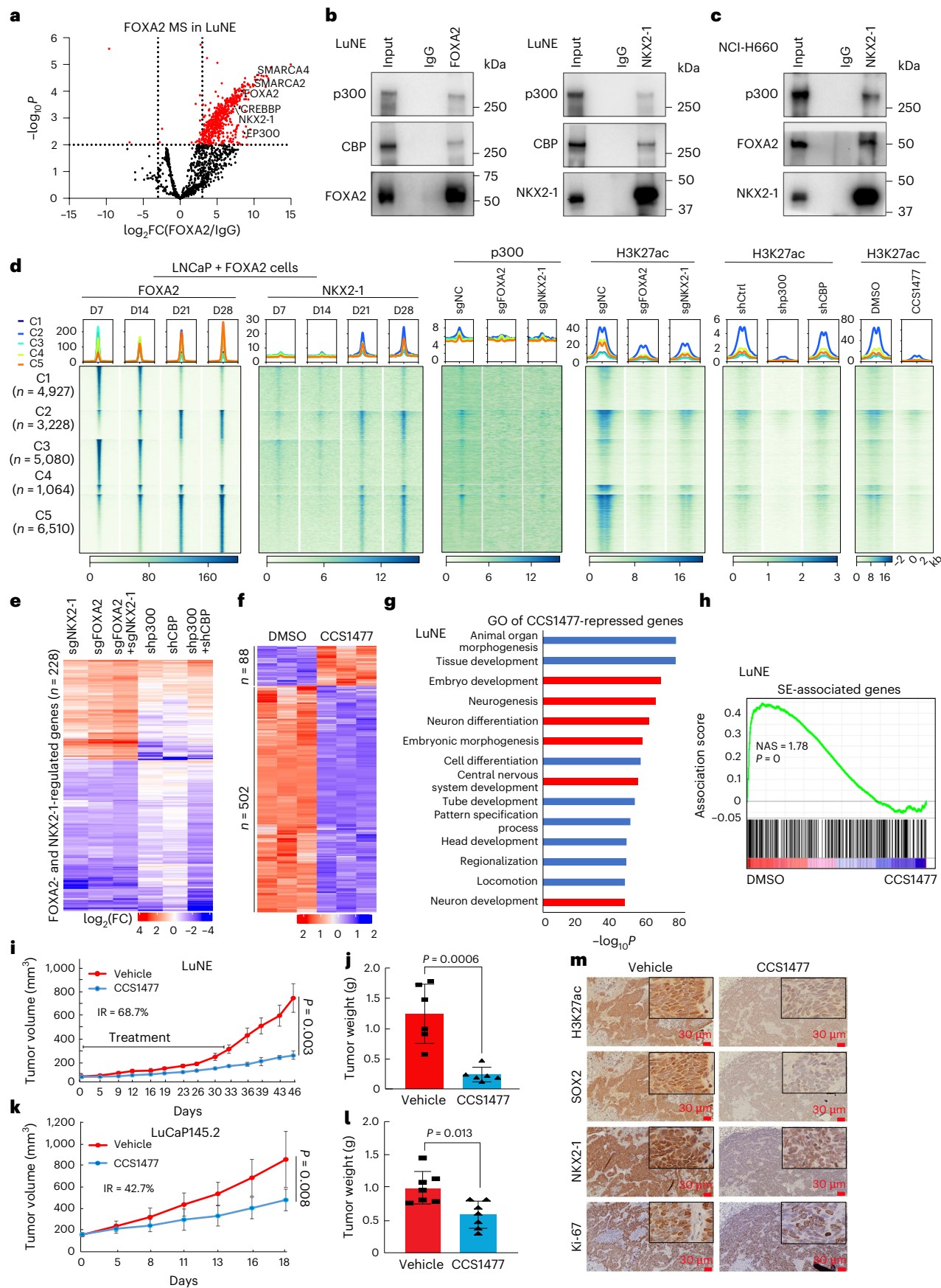

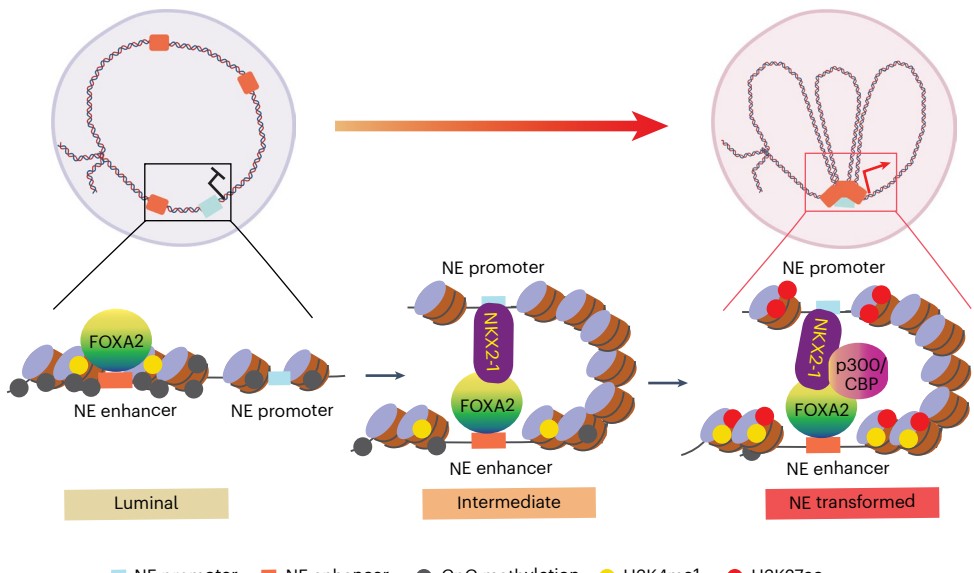

**Fig. 8 | Epigenetic remodeling and 3D chromatin reorganization governed by NKX2-1 and FOXA2 drive NE transdifferentiation of PCa.** As a pioneer factor, ectopic FOXA2, once overexpressed in luminal cells, is capable of binding to inaccessible NE lineage enhancers. It induces regional DNA demethylation and increases local chromatin accessibility, thereby activating the expression of NE lineage TFs, such as NKX2-1. Once induced, NKX2-1 preferentially binds to gene promoters and interacts with enhancer-bound FOXA2 through chromatin looping (intermediate stage). This interaction further strengthens FOXA2 occupancy at NE enhancers. Together, FOXA2 and NKX2-1 recruit p300/CBP, which catalyze H3K27 acetylation, further promoting 3D chromatin reorganization, activating NE enhancers and driving the expression of NE lineage-specific transcriptional programs.

(Extended Data Fig. 9g). Notably, most FOXA2/NKX2-1-regulated genes were similarly regulated by *CBP* or *p300* KD (Fig. 7e), with their induced genes involved in neurogenesis and development (Extended Data Fig. 9h,i). CCS1477 treatment of LuNE cells led to the repression of a large number of genes that were enriched for development, differentiation and neurogenesis and dependent on FOXA2, NKX2-1 and CBP/p300 (Fig. 7f,g and Extended Data Fig. 9j). Of note, LuNE cell SE-associated genes were significantly suppressed by CCS1477, which, when used in LNCaP cells, inhibited a distinct set of genes associated with luminal SEs, suggesting its ability to preferentially target lineage-specific TFs and proto-oncogenes controlled by their respective SEs[54,55] (Fig. 7h, Supplementary Fig. 2a,b and Supplementary Note 5).

Finally, we evaluated the efficacy of CCS1477 in inhibiting NEPC. Depletion of *NKX2-1*, *FOXA2*, *p300* or, to a lesser extent, *CBP* substantially inhibited LuNE cell colony formation, supporting the essentiality of their target genes (Extended Data Fig. 10a). Consistently, CCS1477 massively inhibited colony formation of LuNE and LNCaP cells by inducing cell cycle arrest, without affecting the benign and normal prostate cell lines (Extended Data Fig. 10b–e and Supplementary Note 5). Moreover, CCS1477 achieved a similarly strong growth inhibition of additional NEPC cells, LuCaP145.2 and NCI-H660 (Extended Data Fig. 10f). To examine the efficacy of CCS1477 in targeting NEPC tumors in vivo, we performed subcutaneous injection of LuNE cells in severe combined immunodeficient (SCID) mice. We started to see apparent growth reduction after 3 weeks of treatment (Fig. 7i). Interestingly, despite treatment ending at day 33, tumor inhibition continued, with an inhibition rate (IR) of 68.7% (*P* = 0.003) at day 46 when the experiment was terminated. There was a remarkable difference in endpoint tumor weight (*P* = 0.0006) and volume, while the body weights of the mice were not significantly affected (Fig. 7j and Extended Data Fig. 10g,h). CCS1477 drastically reduced H3K27ac and NE TFs NKX2-1 and SOX2, as expected, but it was unable to rescue luminal markers AR and PSA (Extended Data Fig. 10i,j). Similar tumor-inhibitory effects of CCS1477 were also observed in NEPC PDX (LuCaP145.2) tumors (Fig. 7k–m and Extended Data Fig. 10k,l). In summary, these data support that p300/CBP has an important role in mediating NEPC SE addition and growth, which can be effectively mitigated by CCS1477.

## Discussion

NEPC is a subtype of CRPC that exhibits NE features and displays distinct epigenetic and transcriptional programs compared to most CRPC tumors. Consistent with this, our data showed that the 3D genome conformation was reorganized in NEPC compared to CRPC PDX tumors, despite substantial heterogeneity within each group, especially among NEPC samples. This chromatin reorganization is marked by the loss of luminal E–P loops and the formation of new E–P loops and TADs that favor the NE program, which was recapitulated by isogenic LNCaP cells undergoing NET upon FOXA2 OE. New E–P loops mediated by the interactions between enhancer-bound FOXA2 and promoter-bound NKX2-1 emerged at D14, following the induction of NKX2-1 expression, and drastically increased at D21 and D28. This may be further strengthened as the NE enhancers get activated by p300/CBP (Fig. 8), which is known to promote the formation of subnuclear condensates that mediate long-range DNA interactions[56,57].

Closely aligned with the 3D chromatin reorganization are the changes in the epigenetic landscape in both PDX tumors and LNCaP+FOXA2 cells undergoing NET. We found that FOXA2 OE leads to regional DNA demethylation around its binding sites, which is consistent with recent reports of FOXA2 function in endodermal lineage intermediates to catalyze regional DNA demethylation that is critical for liver and pancreatic endocrine cell specification[58,59]. We believe regional DNA demethylation at NE enhancers following FOXA2 binding is a critical step toward subsequent deposition of H3K4me1 to prime these lineage-specific enhancers for following activation by lineage-inductive TFs, as previously observed during embryonic development[50,59–64]. Our study highlighted the important role of FOXA2 in initiating an epigenetic memory of the NE cellular identity. While how FOXA2 mediates DNA demethylation and how it leads to enhancer priming are beyond the scope of the current study, FOXA proteins have been shown to recruit TET1 for DNA demethylation[65] and MLL3 for H3K4me1 deposition at target enhancers in other cell types[60].

We report an essential feed-forward loop, orchestrated by lineage-inductive pioneer factors and TFs, for NET of PCa. *NKX2-1* is a direct transcriptional target of FOXA2. Once NKX2-1 is induced by FOXA2 OE, it binds to promoters and interacts with enhancer-bound FOXA2

by forming E–P loops. NKX2-1 binding further stabilizes FOXA2 recruitment to NE enhancers, facilitating subsequent NE enhancer priming and activation (Fig. 8), which is consistent with recent findings for its role in enforcing organ cell type-specific gene expression during pancreatic, hepatic and lung development[66]. Although our current study deciphers this fundamental mechanism of NET using the LNCaP+FOXA2 model, the fact that FOXA2 and NKX2-1 are co-expressed in a majority of patients with NEPC might suggest its generalizability. We did observe a more specific and broader induction of NKX2-1 in NEPC tumors than FOXA2, suggesting that NKX2-1 might collaborate with and function through other pioneering factors. For instance, a previous study has noted that ectopic NKX2-1 could enhance the ability of ASCL1 to redistribute FOXA1 to NE enhancers and induce NET in LNCaP cells that are negative for FOXA2 (ref. 11). Indeed, we found significantly overlapping binding sites between NKX2-1 and ASCL1 in FOXA2-low PDX (Extended Data Fig. 7e). Therefore, NKX2-1 may cooperate with ASCL1 in FOXA2-low NEPC and with both pioneer factors in NEPC tumors expressing all three proteins.

Presently, there are no effective treatments for NEPC except platinum-based chemotherapy. Our study delineated that p300/CBP is essential for FOXA2 and NKX2-1-mediated NE gene expression and NEPC growth. We found that CCS1477, a p300/CBP inhibitor that targets AR and c-Myc in CRPC and is currently in clinical trial[53], also significantly inhibited NEPC cell growth. CCS1477 inhibits lineage-specific SEs to suppress cancer type-specific TFs and proto-oncogenes that are distinct in LNCaP and LuNE cells. CCS1477 thus reduced the growth of both CRPC and NEPC, but not benign prostate cells, likely due to PCa being enhancer-addicted[67]. Consistent with our findings, a recent study reported that CCS1477 concurrently inhibits two distinct hematological malignancies—myeloid leukemia and multiple myeloma—by impairing their respective lineage-specific enhancer activities[68]. However, CCS1477 was unable to convert NEPC tumors back to adenocarcinoma, likely due to the chromatin/epigenetic state of the terminally transformed NEPC cells being fairly stable. Therefore, our results suggest the therapeutic potential of p300/CBP inhibitors in simultaneously targeting CRPC and NEPC and indicate the necessity of early detection of NET to prevent/reverse NEPC progression.

## Online content

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

## Methods

### Ethics statement

Our research complies with all relevant ethical regulations. Mouse handling and experimental procedures were approved by the Institutional Animal Care and Use Committee at Northwestern University in accordance with the US National Institutes of Health Guidelines for the Care and Use of Laboratory Animals and the Animal Welfare Act. TMAs containing metastatic CRPC specimens were obtained as part of the University of Washington Medical Center Prostate Cancer Donor Program, which is approved by the University of Washington Institutional Review Board.

### Cell lines, chemical reagents and antibodies

PCa cell lines LNCaP, 22Rv1, DU145, NCI-H660, human benign prostatic hyperplasia cell line BPH-1, human normal prostate epithelial cell line RWPE-1 and human embryonic kidney cell line HEK293T cells were obtained from the American Type Culture Collection (ATCC). LNCaP, 22Rv1, DU145, BPH-1 and HEK293T were maintained in either RPMI1640 or Dulbecco's modified Eagle's medium with 10% FBS and 1% penicillin and streptomycin. RWPE-1 cells were maintained in keratinocyte serum-free medium with 0.05 mg ml$^{-1}$ BPE and 5 ng ml$^{-1}$ epidermal growth factor. NCI-H660 cells were maintained in ATCC-formulated RPMI1640 medium (30-2001) with 0.005 mg ml$^{-1}$ insulin, 0.01 mg ml$^{-1}$ transferrin, 30 nM sodium selenite, 10 nM hydrocortisone, 10 nM β-estradiol and 5% FBS. The 42D cells were maintained in RPMI1640 with 5% FBS, 1% penicillin and streptomycin and 10 μM ENZ for all experiments. All the cells were authenticated within 6 months of growth, and cells under culture were frequently tested for potential mycoplasma contamination. CCS1477 (CT-CCS1477) was purchased from Chemietek. All antibodies used in this study are listed in Supplementary Table 2.

### Constructs and lentivirus infection

Human *FOXA2* coding region (CDS) was first amplified using cDNA from PC-3 cells as a template. *NKX2-1* and mouse *Foxa2* CDS were amplified using *NKX2-1* (Addgene, 119173) and m*Foxa2* (Addgene, 33014) as templates, respectively, and then cloned into plenti CMV Neo DEST (705-1) (Addgene, 17392) vector by In-Fusion HD Cloning Kits (TaKaRa, 638948). *FOXA2*, *NKX2-1*, *RB1* and *TP53* gRNAs were cloned into lentiCRISPR v2 (Addgene, 52961) vector. The shRNAs targeting CBP and p300 were cloned into pLKO.1-TRC lentiviral vector (Addgene, 10878). si*NKX2-1*1 (s224731), si*NKX2-1* 2 (s14152) and si*ASCL1* (s1657) were purchased from Thermo Fisher Scientific. The primers and oligonucleotides used in this study are listed in Supplementary Table 3, and all the plasmids were verified by Sanger sequencing. For details on lentivirus generation and infection, see Supplementary Method 1.

### Co-IP

Nuclear fraction was used for all Co-IP experiments in this study. Nuclear proteins were isolated as previously described with some modifications[42,69]. For detailed, step-by-step protocols, see Supplementary Method 2. All antibodies used in WB and Co-IP are listed in Supplementary Table 2.

### 3D organoid cell culture and colony formation assay

LuCaP145.2 and LuCaP93 PDX-derived organoids were generated as previously described[70]. For detailed, step-by-step protocols, see Supplementary Method 3. For details on the colony formation assay, see Supplementary Method 4.

### MS analysis

Chromatin fraction was used for the MS experiments in this study. Chromatin proteins were isolated as previously described with some modifications[42,69]. For detailed, step-by-step protocols, see Supplementary Method 5. MS analysis was performed using the Orbitrap Velos Pro system. SEQUEST was used for protein identification and peptide sequencing. The data in Fig. 7a were analyzed as previously[71]. Briefly, the sum of all peptide intensities for each protein in triplicate was log$_2$-transformed and normalized by the average. Missing peptide intensities were imputed from a Gaussian distribution with a width of 0.3, centered at the sample distribution mean minus 2.5 times the sample s.d. The criterion for statistically significant differential interactions was determined using an unpaired two-sided *t* test with a *P* value < 0.01 and a fold change (FC; FOXA2 versus IgG) greater than 8.

### RNA extraction, RT–qPCR and RNA-seq

RNA extraction, RT–qPCR and RNA-seq were performed as previously with some modifications[42,69]. For detailed, step-by-step protocols, see Supplementary Method 6. All primers used in RT–PCR analysis are listed in Supplementary Table 3.

### ChIP, ChIP–seq and ATAC–seq

ChIP, ChIP–seq and ATAC–seq were performed using the previously described protocol with some modifications[42,69]. For detailed, step-by-step protocols, see Supplementary Method 7. The information on antibodies used in ChIP is listed in Supplementary Table 2.

### Bulk RNA-seq analysis

Human PCa cell RNA-seq reads were mapped to the National Center for Biotechnology Information human genome GRCh38. Raw counts of genes were calculated by STAR. Fragments per kilobase of transcript per million mapped reads (FPKM) values were calculated by an in-house Perl script. The details for differential gene expression analysis were summarized here. Briefly, as Fig. 1c,d data were derived from the public data and did not have raw count data, which is required for DESeq2, we used FPKM value for each gene and performed differential expression analyses using *t* tests between the two groups. Figure 2a used DESeq2 with the likelihood ratio test, considering the time-course nature of the data, whereas Figs. 5a (GSE126078 and ref. 3 datasets) and 7e,f used DESeq2 with the default Wald test in a pairwise manner. DESeq2 has built-in multiple-test corrections for all tests. Figure 5a (GSE74685 (microarray)) dataset used the Bioconductor limma package. The ARS and NES genes used in Fig. 2c were from a previous study[3]. The scatterplot was generated using Gene Set Variation Analysis (1.52.3)[72]. RNA-seq data of patients with PCa were downloaded from GSE126078 (ref. 40).

### ChIP–seq, SE and ATAC–seq analyses

For details on ChIP–seq, SE and ATAC–seq analyses, see Supplementary Method 8.

### Single-cell sequencing sample preparation and analysis

A total of 5,000 D0, D14 and D21 LNCaP+FOXA2 cells were used for a single-cell Multiome ATAC + Gene Expression assay, and the Multiome libraries were prepared using 10X Genomics Chromium Next GEM Single-Cell Multiome ATAC + Gene Expression Reagent Kits (1000285) per the manufacturer's protocol. Single-cell libraries were sequenced in a NovaSeq 6000 with an average depth of 30,000 reads per cell. Raw sequencing data were processed and aligned to hg38 using Cell Ranger ARC (2.0.2). Low-quality cells with low unique molecular identifier counts and high mitochondrial ratios were filtered out, and the sequence metrics were updated in Supplementary Table 4. Each time point was analyzed individually at the expression and accessibility modality and then combined across time points within each modality using Seurat (4.3.0) and Signac (1.6.0) in R (4.0.3). For each time point, we performed quality control on ATAC fragments, RNA counts, nucleosome signal and transcription start site enrichment metrics, resulting in 2,110, 1,739 and 2,344 cells for D0, D14 and D21, respectively. The RNA modality was normalized using SCTransform. ATAC peaks were called using MACS2 and quantified. Counts were normalized using latent semantic indexing. These processes were performed separately for each condition.

To correct for batch effects, integration of the expression modality was performed using the Seurat V3 integration pipeline, which uses reciprocal PCA to identify anchors with 3,000 integration features. UMAP was constructed using the first 30 PCs. To visualize the gene expression, log-normalized data were used. ARS and NES were calculated using 'AddModuleScore'. $AR^+$ was defined as ≥0.6, which is the mean of normalized AR expression in D21. $ARS^+$ and $NES^+$ cells are defined as cells with a signature score ≥0. Monocle3 (0.2.3.0) was used to learn the trajectory graph and calculate the pseudotime. Integration of the chromatin accessibility was done by first finding a common peak set between all the conditions, counting features and renormalizing. Integration anchors were found using 'rlsi' using 2:50 dimensions. ARS and NES for the chromatin modality were generated by identifying peaks located within 5 kb upstream or within the gene body of the respective genes. Scores were obtained using Signac's 'AddChromatinModule', which calculates chromVAR deviations.

Velocyto (v0.17.17) was used to generate spliced and unspliced counts for all cells. RNA velocity analysis was performed on a subset of previously identified high-quality cells. Genes included in the analysis were filtered based on detection levels, and the top 3,000 highly variable genes were selected and normalized. K-nearest neighbor (KNN) imputation was applied to 20 PCA dimensions. The 'constant velocity' assumption was used for velocity estimation. RNA velocities were projected onto the UMAP calculated in Seurat.

For details on the quantification of *FOXA2* and *NKX2-1* double-positive cells in D21, as well as the integrative analysis of LNCaP+FOXA2 scRNA-seq data with human PCa, see Supplementary Method 9.

## Copy number variation (CNV) analysis

Quality control of scRNA-seq and scATAC-seq data was performed as described above. CNV analyses of scRNA-seq data (D0, D14 and D21) were performed by CopyKAT (1.1.0)[73], which applies a Bayesian segmentation approach to estimate copy number profiles with 220 Kb genomic windows. Briefly, the raw count matrix is variance-stabilized and smoothed for outliers. The cell-line mode was enabled, meaning a synthetic baseline derived from data variations was used. Consensus chromosome breakpoints among single-cell clusters were detected, and final copy number profiles were calculated per cell. Raw gene expression matrices for each time point were provided to CopyKAT using default parameters with cell-line mode enabled. Cells in each time point that had CNV variations between −0.1 and 0.1 in more than 80% of the regions were excluded, resulting in 2,352, 2,138 and 4,558 cells for D0, D14 and D21, respectively. The matrices for resulting high-quality cells were then combined. Clones were identified using hierarchical clustering with the Ward D2 method and Euclidean distance. The heatmap was produced with heatmap.3 function.

## Single-cell nanopore RNA-seq (scNanoRNA-seq) sample preparation and analysis

scNanoRNA-seq was performed as previously described[74]. Briefly, barcoded full-length cDNAs of single cells were prepared using the 10X Genomics Next GEM Single-Cell 3′ gene expression kit with elongation time extending to 3 min to enrich longer molecules as previously described[75]. Then, the full-length cDNA libraries were prepared for nanopore long-read sequencing using the SQK-LSK114 ligation sequencing kit (Oxford Nanopore). The final sequencing was run on a PromethION flow cell (R10.4) with one sample per flow cell. Base calling was done using Guppy (v6.4.6). For details on single-nucleotide variant and germline mutation analyses in LNCaP+FOXA2 time-course cells (D0 and D28), see Supplementary Method 10.

## Hi-C sample preparation and analysis

FOXA2-D0, D14, D21 and D28 LNCaP cells and NCI-H660 cells were fixed with 2% formaldehyde (final concentration) at room temperature for 10 min, whereas tumor tissues from LuCaP PDX were fixed with 2%

formaldehyde at room temperature for 20 min, followed by adding 2.5 M glycine to a final concentration of 0.2 M to quench the reaction. Hi-C libraries were generated using the Arima-Hi-C[+] kit (Arima Genomics, A510008) per the manufacturer's protocol. The Hi-C libraries were sequenced using paired-end 2 × 150 bp reads on the Illumina NovaSeq 6000 or NovaSeq X Plus. The sequence metrics are provided in Supplementary Table 1.

Hi-C data were processed into .mcool maps with multiresolution using the runHiC (0.8.6) pipeline, with chromap as the aligner to hg38, and iterative correction and eigenvector decomposition (ICE)-normalization was applied. Maps were uploaded to and visualized on resgen.io, which relies on HiGlass. A/B compartments were called using cooltools (0.5.4) eigs_cis function with GC content phasing at 100 kb resolution. The first eigenvector was extracted for each sample, and bins with nonzero values across all samples were retained. The top 25% most variable bins were identified separately for the 9 samples in Extended Data Fig. 1a and the 17 samples in Extended Data Fig. 2g. Pearson correlations between the most variable eigenvectors were then calculated pairwise.

NEPC- and CRPC-enriched loops were identified using the diffMustache module in Mustache (Fig. 1a,b). Loops from each CRPC and NEPC PDX sample were called at 10 kb resolution with an FDR of 0.05 and a sparsity threshold (-st) of 0.8. These criteria were used to evaluate loop intensity as well as the local background to exclude artifacts. To identify NEPC-enriched loops, first, each NEPC PDX was compared to each of the four CRPC PDXs independently to identify differential loops with an FDR of 0.001. The differential loops shared in at least two of such pairwise NEPC–CRPC comparisons were then defined as gained loops for that NEPC PDX, and this process was done for all four NEPC PDXs. Finally, differential loops gained in two or more NEPC PDXs were defined as NEPC subtype-enriched loops. Analogous analysis was performed to identify CRPC subtype-enriched loops.

APAs to determine genome-wide loop intensity in Figs. 1a,b, 2d and 4f were made with coolpup.py (1.0.0). Loops called at 10 kb resolution were piled up and centered at a 50 kb × 50 kb matrix. The averaged loop intensity was normalized to the expected value. APA scores are the mean of the central 3 × 3 square of the matrix. Significance of the differences between NEPC-enriched loops and CRPC-enriched loops was performed using a Student's *t* test by comparing the APA scores of the NEPC samples to those of the CRPC samples.

Genes linked to NEPC- or CRPC-enriched loops were defined as genes within ±3 kb of loop anchors. Each gene's expression was obtained as publicly available FPKM values, and differential expression was determined with a *t* test between the two groups, in Fig. 1c,d. CRPC- or NEPC-enriched loops that overlap with CREs were defined as loops with either anchor having an overlap with CREs. Ad/NE-CREs were converted from hg19 to hg38 using UCSC's liftOver tool. Statistical tests of the significance of the overlap between subtype-enriched loops and CREs were calculated using a permutation test with 1,000 randomizations where the loop anchors were sampled from random regions within the same chromosomes. Enriched motifs were identified using these overlapping CREs with HOMER. FOXA2- and NKX2-1-anchored loops were annotated based on the presence of matching ChIP–seq binding sites within the anchors. Figure 4f shows the set of loops in D28 with FOXA2 and NKX2-1 at loop anchors, excluding any loops that overlap with those in D0.

## Reduced representation methylation sequencing using nanopore technology

Reduced representation methylation sequencing libraries were generated using the Oxford Nanopore protocol. Briefly, genomic DNA from LuCaP PDX was extracted using the Quick-DNA Miniprep Plus Kit (Zymo, D4068) and fragmented to an average size of 8 kb using g-TUBE (Covaris, 520079). DNA libraries were prepared with the Ligation Sequencing Kit (ON SQK-LSK110) following the manufacturer's

protocol. The sequencing was performed on an Oxford Nanopore GridION sequencer using R9.4.1 flow cells (FLO-MIN106D). The adaptive sampling method was used to target regions of interest, which included CpGs, CpG islands, shores, shelves and promoter regions. Sequence metrics are provided in Supplementary Table 5. Base calling and alignment were performed using Guppy (v6.2.1_gpu), Minimap2 (v2.26) and Remora (v2.0.0).

### DiMeLo-seq sample preparation and analysis

DiMeLo-seq, a method that uses antibody-tethered enzymes to methylate DNA near a target protein's binding sites in situ, was performed using the previously described protocol with some modifications[52]. For detailed, step-by-step protocols, see Supplementary Method 11. The sequence metrics were updated in Supplementary Table 5.

### Mouse xenograft studies

Mice were housed in specific pathogen-free animal facilities (at 20–23 °C, with 40–60% humidity and 12-h light/12-h dark cycle). LuCaP PDX were derived from resected metastatic PCa with the informed consent of patient donors as described previously[76] under a protocol approved by the University of Washington Human Subjects Division institutional review board. Non-obese diabetic (NOD)/SCID male mice at 6–7 weeks old were purchased from Charles River Laboratories. For the LuNE xenograft, $2 \times 10^6$ cells were subcutaneously injected into the right flank of the mice. Tumor volumes were measured twice per week with digital calipers, using the following formula: $V = L \times W^2/2$ ($V$, mm³; $L$, mm; $W$, mm). Once the size of tumors reached approximately 70 mm³, the mice were randomized to receive vehicle (5% DMSO:95% methylcellulose) or CCS1477 (20 mg kg⁻¹) daily by oral gavage for 33 days. For LuCaP145.2 PDX xenograft, fresh tumors were cut into small pieces (8 mm³) using a scalpel and were subcutaneously implanted into both sides of SCID mice. The size of tumors reached approximately 150 mm³ at 1 month after implantation, and then mice were randomized to receive vehicle (5% DMSO:95% methylcellulose) or CCS1477 (20 mg kg⁻¹) every day by oral gavage for 18 days. The tumors and other organs of mice were collected for downstream analyses at the endpoint (tumor size did not exceed the maximal tumor volume of 2,000 mm³, as specified in the approved protocol). The IR of tumor growth was calculated as (tumor volume of vehicle − tumor volume of CCS1477)/tumor volume of vehicle.

### Immunohistochemistry (IHC) and TMA analysis

For detailed step-by-step protocols on IHC, see Supplementary Method 12. For human TMA staining, TMAs containing metastatic CRPC specimens were obtained as part of the University of Washington Medical Center Prostate Cancer Donor Program, which is approved by the University of Washington Institutional Review Board. All specimens for IHC were formalin-fixed (decalcified in formic acid for bone specimens), paraffin-embedded and examined histologically for the presence of a non-necrotic tumor. TMAs were constructed using 1-mm-diameter duplicate cores ($n = 126$) from CRPC tissues ($n = 23$ patients), which consisted of visceral metastases and bone metastases ($n = 63$ sites) from patients who died within 8 h. TMAs staining of FOXA2 and NKX2-1 was conducted by the Northwestern Pathology Facility. The information of the antibodies used for IHC is listed in Supplementary Table 2. Images were captured with TissueFax Plus from TissueGnostics, exported to TissueFAX viewer and analyzed using Photoshop CS4 (Adobe). FOXA2 and NKX2-1 immunostaining was scored blindly by a pathologist using a score of 0–3 for intensities of negative, weak, moderate or intense and multiplied by the percentage of stained cancer cells.

### Statistics and reproducibility

No statistical methods were used to predetermine sample sizes, but our sample sizes are similar to those reported in previous publications[20,42]. For each independent in vitro experiment, at least three technical replicates were used. All in vitro experiments were independently repeated at least two or three times, as specified in Fig. 2b, Fig. 4b,c,g, Fig. 5d,f–h, Fig. 7b–c, Extended Data Fig. 2b–c, Extended Data Fig. 7d,f,g, Extended Data Fig. 9g and Extended Data Fig. 10d,j. No data were excluded from the analyses. The error bars in the figures represent the s.d. of the mean, unless stated otherwise. Two-sided Student's $t$ tests were used to assess statistical significance in RT–qPCR experiments and cell-based functional assays. The statistical analysis methods and software version used for NGS data analysis are provided in the corresponding code and figure legends (Fig. 1c–f,i–j, Fig. 5a–b, Fig. 7e–f,h, Extended Data Fig. 6a,c, Extended Data Fig. 7b and Extended Data Fig. 9j). For in vivo experiments, the number of animals was determined based on the variability in tumor take rate and growth and is provided in the respective figure legends. Animals were randomly assigned to treatment groups once tumors reached the mean volume specified in Fig. 7i,k. No data were excluded from the analyses. Data distributions were assumed to be normal but were not formally tested. Statistical analyses were performed using a two-sided $t$ test.

### Reporting summary

Further information on research design is available in the Nature Portfolio Reporting Summary linked to this article.

### Data availability

All sequencing data (RNA-seq, ATAC–seq, ChIP–seq, Hi-C, reduced representation methylation sequencing, DiMeLo-seq and single-cell sequencing) generated for the study have been deposited in the Gene Expression Omnibus (GEO; GSE239278) at https://www.ncbi.nlm.nih.gov/geo/query/acc.cgi?acc=GSE239278.

The MS proteomic data have been deposited in the ProteomeXchange Consortium via the PRIDE[77] partner repository with the dataset identifiers PXD061127 and PXD061080. Source data are provided with this paper. The published RNA-seq, ATAC–seq and H3K27ac ChIP–seq data from LuCaP PDX, referenced in this study, are available in the GEO database under the accessions GSE126078, GSE156292 and GSE161948, respectively. The WGBS and RNA-seq data from human patients, referenced in this study, are available in dbGaP (phs001648 and phs000909.v.p1) and in the GEO database under the accessions GSE74685, GSE126078, GSE21034, GSE6919 and GSE77930. The scRNA-seq data from human patients, referenced in this study, are from ref. 46. Source data are provided with this paper.

### Code availability

The code for NGS analyses performed in this paper has been uploaded to GitHub (https://github.com/JYULAB/NKX2-1_NEPC_project) and Zenodo (https://doi.org/10.5281/zenodo.15610492)[78].

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

## Acknowledgements

This work was partially supported by the Northwestern University Pathology Core Facility, the Center for Advanced Microscopy/Nikon Imaging Center and Emory University's Integrated Cellular Imaging Core and Integrated Genomics Core. Next-generation sequencing was done at the University of Chicago Genomics Facility, Northwestern Sequencing Core, Admera Health and Florida State University College of Medicine. IP–MS was done at the Taplin Mass Spectrometry Facility of Harvard Medical School. We thank members of the Yu Lab for helpful discussions. We thank E. Corey (University of Washington) for generating and maintaining the LuCaP PDX models, which were partially funded by NIH (awards P50CA97186 and PO1CA163227), and for sharing them with us. The CRPC TMA was graciously provided by C. Morrissey (Department of Urology, University of Washington School of Medicine, Seattle, WA, USA) through the Prostate Cancer Biorepository Network. We thank J. Huang (Department of Pathology, Duke University School of Medicine, Durham, NC, USA) for sharing their scRNA-seq data of clinical PCa samples. We thank A. Zoubeidi (Department of Urologic Sciences, University of British Columbia, Vancouver, BC) for sharing the ENZ-resistant AR$^+$/PSA$^-$ cell line 42D. This work was supported by the National Institutes of Health (NIH)–National Cancer Institute (NCI) (training grant T32CA09560 to G.G.), NIH (grants R35GM142539 to R.G., R01CA257446 to J.Y., R50CA211271 to J.C.Z., U01CA113913 to M.G.S. and R01CA286147 to J.C.Z. and J.Y.), the Prostate Cancer SPORE (grant P50CA180995 to J.Y. and X.Y.), the Department of Defense (grants PC210266 to J.Y., and PC220151 and PC220158 to J.C.Z.) and the Prostate Cancer Foundation (2017CHAL2008 to J.Y. and J.C.Z.).

## Author contributions

J.Y., J.C.Z. and X.L. conceived the project and designed the experiments. J.C.Z., V.K., I.C., J.W., L.L., C.S., Y.H., R.G. and J.Y. conducted bioinformatic and statistical analyses. G.G. and W.X. assisted with mouse experiments. X.L. and W.X. performed H&E and IHC experiments. Q.J. performed LNCaP$^+$FOXA2 D14 and D28 Hi-C experiments. X.L. performed the rest of the experiments. N.A. provided pA-Hia5 and consulted on DiMeLo-seq experiments. M.S., P.J. and V.C. provided valuable comments on the project. X.L. V.K., J.Z. and J.Y. wrote the paper.

## Competing interests

All authors declare no competing interests.

## Additional information

**Extended data** is available for this paper at https://doi.org/10.1038/s41588-025-02265-4.

**Correspondence and requests for materials** should be addressed to Jonathan C. Zhao or Jindan Yu.

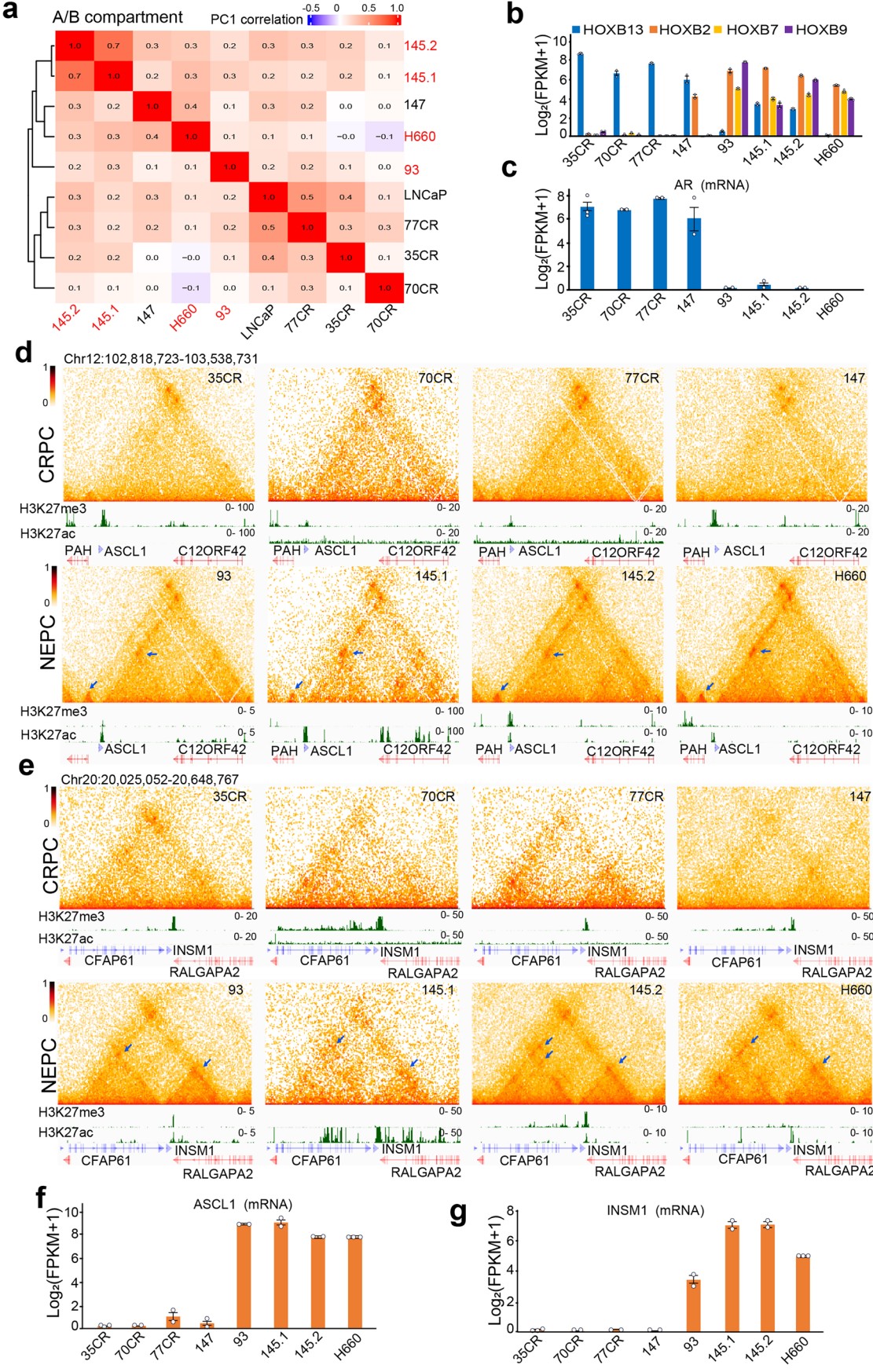

**Extended Data Fig. 1 | See next page for caption.**

**Extended Data Fig. 1 | Distinct 3D chromatin architecture in NEPC compared to CRPC tumors. a**, Unsupervised hierarchical clustering of Hi-C samples using the top 25% most variable PC1 of Hi-C matrices. CRPC PDXs: 35CR, 70CR, 77CR, 147. NEPC PDX: 93, 145.1 and 145.2. NEPC cell line: NCI-H660. Androgen-dependent PCa cell line: LNCaP. **b,c**, RNA-seq data of *HOXB* genes (**b**) and *AR* (**c**) in the indicated models. RNA-seq data for NCI-H660 were generated in-house, and RNA-seq data for the rest models were obtained from GSE126078.

RNA-seq was performed in triplicate for 35CR and NCI-H660, and in duplicate for the other models. **d,e**, Hi-C contact maps (5 kb resolution) of *ASCL1* (**d**) and *INSM1* (**e**) locus in CRPC (top row) and NEPC (bottom row) models. Gene boxes and 1D ChIP–seq tracks of matched models are aligned at the bottom. Blue arrows indicate NEPC-enriched loops. The scale bars indicate ICE-normalized contact frequency. **f,g**, RNA-seq data of *ASCL1* (**f**) and *INSM1* (**g**) in the indicated models. The data source is the same as in **b,c**.

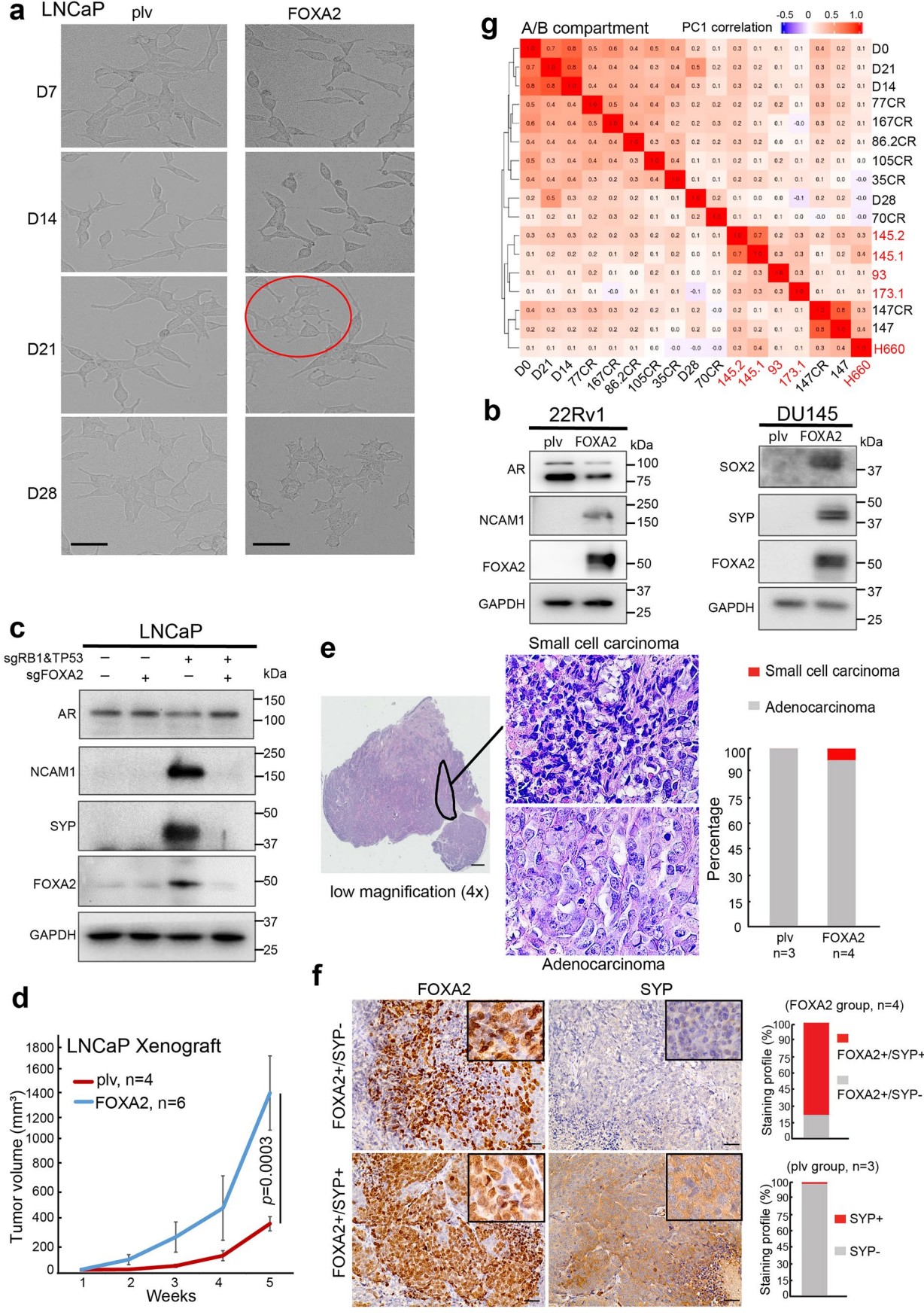

**Extended Data Fig. 2 | See next page for caption.**

**Extended Data Fig. 2 | Remarkable 3D chromatin reorganization during NE transformation of isogenic PCa cells. a**, Representative brightfield images showing the morphology of FOXA2-OE cells at days 7, 14, 21 and 28. LNCaP cells were infected with control (plv) or FOXA2 virus and followed up for 28 days. Cells were imaged every week. Scale bar: 100 μm. **b**, FOXA2 OE induced NET in 22Rv1 (left) and DU145 (right) cells. 22Rv1 or DU145 cells were infected with control (plv) or FOXA2 virus, and 22Rv1 cells were collected for WB analysis at week 4, whereas DU145 cells were collected at day 7. The data shown for 22Rv1 are from one of two independent experiments (n = 2), and those for DU145 are from one of three independent experiments (n = 3). **c**, LNCaP cells were co-infected with sg*RB1*&*TP53* virus along with either sg*NC* or sg*FOXA2* virus for 4 weeks before WB. The data shown are from one of two independent experiments (n = 2). **d**, Tumor growth curve of control (plv, n = 4 mice) and FOXA2-OE (n = 6 mice) LNCaP xenograft tumors. Data are mean ± s.d., and *p* value by two-sided *t* test.

**e**, Hematoxylin and Eosin (H&E) of control (plv) and FOXA2-OE LNCaP xenograft tumors. Representative H&E images (×4) of FOXA2-OE tumors (n = 4) are shown on the left. The circled region indicates the tumor region with small-cell carcinoma-like features, and the high magnification images of small-cell carcinoma and adenocarcinoma tumors (×40) are shown in the middle. Scale bar, 30 μm. The percentage of small-cell carcinoma and adenocarcinoma tumors in the control and FOXA2-OE groups is quantified on the right. **f**, IHC of FOXA2 and SYP proteins in control (plv) and FOXA2-OE LNCaP xenograft tumors. Representative IHC images of FOXA2+/SYP+ and FOXA2+/SYP− tumors in the FOXA2-OE group (n = 4) are shown on the left, and the quantification of them is shown on the right. Scale bar, 30 μm. Insets are shown at higher magnification. **g**, Unsupervised hierarchical clustering of time-course LNCaP+FOXA2 samples, CRPC and NEPC (in red font) based on the top 25% most variable PC1 of Hi-C matrices.

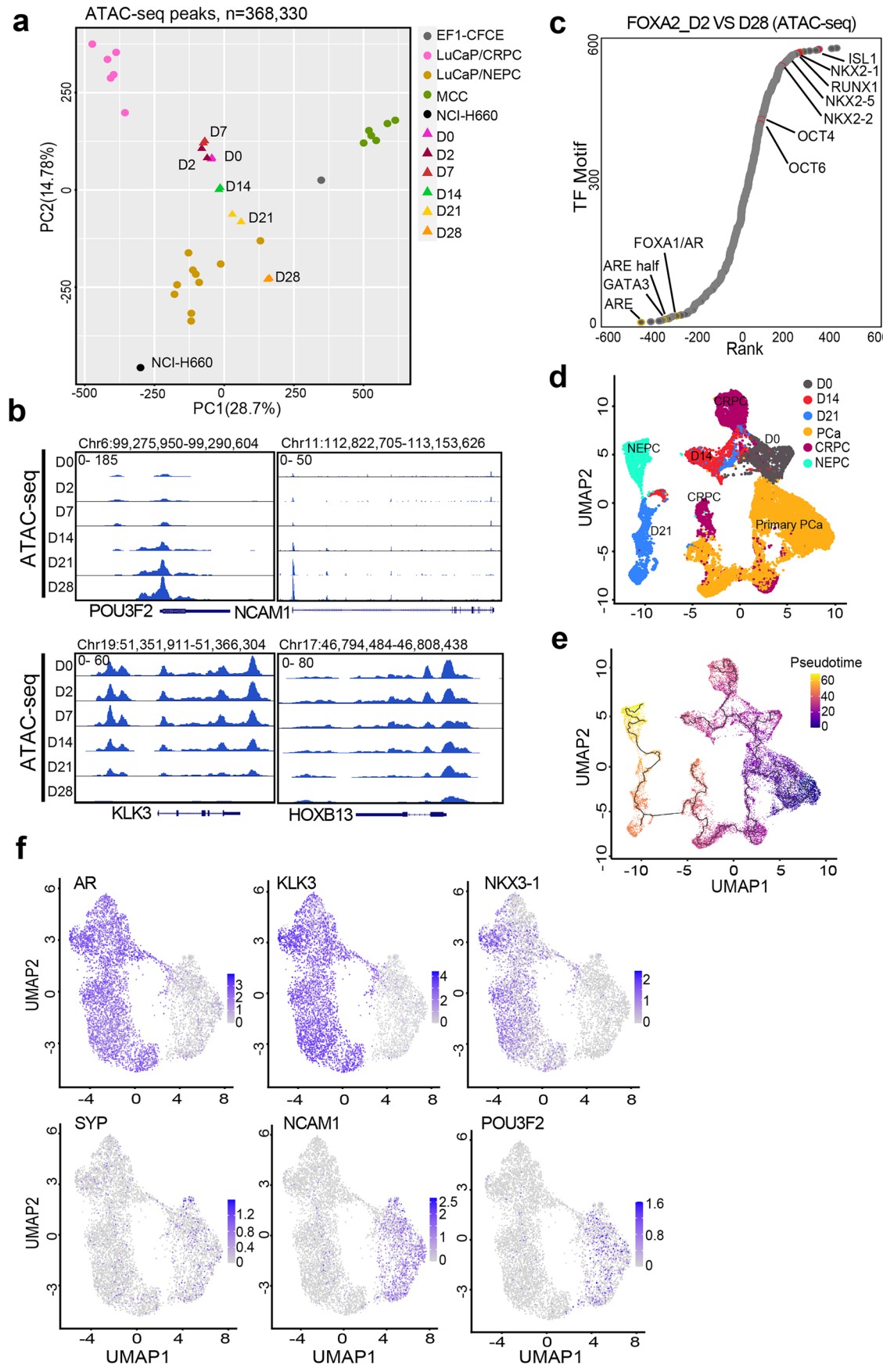

**Extended Data Fig. 3 | See next page for caption.**

**Extended Data Fig. 3 | Single-cell multiome analyses identified transitioning individual cells with intermediate transcriptome and chromatin states.**
**a**, PCA analyses of chromatin accessibility of LNCaP⁺FOXA2 cells, along with LuCaP PDX, Merkel cell carcinoma (MCC) and NEPC cell lines (NCI-H660 and EF1-CFCE) using differentially accessible peaks (n = 368,330) identified by DESeq2 (adjusted $p$ < 0.05). Adjusted $p$ values were calculated by a likelihood ratio test with Benjamini–Hochberg correction. **b**, Genome browser views of ATAC–seq signal at NE lineage genes (*POU3F2* and *NCAM1*, top row) and luminal lineage genes (*KLK3* and *HOXB13*, bottom row) in time-course LNCaP+FOXA2 cells. **c**, TF motifs were plotted by ranks generated from their associated differential $p$ values at ATAC–seq peaks in D2 and D28 LNCaP⁺FOXA2 samples. $P$ values by two-sided hypergeometric test with Benjamini–Hochberg correction. **d**, UMAP integration of D0, D14 and D21 LNCaP⁺FOXA2 cells with previously published clinical PCa cells. **e**, Pseudotime trajectories in PCa cells shown in **d**. Color reflects pseudotime distance (primary PCa located at time = 0). **f**, UMAP visualization of AR and AR target genes (*KLK3*, *NKX3-1*) and NE lineage genes (*SYP*, *NCAM1*, *POU3F2*) in D0, D14 and D21 LNCaP⁺FOXA2 scRNA-seq samples. Color bars: normalized expression.

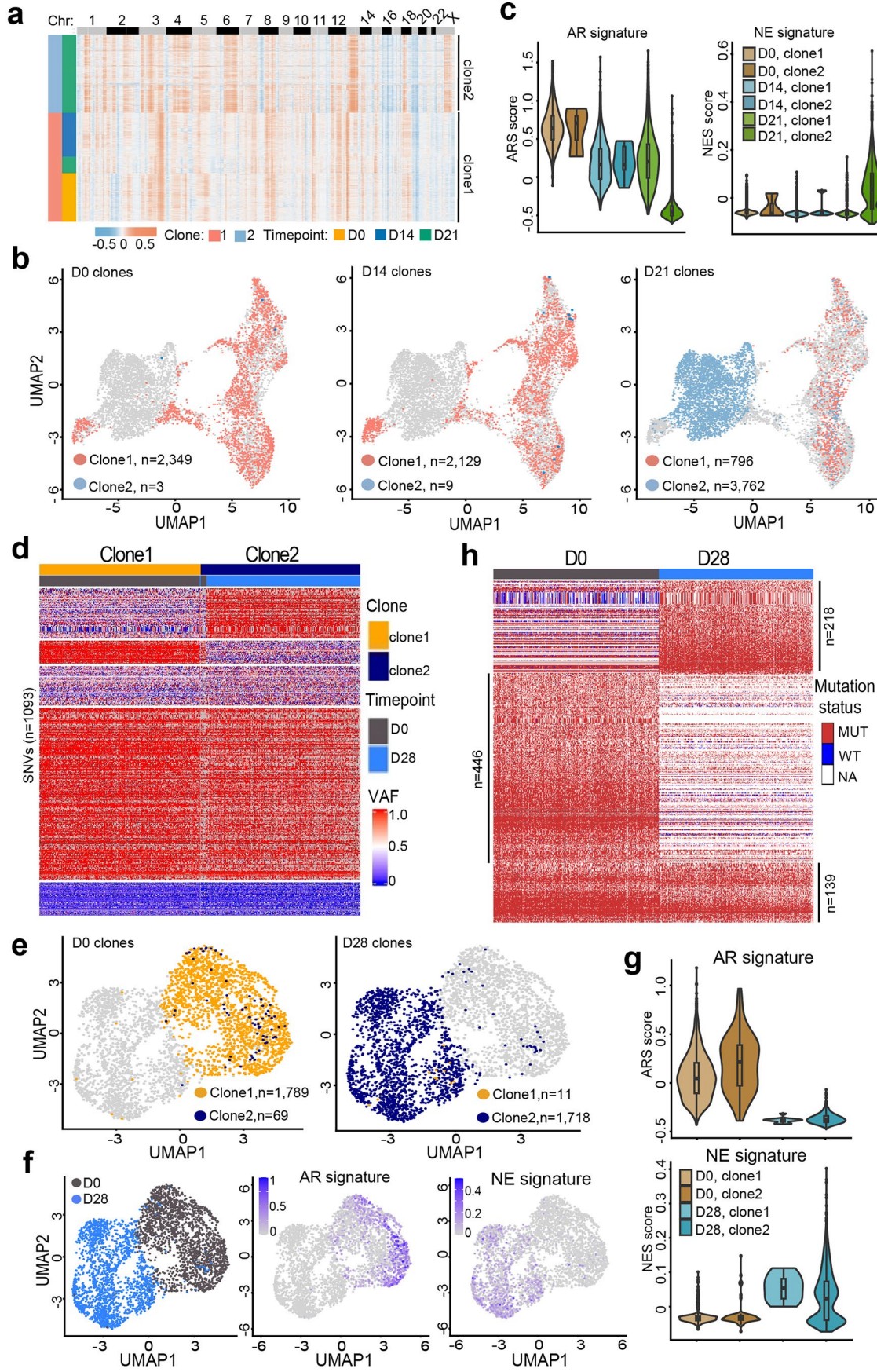

**Extended Data Fig. 4 | See next page for caption.**

**Extended Data Fig. 4 | Clonality analyses of single LNCaP⁺FOXA2 cells support a mixed model of clonal transformation and expansion during NET.** **a**, Heatmap showing two clones delineated by single-cell CNV profiles inferred from scRNA-seq data by CopyKAT in FOXA2-driven NET model. Columns are gene windows ordered by genomic positions, while rows are individual cells that have been clustered using hierarchical clustering. Red color in the heatmap indicates amplification, and blue color indicates deletion. Color bar indicates log copy number ratios of each segment. **b**, scRNA-seq UMAP visualization of D0, D14 and D21 LNCaP⁺FOXA2 cells indicating clone 1 and clone 2 D0, D14 and D21 LNCaP⁺FOXA2 cells. **c**, Violin plots showing AR and NE signature scores of clone 1 and clone 2 in D0, D14 and D21 LNCaP⁺FOXA2 scRNA-seq samples. **d**, Heatmap showing two clones separated by SNVs (n = 1,093) inferred from scNanoRNA-seq data of D0 and D28 LNCaP⁺FOXA2 cells, clustered by *k*-means. Each row is one SNV, and each column is a cell. Variant allele frequency (VAF) is calculated as alternate allele read counts divided by all read counts for that nucleotide such as red represents mutant and blue for WT. **e**, UMAP visualization of scNanoRNA-seq data of D0 and D28 LNCaP⁺FOXA2 cells and their clonality. **f**, UMAP visualization of scNanoRNA-seq data of D0 and D28 LNCaP⁺FOXA2 cells and their expression of AR and NE signature genes. **g**, Violin plots showing AR and NE signature scores of clone 1 and clone 2 in D0 and D28 LNCaP⁺FOXA2 cells based on scNanoRNA-seq data. **h**, Heatmap showing germline mutations in D0 and/or D28 LNCaP⁺FOXA2 cells based on scNanoRNA-seq data. Each row represents one germline mutation, and each column is a cell. WT represents SNVs with supporting reads for the wildtype allele only, MUT are those that have at least 1 supporting read for the alternative allele and NA are those with 0 read coverage. Germline mutations were separately identified for each time point, with their mutation status plotted across the other time point without pre-filtering, wherein they might have 0 (showing as NA) or insufficient read coverage to capture the mutant allele (appearing as WT). Violin plots in **c** and **g** show median values (black line), interquartile range (IQR, black box) and the range of all data points within 1.5× IQR (whiskers).

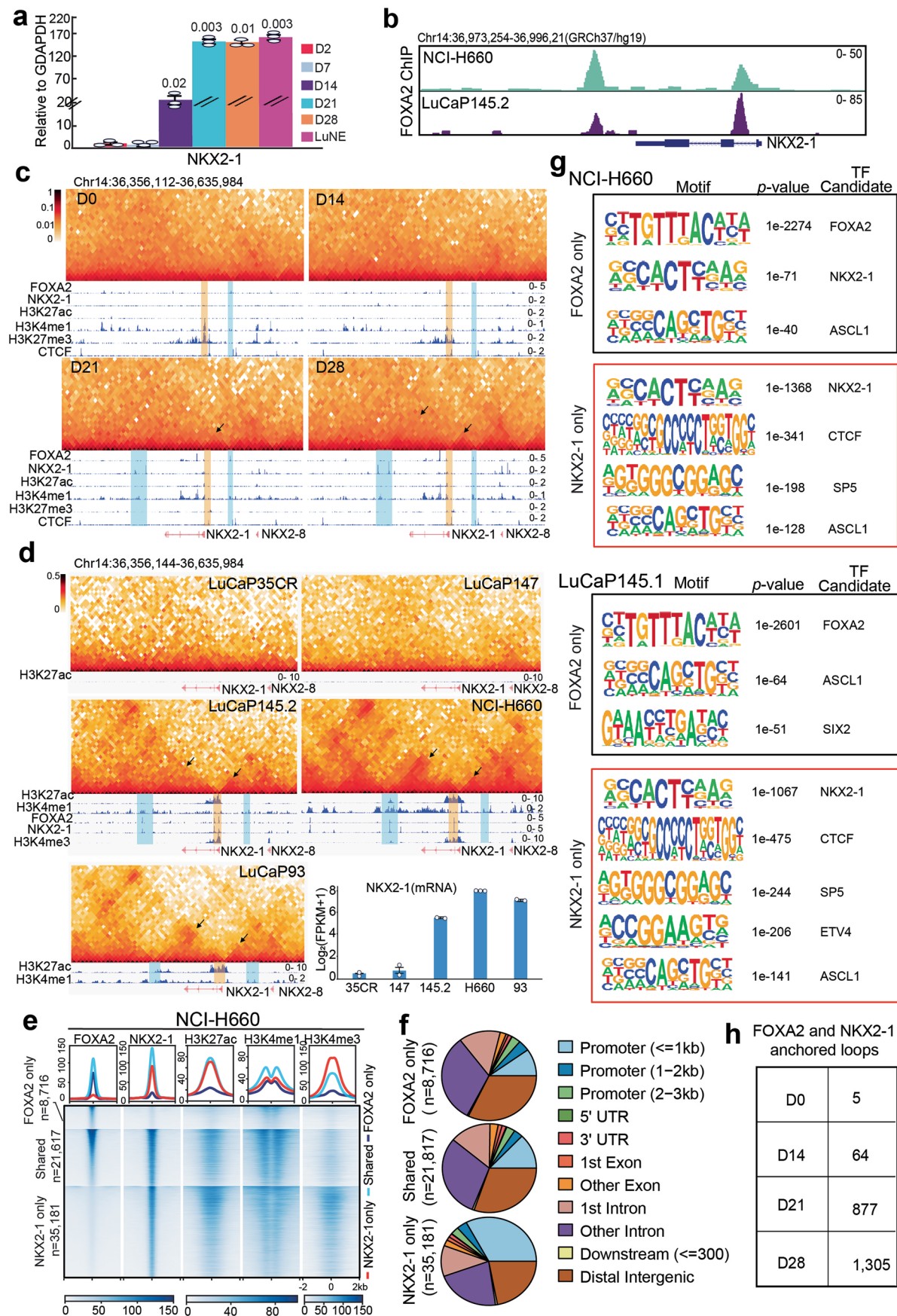

**Extended Data Fig. 5 | See next page for caption.**

**Extended Data Fig. 5 | NKX2-1 is induced by FOXA2 and required for FOXA2-driven NET. a**, RT–PCR analysis of *NKX2-1* mRNA in time-course LNCaP⁺FOXA2 cells and LuNE cells. Data were normalized to GAPDH. Shown are the mean ± s.e.m. of technical replicates from one of three (n = 3) independent experiments. *P* values by two-sided *t* test. **b**, Genome browser view of FOXA2 ChIP–seq signal around the *NKX2-1* gene in NEPC cells (NCI-H660, LuCaP145.2). **c,d**, Hi-C contact maps (5 kb resolution) of *NKX2-1* gene region in time-course LNCaP⁺FOXA2 cells (**c**), NCI-H660 and LuCaP PDX (**d**). The mRNA expression of *NKX2-1* in the indicated samples is shown on the bottom right. Gene boxes and 1D ChIP–seq tracks of matched models are aligned at the bottom. Black arrows indicate enhancer–promoter loops at *NKX2-1* locus. Enhancers are highlighted in light blue, and promoters in light orange color. FOXA2, NKX2-1, H3K27ac,

H3K4me1, H3K27me3 and CTCF tracks under the D0 contact map used their corresponding ChIP–seq performed in closely related D2 cells. All others were done in conditions exactly matched to Hi-C. The scale bars indicate ICE-normalized contact frequency. **e**, Heatmap showing active (H3K27ac) enhancer (H3K4me1) and promoter (H3K4me3) marks at NKX2-1-only, FOXA2-only and shared binding sites in NCI-H660 cells. Scale bar: enrichment intensity. **f**, Pie chart showing the genomic distribution of FOXA2-only, NKX2-1-only and shared binding sites identified in NCI-H660 cells. **g**, TF motifs that significantly enriched at NKX2-1-only, FOXA2-only binding sites in NCI-H660 (top) and LuCaP145.1 (bottom). *P* values by two-sided hypergeometric test with Benjamini–Hochberg correction. **h**, The number of chromatin loops anchored by FOXA2 and NKX2-1 in time-course LNCaP⁺FOXA2 Hi-C samples.

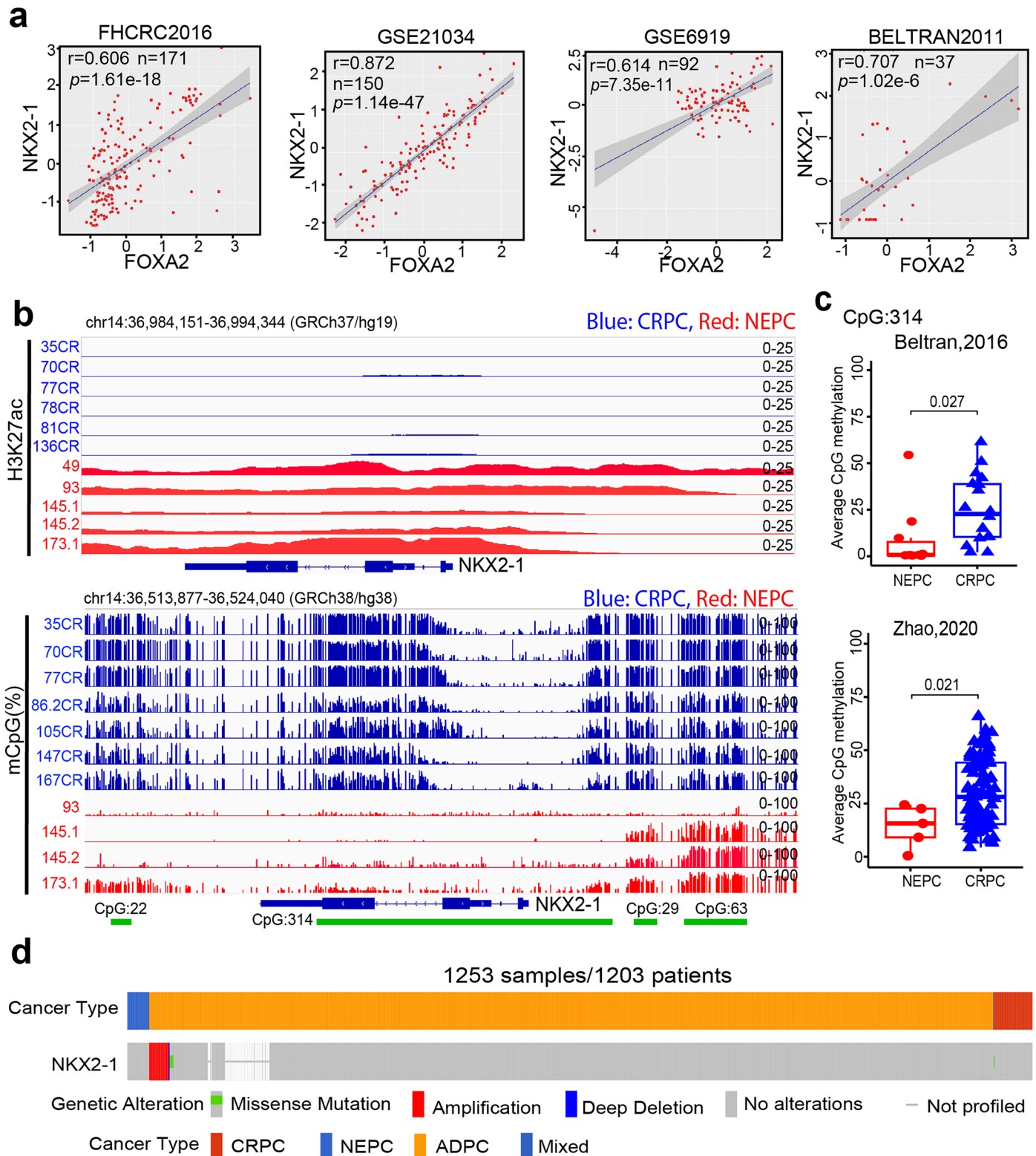

**Extended Data Fig. 6 | NKX2-1 is highly expressed in NEPC tumors and critical for NET of PCa. a,** Positive correlation between *NKX2-1* and *FOXA2* mRNA levels in PCa datasets. Shown on x and y axis are z scores of $\log_2$(MAS5.0 signal intensity + 1). The linear regression line (blue) with its 95% confidence interval (gray) is shown. *P* value by two-sided *t* test without multiple comparisons. **b,** IGV views of H3K27ac (top) and DNA methylation (bottom, 5mC) at *NKX2-1* locus in LuCaP PDX. The green boxes indicate CpG islands. **c,** Boxplots showing average CpG methylation level at CpG island (CpG:314) of *NKX2-1* in CRPC and NEPC patient samples in ref. 3 (top, CRPC, n = 18; NEPC, n = 10) and in Zhao (2020)

(bottom, CRPC, n = 93; NEPC, n = 5) dataset. Shown on the y axis are average percentage of methylated CpG at CpG island: 314. Boxplots represent the median and interquartile range (IQR); whiskers indicate the smallest and largest non-outlier values. *P* value by two-sided *t* test without multiple comparisons. **d,** Genomic alteration of *NKX2-1* in the published PCa dataset. The data presented here consists of combined samples from following five studies: MSK, Clin Cancer Res 2020; SU2C/PCF Dream Team, PNAS 2019; Multi-Institute, Nat Med 2016; SU2C/PCF Dream Team, Cell 2015; MCTP, Nature 2012.

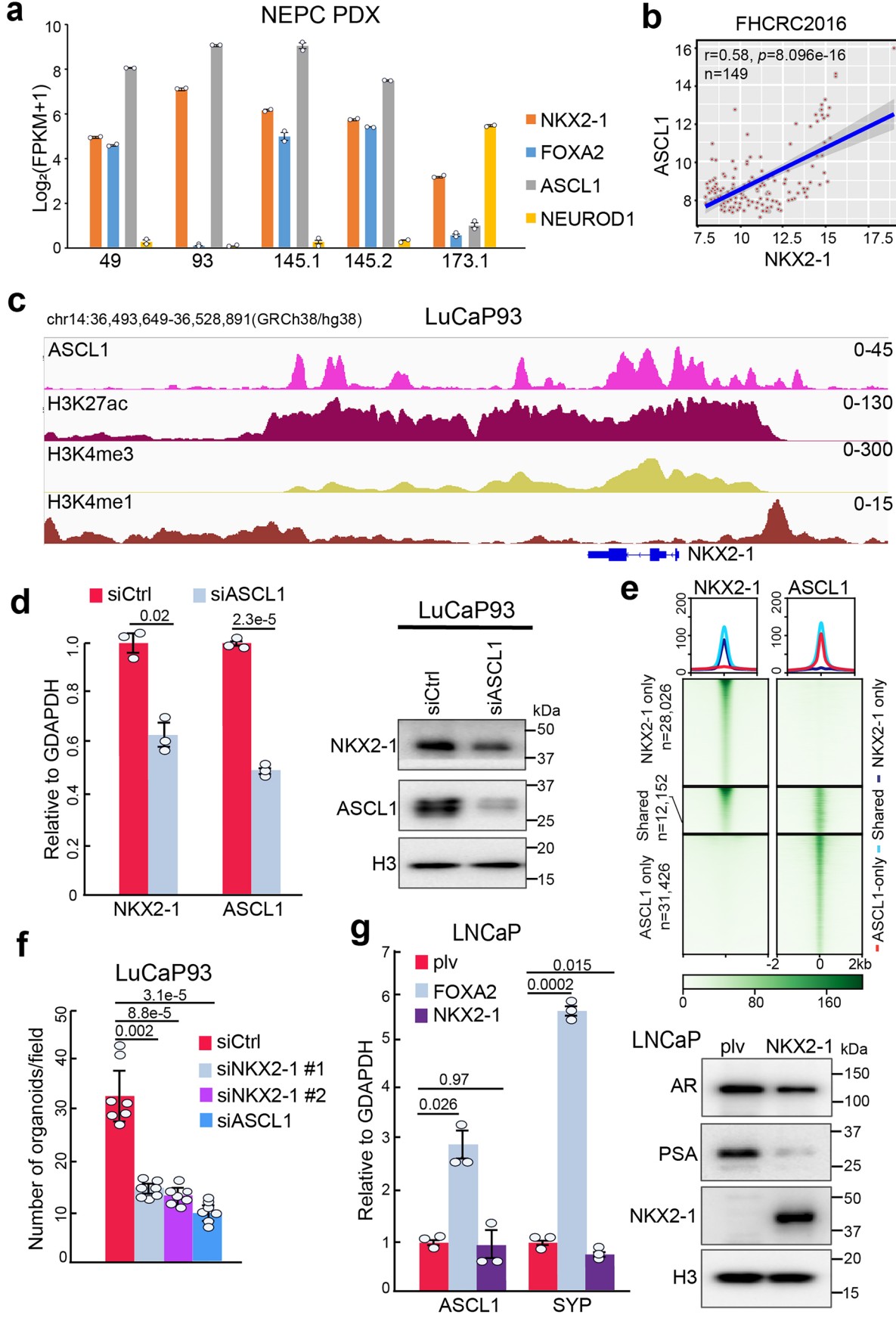

**Extended Data Fig. 7 | See next page for caption.**

**Extended Data Fig. 7 | NKX2-1 is regulated by ASCL1 in FOXA2-low NEPC.**
**a**, The RNA-seq data of *FOXA2*, *NKX2-1*, *ASCL1* and *NEUROD1* in LuCaP NEPC PDX.
*NKX2-1* is expressed in both FOXA2-high (49, 145.1 and 145.2) and FOXA2-low
(93 and 173.1) PDX. RNA-seq data were obtained from GSE126078. **b**, Positive
correlation between *NKX2-1* and *ASCL1* mRNA levels in PCa dataset. Shown on x
and y axes are z scores of $log_2$(MAS5.0 signal intensity + 1). The linear regression
line (blue) with its 95% confidence interval (gray) is shown. *P* value by two-sided
*t* test without multiple comparisons. **c**, IGV showing ASCL1, active enhancer
marker (H3K4me1 and H3K27ac), promoter marker (H3K4me3) at *NKX2-1* locus
in LuCaP93 PDX. ASCL1 strongly binds to promoter and enhancer regions of
NKX2-1. **d**, *ASCL1* knockdown decreased expression of NKX2-1 at both mRNA

(left) and protein (right) levels in LuCaP93 organoids. RT–PCR data shown are
the mean ± s.e.m. of technical replicates from one of three (n = 3) independent
experiments. *P* values by unpaired two-sided *t* test. WB data shown are from one
of two (n = 2) independent experiments. **e**, Heatmap showing NKX2-1, ASCL1
binding at NKX2-1-only, shared and ASCL1-only sites in LuCaP93 PDX. Scale bar:
enrichment intensity. **f**, *NKX2-1* knockdown reduced LuCaP93 organoids growth.
The quantification of the average number of organoids per field (n = 7 fields) is
shown. *P* values by two-sided *t* test. **g**, NKX2-1 OE alone was unable to induce NET
in LNCaP cells. LNCaP cells were infected with FOXA2 or NKX2-1 virus, and cells
were collected for RT–PCR (left) and WB (right) analyses at week 4. The data are
shown as in **d**.

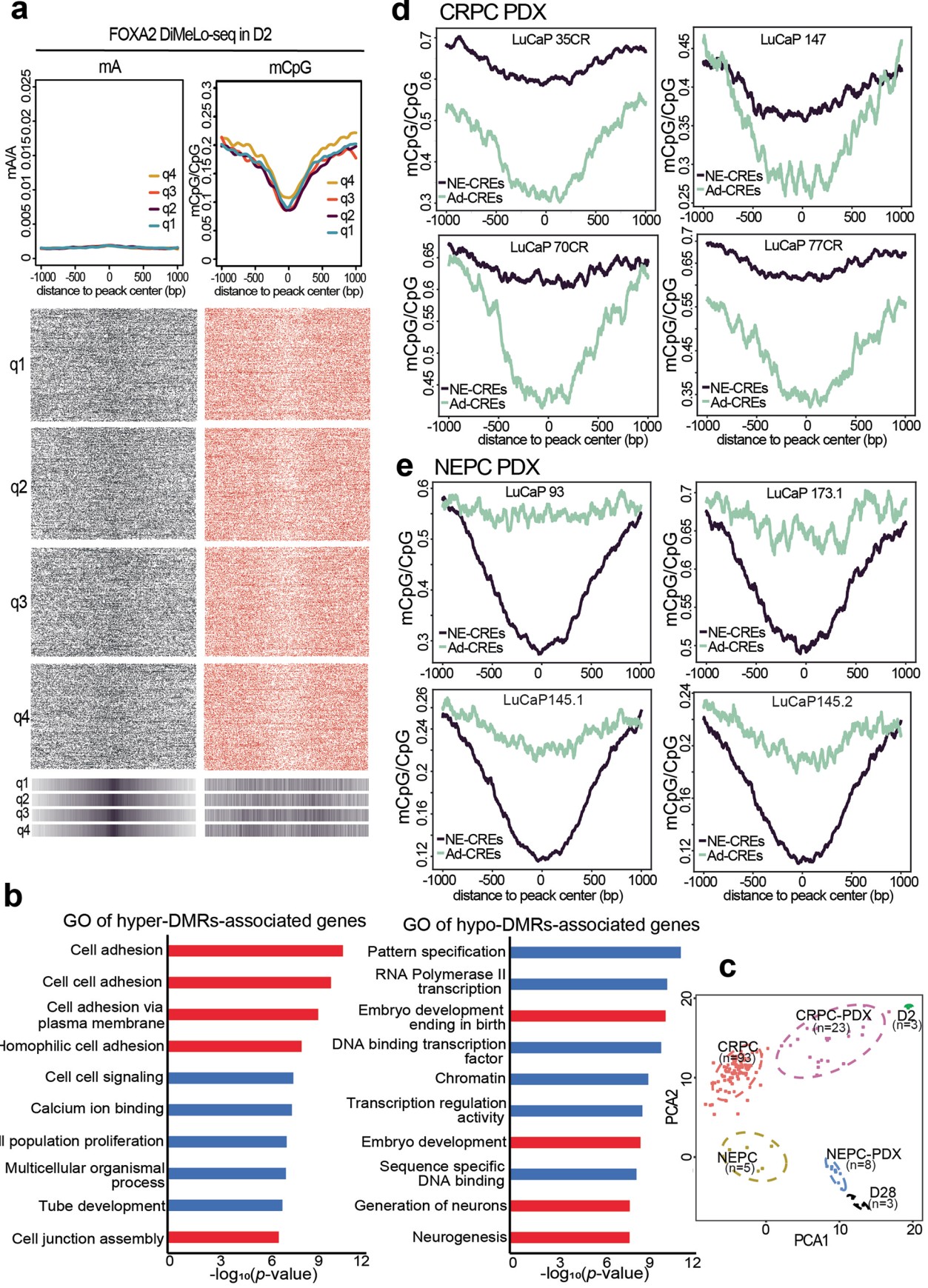

**Extended Data Fig. 8 | See next page for caption.**

**Extended Data Fig. 8 | FOXA2 induces regional DNA demethylation during NET of PCa. a**, The mA and mCpG profiling on single DNA molecules spanning FOXA2 peak center at 4 quartiles of FOXA2 peaks as in Fig. 6b in D2 LNCaP⁺FOXA2 cells. Aggregate curves (top) for each quartile were created with a 50-bp rolling window. Base density across the 2-kb region for each quartile is indicated in the one-dimensional heatmaps (bottom); the scale bars indicate the number of adenine bases and CG dinucleotides sequenced at each position relative to FOXA2 peaks center as in Fig. 6b. Color scales are shown as in Fig. 6a. **b**, GO analyses by MSigDB of the top 200 genes associated with hyper-DMRs (left) and hypo-DMRs (right) between D2 and D28 LNCaP⁺FOXA2 cells. Top enriched molecular concepts are shown on the y axis, while the x axis indicates enrichment significance in −log$_{10}$(p value); *p* values were calculated by the one-sided hypergeometric test. **c**, PCA analyses of CRPC/NEPC PDX and patient samples using the top 200 genes as in **b**. **d**,**e**, Aggregated mCpG intensity plots showing the DNA methylation profiles of CRPC PDX (**d**) and NEPC PDX (**e**) at previously reported NE-CREs and Ad-CREs[11].

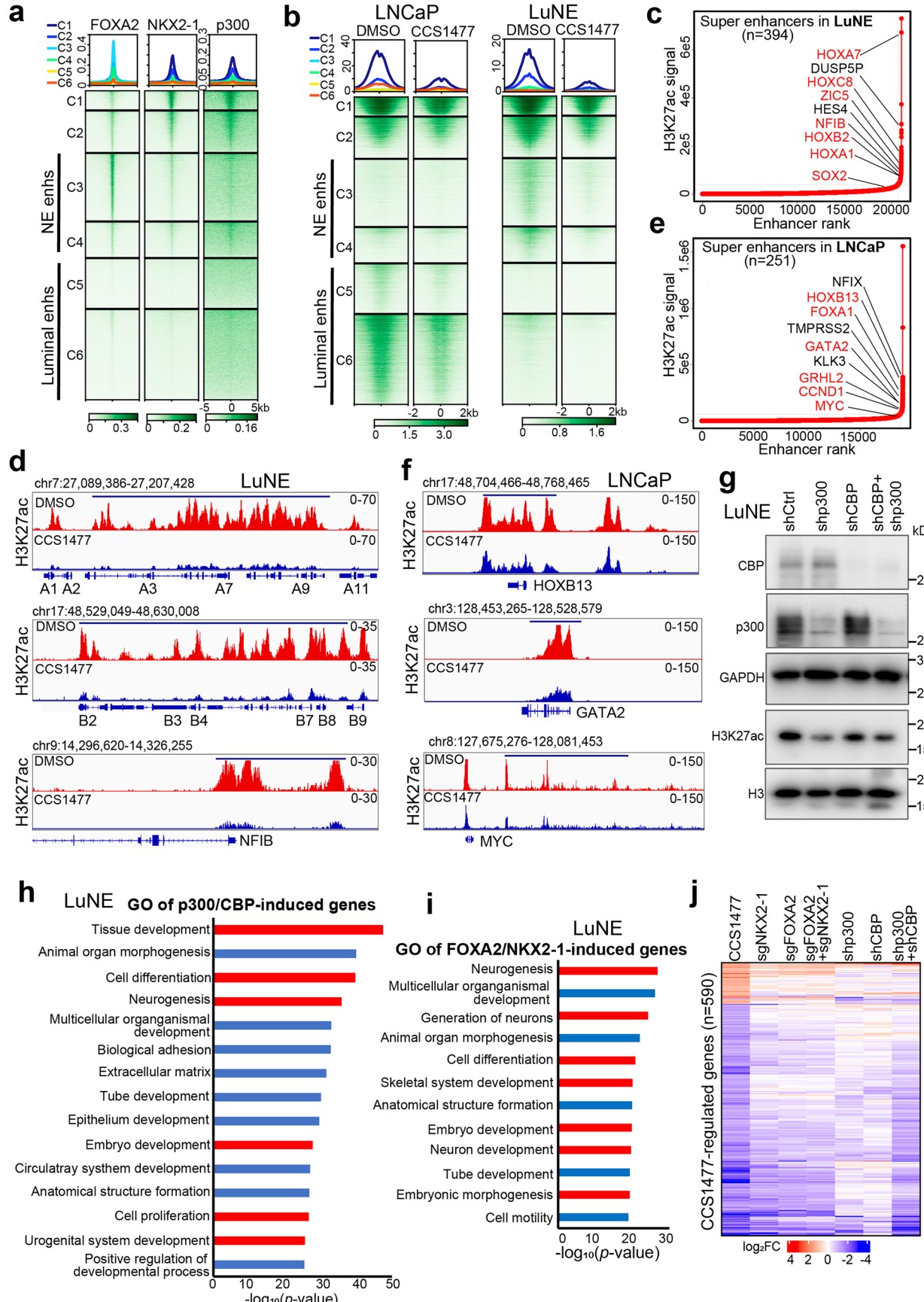

**Extended Data Fig. 9 | See next page for caption.**

**Extended Data Fig. 9 | CCS1477 targets lineage-specific enhancers active in a particular cell. a**, Heatmap showing FOXA2, NKX2-1 and p300 co-occupation at NE (C3–C4) and shared enhancers (C1–C2) in LuNE cells. The luminal/NE enhancers (enhs) are identified in Fig. 3a. Color bar at the bottom indicates the scale of ChIP–seq enrichment intensity. **b**, Heatmap showing H3K27ac at luminal and NE enhancers were, respectively, inhibited by CCS1477 in LNCaP and LuNE cells. The luminal/NE enhancers (enhs) are identified as in **a**. Color bar is shown as in **a**. **c**, Super-enhancers (SEs) were identified using H3K27ac ChIP–seq in LuNE cells. Hockey-stick plot showing the normalized rank and signals of H3K27ac. Representative SE-associated lineage-specific TFs and oncogenic genes are highlighted in red. **d**, IGV showing H3K27ac enrichment at SE-associated *HOXA* gene cluster, *HOXB* gene cluster and *NFIB* gene locus in LuNE cells treated with DMSO or CCS1477. The blue horizontal lines on the top of the tracks mark SEs. **e**, SEs were identified in LNCaP cells and depicted as in **c**. Representative SE-associated lineage-specific TFs and oncogenic genes are highlighted in red.

**f**, IGV showing H3K27ac status at SE-associated gene loci in LNCaP cells treated with DMSO or CCS1477. The blue horizontal lines on the top of the tracks mark SEs. **g**, WB confirming *p300* and *CBP* KD in LuNE cells subjected to sh*CBP* and/or shp*300*. The data shown are from one of two (n = 2) independent experiments. **h**, GO analysis of p300/CBP-induced genes in LuNE cells. GO analysis was performed by MSigDB. Top enriched molecular concepts are shown on the y axis, while the x axis indicates enrichment significance. $-\log_{10}$(p value) is shown, p values by one-sided hypergeometric test. **i**, GO analysis of FOXA2/NKX2-1-induced genes in LuNE cells, performed as in **g**. **j**, Heatmap showing that most of CCS1477-regulated genes (FC ≥ 2, adjusted p < 0.05) were similarly regulated by p300/CBP, NKX2-1 and FOXA2. Data shown are the $\log_2$(FC) of CCS1477 relative to DMSO, and of sg*NKX2-1*, sg*FOXA2* or sg*FOXA2*⁺sg*NKX2-1* relative to sgNC, and of sh*p300*, sh*CBP* or sh*p300*⁺sh*CBP* to shCtrl. Adjusted p values by Wald test with Benjamini–Hochberg correction.

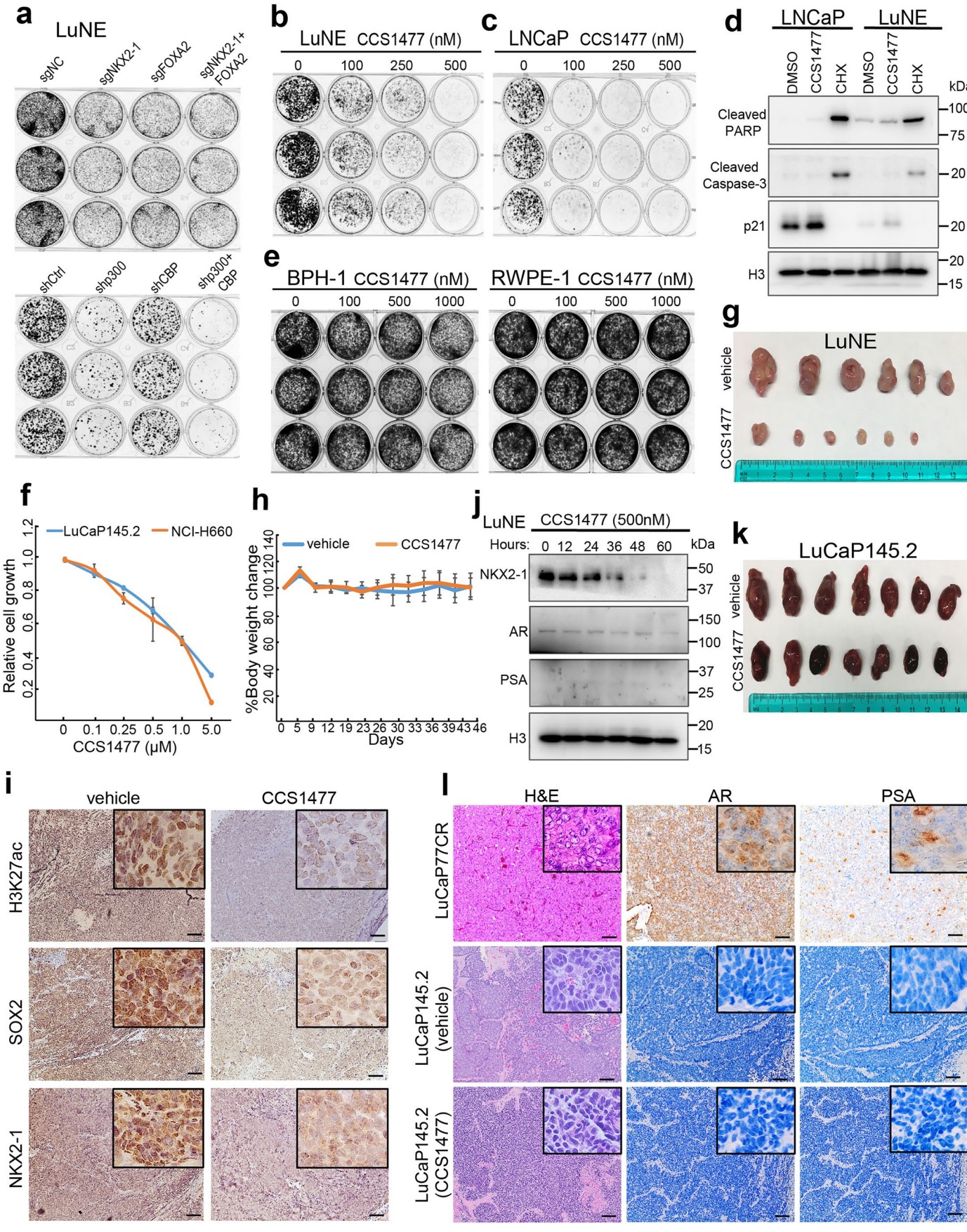

**Extended Data Fig. 10 | See next page for caption.**

**Extended Data Fig. 10 | p300/CBP inhibition suppresses NEPC cell growth in vitro and in vivo. a**, Colony formation assay of LuNE cells with the indicated gene KD. **b,c**, Colony formation assay of LuNE (**b**), LNCaP (**c**) cells treated with the indicated concentration of CCS1477. **d**, WB analysis of markers for cell cycle arrest (p21) and apoptosis (cleaved PARP and cleaved caspase 3) in LNCaP and LuNE cells treated with 500 nM CCS1477 for 24 h. Cycloheximide (CHX, 100 µg ml⁻¹, 24 h)-treated cells were used as a control for cytotoxic effects. The data shown are from one of two (n = 2) independent experiments. **e**, Colony formation assay of benign prostatic hyperplasia BPH-1 and normal prostate epithelial RWPE-1 treated with the indicated concentration of CCS1477. **f**, Growth of LuCaP145.2 organoids and NCI-H660 cells treated with increasing doses of CCS1477. **g**, Images of LuNE xenograft tumors treated with vehicle or CCS1477.

**h**, Body weight of mice treated with vehicle or CCS1477 for 33 days. Data in each time point are mean ± s.d. Y axis shows the percentage of body weight change. **i**, Representative IHC images of H3K27ac, SOX2 and NKX2-1 in LuNE xenograft tumors treated with vehicle or CCS1477 (n = 5 tumors per group). Scale bar, 30 µm. Insets are shown at higher magnification (×40). **j**, WB analysis of LuNE cells treated with 500 nM CCS1477 up to 60 h. The data shown are from one of two (n = 2) independent experiments. **k**, Images of LuCaP145.2 xenograft tumors treated with vehicle or CCS1477. **l**, H&E (left) and IHC of AR and PSA proteins (right) in LuCaP77CR (as a CRPC control) and in LuCaP145.2 PDX tumors treated with vehicle or CCS1477. Representative H&E and IHC images (n = 5 tumors per group) are shown. Scale bar, 30 µm. Insets are shown at higher magnification (×40).

# Reporting Summary

## Statistics

For all statistical analyses, confirm that the following items are present in the figure legend, table legend, main text, or Methods section.

| n/a | Confirmed | |
|---|---|---|
| ☐ | ☒ | The exact sample size ($n$) for each experimental group/condition, given as a discrete number and unit of measurement |
| ☐ | ☒ | A statement on whether measurements were taken from distinct samples or whether the same sample was measured repeatedly |
| ☐ | ☒ | The statistical test(s) used AND whether they are one- or two-sided *Only common tests should be described solely by name; describe more complex techniques in the Methods section.* |
| ☒ | ☐ | A description of all covariates tested |
| ☐ | ☒ | A description of any assumptions or corrections, such as tests of normality and adjustment for multiple comparisons |
| ☐ | ☒ | A full description of the statistical parameters including central tendency (e.g. means) or other basic estimates (e.g. regression coefficient) AND variation (e.g. standard deviation) or associated estimates of uncertainty (e.g. confidence intervals) |
| ☐ | ☒ | For null hypothesis testing, the test statistic (e.g. $F$, $t$, $r$) with confidence intervals, effect sizes, degrees of freedom and $P$ value noted *Give P values as exact values whenever suitable.* |
| ☒ | ☐ | For Bayesian analysis, information on the choice of priors and Markov chain Monte Carlo settings |
| ☐ | ☒ | For hierarchical and complex designs, identification of the appropriate level for tests and full reporting of outcomes |
| ☐ | ☒ | Estimates of effect sizes (e.g. Cohen's $d$, Pearson's $r$), indicating how they were calculated |

*Our web collection on statistics for biologists contains articles on many of the points above.*

## Software and code

Policy information about availability of computer code

| Data collection | Western Blot and Quantification: Bio-Rad ChemiDocTM with Image lab6.0.1<br>QPCR:Applied Biosystems with StepOneTM software v2.3 and QuantStudio™ Design & Analysis Software (DA2.7.0)<br>NGS:HiSEQ4000 platform with Illumina RTA 2.7.7; NOVASEQ6000,S4 Reagent Kit v1.5; NOVASEQ X PLUS 10B Reagent Kit (Cat# 20085594).<br>Oxford Nanopore Long Read Sequencing: GridION X5; Ligation Sequencing Kit: SQK-LSK110; Flow cell (R9.4.1)<br>IHC and H&E image acquisition: TissueGnostics with HistoFaxs software package/Keyence BZ-X810<br>Mass spectrometry: Orbitrap Velos Pro™ system |
|---|---|
| Data analysis | The SEQUEST was used for protein identification and peptide sequencing in FOXA2 and IgG mass spectrometry<br>For ChIP-seq analysis:<br>ChIP-seq reads were aligned to the Human Reference Genome (assembly hg19) using Bowtie2/2.0.5.<br>FastQC/0.11.5 was used to check data quality.<br>ChIP-seq peak identification, overlapping, subtraction and feature annotation of enriched regions were performed using HOMER/4.8.3 (Hypergeometric Optimization of Motif EnRichment) suite. Weighted Venn diagrams were created by R package Vennerable/3.0. Heatmap views of ChIP-seq were generated by deepTools/2.0.<br>Genomic distribution of ChIP-seq binding sites was generated by the R Bioconductor package ChIPseeker/1.36.<br>For Bulk RNA-seq analysis:<br>RNA-seq reads were mapped to NCBI human genome GRCh38 using STAR/2.6.0. Raw counts of genes were calculated by STAR. Differential gene expression was analyzed by DESeq2/1.26.0 using LTR test.<br>Heatmaps were generated by ComplexHeatmap/2.2.0.<br>Gene Set Enrichment Analysis was done by GSEA/4.1.0.<br>The scatterplot was generated using GSVA/1.52.3. |

For ATAC-seq analysis:
The sequenced reads were trimmed, filtered, and aligned against hg38 using Bowtie2/2.0.5.
PCR duplicates and reads mapped to the mitochondrial chromosome or repeated regions were removed by with ENCODE ATAC-Seq pipeline 1.8.0/
Peak calling was performed using MACS2, with a p-value < 0.01 as the cutoff. Reproducible peaks from two biological replicates were defined as peaks with Irreproducibility Discovery Rates (IDR) < 0.05.
Differential ATAC-seq peaks were identified by DESeq2 using the LRT test.
For single-cell Multiome analysis:
Raw sequencing data were preprocessed and aligned to hg38 using Cell Ranger ARC (2.0.2).
Low-quality cells with low UMI counts and high mitochondrial ratios were filtered out in Seurat/4.3.0
Each time point was analyzed individually at the expression and accessibility modality, then combined across time points within each modality using Seurat (4.3.0) and Signac (1.6.0) in R (4.0.3).
The RNA modality was normalized using SCTransform/0.4.1. ATAC peaks were called using MACS2/2.1.0 and quantified.
Batch correction was done using the Seurat V3 intergration using reciprocal PCA.
Monocle3 (0.2.3.0) was used to learn the trajectory graph and calculate the pseudotime. Velocyto (0.17.17) was used to generate spliced and unspliced matrices. RNA velocities were projected into the UMAP produced from earlier steps in Seurat, using Pyplot (matplotlib 3.7.0).
The expression and chromatin accessibility of AR and NE signatures were calculated using 'AddModuleScore' in Seurat.
Correction for dropouts was done using RunALRA() in SeuratWrappers (0.3.0)
For clonality analysis:
scRNA-Seq CNV was calculated using copyKat (1.1.0).
scNanoRNA-seq SNV analysis used  single cell Nanopore sequencing analysis of Genotypes and Phenotypes Simultaneously (scNanoGPS) (v1.1)
For Hi-C analysis:
Hi-C data were processed into .mcool maps using the runHiC (0.8.6) pipeline, with chromap as the aligner.
A/B compartments were called using cooltools (0.5.4) eigs_cis function with GC content phasing at 100kb resolution. Loops were called using mustache (1.2.0) at 10kb resolution
APA plots were made with coolpup.py (1.0.0).
For Nanopore long read sequencing data analysis:
Base calling and alignment were processed by guppy/6.2.1_gpu + minimap2/2.26 + remora/2.0.0.
QC was performed by mosdepth/0.3.4 and NanoStat/1.6.0.
Aggregate modified base counts for 5mC and 6mA were performed by modbam2bed/0.9.5.
Bigwig file of IGV track was generated by bedGraphToBigWig from kentUtils/302.1.
A and CpG methylation profiles, intensity plots, heatmaps and single DNA molecule tracks were generated by dimelo/0.1.0 python package.
Differentially methylated regions (DMRs) were identified by R Bioconductor package DSS/2.48.0.
The code for all NGS and Nanopore long read sequencing data analyses performed in this paper are in the process of being uploaded to Github at https://github.com/JYULAB/NKX2-1_NEPC_project.

For manuscripts utilizing custom algorithms or software that are central to the research but not yet described in published literature, software must be made available to editors and reviewers. We strongly encourage code deposition in a community repository (e.g. GitHub). See the Nature Portfolio guidelines for submitting code & software for further information.

# Data

Policy information about availability of data

All manuscripts must include a data availability statement. This statement should provide the following information, where applicable:
- Accession codes, unique identifiers, or web links for publicly available datasets
- A description of any restrictions on data availability
- For clinical datasets or third party data, please ensure that the statement adheres to our policy

All sequencing data (RNA-seq, ATAC-seq, ChIP–seq, Hi-C, RRMS, DiMeLo-seq and single cell sequencing) generated for the study have been deposited in the Gene Expression Omnibus (GSE239278) at https://www.ncbi.nlm.nih.gov/geo/query/acc.cgi?acc=GSE239278
The mass spectrometry proteomic data have been deposited to the ProteomeXchange Consortium via the PRIDE96 partner repository with the dataset identifier PXD061127 and PXD061080.
The published RNA-seq, ATAC-seq, and H3K27ac ChIP-seq data from LuCaP PDX, referenced in this study, are available in the GEO database under the accession numbers GSE126078, GSE156292, and GSE161948, respectively. The WGBS and RNA-seq data from human patients, referenced in this study, are available in dbGaP (phs001648 and phs000909.v.p1) and in the GEO database under the accession numbers GSE74685, GSE126078, GSE21034, GSE6919, and GSE77930. The scRNA-seq data from human patients, referenced in this study, are from SRA PRJNA699369.

# Research involving human participants, their data, or biological material

Policy information about studies with human participants or human data. See also policy information about sex, gender (identity/presentation), and sexual orientation and race, ethnicity and racism.

| Reporting on sex and gender | Not applicable |
|---|---|
| Reporting on race, ethnicity, or other socially relevant groupings | Not applicable |
| Population characteristics | Not applicable |
| Recruitment | Not applicable |

| Ethics oversight | Not applicable |
|---|---|

Note that full information on the approval of the study protocol must also be provided in the manuscript.

# Field-specific reporting

Please select the one below that is the best fit for your research. If you are not sure, read the appropriate sections before making your selection.

☒ Life sciences          ☐ Behavioural & social sciences          ☐ Ecological, evolutionary & environmental sciences

For a reference copy of the document with all sections, see nature.com/documents/nr-reporting-summary-flat.pdf

# Life sciences study design

All studies must disclose on these points even when the disclosure is negative.

| Sample size | No statistical methods were used to pre-determine sample sizes but our sample sizes are similar to those reported in previous publications (PMID: 36332622, PMID: 35468964). For each independent in vitro experiment, at least three technical replicates were used. All in vitro experiments were independently repeated at least two or three times. For in vivo experiments, the number of animals was determined based on the variability in tumor take rate and growth and is provided in the respective figure legends.<br>RNA-seq were performed in three biological replicates and ATAC-seq were performed in two biological replicates. One to two replicates were performed for ChIP-seq. There was a good correlation between replicates for the different experiments justifying the chosen sample size in generally. |
|---|---|
| Data exclusions | No data was excluded |
| Replication | The experiments were performed in two biological independent replicates that generally showed high correlations |
| Randomization | Animals were randomly assigned to treatment groups |
| Blinding | Tumor measurements in animals were performed blinded |

# Reporting for specific materials, systems and methods

We require information from authors about some types of materials, experimental systems and methods used in many studies. Here, indicate whether each material, system or method listed is relevant to your study. If you are not sure if a list item applies to your research, read the appropriate section before selecting a response.

## Materials & experimental systems

| n/a | Involved in the study |
|---|---|
| ☐ | ☒ Antibodies |
| ☐ | ☒ Eukaryotic cell lines |
| ☒ | ☐ Palaeontology and archaeology |
| ☐ | ☒ Animals and other organisms |
| ☒ | ☐ Clinical data |
| ☒ | ☐ Dual use research of concern |
| ☒ | ☐ Plants |

## Methods

| n/a | Involved in the study |
|---|---|
| ☐ | ☒ ChIP-seq |
| ☒ | ☐ Flow cytometry |
| ☒ | ☐ MRI-based neuroimaging |

## Antibodies

| Antibodies used | Rabbit polyclonal anti-AR Millipore Cat# 06-680 WB: 1:3000<br>Rabbit polyclonal anti-AR Santa Cruz Biotechnology Cat# sc-816 WB (1:3000) for mouse sample<br>Rabbit polyclonal anti-BRN2 Bethyl Laboratories Cat#A303-583A-M WB: 1:1000<br>Rabbit monoclonal anti-HOXB13 Cell Signaling Technology Cat#90944S WB: 1:5000<br>Rabbit recombinant anti-AR Abcam Cat#ab108341 IHC: 1:1000<br>Rabbit polyclonal anti-histone H3 Abcam Cat#ab1791 WB: 1:200000<br>Mouse monoclonal anti-NCAM/CD56 Santa Cruz Biotechnology Cat#sc-7326 WB: 1:1000<br>Mouse monoclonal anti-SYP/Synaptophysin Santa Cruz Biotechnology Cat#sc-17750 WB/IHC (1:1000 for WB, 1:100 for IHC)<br>Rabbit recombinant anti-FOXA2 Abcam Cat#ab108422 WB/IHC (1:5000 for WB, 1:1000 for IHC)<br>Rabbit recombinant anti-FOXA2 Abcam Cat#ab256493 ChIP/DiMeLo-seq (5μg for ChIP, 1:50 for DiMeLo-seq)<br>Mouse monoclonal anti-TTF1/NKX2-1 Santa Cruz Biotechnology Cat#sc-53136 IHC/Co-IP (1:100 for IHC, 1:250 for Co-IP)<br>Rabbit polyclonal anti-TTF-1/NKX2-1 Sigma-Aldrich Cat#07-601 WB/ChIP (1:1000 for WB, 5μg for ChIP)<br>Rabbit monoclonal anti- acetyl-histone H3 (Lys27) Cell Signaling Technology Cat#8173S WB/ChIP (1:1000 for WB, 1:250 for ChIP)<br>Rabbit polyclonal anti-histone H3 (mono methyl K4) Abcam Cat#ab8895 ChIP: 5μg<br>Rabbit polyclonal anti-p300 Bethyl Laboratories Cat# A300-358A-M WB/ChIP (1:1000 for WB, 5μg for ChIP)<br>Rabbit monoclonal anti- CBP Cell Signaling Technology Cat#7389S WB/ChIP/Co-IP (1:1000 for WB, 1:100 for ChIP, 1:100 for Co-IP) |
|---|---|

Rabbit monoclonal anti- SOX2 Cell Signaling Technology Cat#14962S WB/IHC (1:1000 for WB, 1:500 for IHC)
Rabbit monoclonal anti- PTEN Cell Signaling Technology Cat#9188S WB: 1:1000
Mouse monoclonal anti- TP53 Cell Signaling Technology Cat#2524S WB: 1:1000
Mouse monoclonal anti-GAPDH Proteintech Cat#60004-1-Ig WB: 1:1000
Mouse monoclonal anti-β-Actin Santa Cruz Biotechnology Cat#sc-47778 WB: 1:1000
Rabbit monoclonal anti-HA-Tag  Cell Signaling Technology Cat#3724S WB: 1:5000
Mouse monoclonal anti-HA-Tag Santa Cruz Biotechnology Cat# sc-7392 Co-IP: 1:100
Mouse monoclonal anti-FOXA2 Abnova Cat#H00003170-M12 Co-IP: 2µg/per Co-IP
Rabbit (DA1E) mAb IgG XP® Isotype Control  Cell Signaling Technology Cat#66362 ChIP/DiMeLo-seq (1:100 for ChIP, 1:50 for DiMeLo-seq)
Rabbit monoclonal anti-Tri-Methyl-Histone H3 (Lys4)  Cell Signaling Technology Cat#9751 ChIP: 1:200
Rabbit monoclonal anti-Tri-Methyl-Histone H3 (Lys27) Cell Signaling Technology Cat#9733 Cut&tag:  1:50
Rabbit polyconal anti-CTCF  Sigma-Aldrich Cat#07-729 ChIP: 5µg
Rabbit monoclonal anti-ASCL1 Cell Signaling Technology Cat#10585T WB: 1:1000
Rabbit  monoclonal anti-Cleaved PARP Cell Signaling Technology Cat#5625S WB: 1:1000
Rabbit  monoclonal anti-Cleaved Caspase-3 Cell Signaling Technology Cat#9664S WB: 1:1000
Rabbit  monoclonal anti-p21 Waf1/Cip1 Cell Signaling Technology Cat#2947S WB: 1:1000

| Validation | All antibodies were validated by manufacturers and previously used in several publications.<br>FOXA2 (Cat# ab256493) used for ChIP-seq assay was validated in PMID: 36332622; NKX2-1 (Cat# 07-601) was validated in PMID: 34466783; H3K27ac and H3K4me3 antibodies used for ChIP-seq were validated in PMID: 34937944 ; CTCF ( Cat#07-729) ChIP-seq antibody was validated in PMID: 26091879; H3K4me1 (Cat#ab8895) and H3K27me3 (Cat#9733) were broadly used for ChIP-seq and Cut&tag assays in literature. |
|---|---|

## Eukaryotic cell lines

Policy information about cell lines and Sex and Gender in Research

| Cell line source(s) | Normal prostate epithelia cell line  RWPE-1 and Prostate cancer cell lines LNCaP, C4-2B, 22Rv1, DU145, NCI-H660 and human embryonic kidney cell line HEK293T cells were obtained from American Type Culture Collection (ATCC). Enzalutamide-resistant AR+/PSA– cell line 42D was generated by Dr. Amina Zoubeidi group at University of British Columbia. |
|---|---|
| Authentication | Authenticated using short tandem repeat (STR)profiling |
| Mycoplasma contamination | Cell lines tested negative for Mycoplasm |
| Commonly misidentified lines (See ICLAC register) | No commonly misidentified lines were used in this study |

## Animals and other research organisms

Policy information about studies involving animals; ARRIVE guidelines recommended for reporting animal research, and Sex and Gender in Research

| Laboratory animals | Mouse, NOD SCID ( Charles River Laboratories, Strain Code. 394), Male, 6-8 weeks |
|---|---|
| Wild animals | The study did not involve wild animals |
| Reporting on sex | Male |
| Field-collected samples | The study did not involve field-collected samples |
| Ethics oversight | Mouse handling and experimental procedures were approved by the Institutional Animal Care and Use Committee at Northwestern University in accordance with the US National Institutes of Health Guidelines for the Care and Use of Laboratory Animals and the Animal Welfare Act. |

Note that full information on the approval of the study protocol must also be provided in the manuscript.

## Plants

| | |
|---|---|
| Seed stocks | *Report on the source of all seed stocks or other plant material used. If applicable, state the seed stock centre and catalogue number. If plant specimens were collected from the field, describe the collection location, date and sampling procedures.* |
| Novel plant genotypes | *Describe the methods by which all novel plant genotypes were produced. This includes those generated by transgenic approaches, gene editing, chemical/radiation-based mutagenesis and hybridization. For transgenic lines, describe the transformation method, the number of independent lines analyzed and the generation upon which experiments were performed. For gene-edited lines, describe the editor used, the endogenous sequence targeted for editing, the targeting guide RNA sequence (if applicable) and how the editor was applied.* |
| Authentication | *Describe any authentication procedures for each seed stock used or novel genotype generated. Describe any experiments used to assess the effect of a mutation and, where applicable, how potential secondary effects (e.g. second site T-DNA insertions, mosaicism, off-target gene editing) were examined.* |

## ChIP-seq

### Data deposition

☒ Confirm that both raw and final processed data have been deposited in a public database such as GEO.

☒ Confirm that you have deposited or provided access to graph files (e.g. BED files) for the called peaks.

| | |
|---|---|
| Data access links<br>*May remain private before publication.* | https://www.ncbi.nlm.nih.gov/geo/query/acc.cgi?acc=GSE239278<br>Reviewer access secure token: odqxmgucrvszzur |
| Files in database submission | GSM7662305 LNCaP, FOXA2-D0, H3K27ac ChIP<br>GSM7662306 LNCaP, FOXA2-D2, H3K27ac ChIP<br>GSM7662307 LNCaP, FOXA2-D7, sgNC, H3K27ac ChIP<br>GSM7662308 LNCaP, FOXA2-D7, sgNKX2-1, H3K27ac ChIP<br>GSM7662309 LNCaP, FOXA2-D14, sgNC, H3K27ac ChIP<br>GSM7662310 LNCaP, FOXA2-D14, sgNKX2-1, H3K27ac ChIP<br>GSM7662311 LNCaP, FOXA2-D21, sgNC, H3K27ac ChIP<br>GSM7662312 LNCaP, FOXA2-D21, sgNKX2-1, H3K27ac ChIP<br>GSM7662313 LNCaP, FOXA2-D28, sgNC, H3K27ac ChIP<br>GSM7662314 LNCaP, FOXA2-D28, sgNKX2-1, H3K27ac ChIP<br>GSM7662315 LNCaP, FOXA2-D0, H3K4me1 ChIP<br>GSM7662316 LNCaP, FOXA2-D2, H3K4me1 ChIP<br>GSM7662317 LNCaP, FOXA2-D7, sgNC, H3K4me1 ChIP<br>GSM7662318 LNCaP, FOXA2-D7, sgNKX2-1, H3K4me1 ChIP<br>GSM7662319 LNCaP, FOXA2-D14, sgNC, H3K4me1 ChIP<br>GSM7662320 LNCaP, FOXA2-D14, sgNKX2-1, H3K4me1 ChIP<br>GSM7662321 LNCaP, FOXA2-D21, sgNC, H3K4me1 ChIP<br>GSM7662322 LNCaP, FOXA2-D21, sgNKX2-1, H3K4me1 ChIP<br>GSM7662323 LNCaP, FOXA2-D28, sgNC, H3K4me1 ChIP<br>GSM7662324 LNCaP, FOXA2-D28, sgNKX2-1, H3K4me1 ChIP<br>GSM7662325 LNCaP, FOXA2-D0, FOXA2 ChIP<br>GSM7662326 LNCaP, FOXA2-D2, FOXA2 ChIP<br>GSM7662327 LNCaP, FOXA2-D7, sgNC, FOXA2 ChIP<br>GSM7662328 LNCaP, FOXA2-D7, sgNKX2-1, FOXA2 ChIP<br>GSM7662329 LNCaP, FOXA2-D14, sgNC, FOXA2 ChIP<br>GSM7662330 LNCaP, FOXA2-D14, sgNKX2-1,FOXA2 ChIP<br>GSM7662331 LNCaP, FOXA2-D21, sgNC, FOXA2 ChIP<br>GSM7662332 LNCaP, FOXA2-D21, sgNKX2-1,FOXA2 ChIP<br>GSM7662333 LNCaP, FOXA2-D28, sgNC, FOXA2 ChIP<br>GSM7662334 LNCaP, FOXA2-D28, sgNKX2-1, FOXA2 ChIP<br>GSM7662335 LNCaP, FOXA2-D7, sgNC, NKX2-1 ChIP<br>GSM7662336 LNCaP, FOXA2-D7, sgNKX2-1, NKX2-1 ChIP<br>GSM7662337 LNCaP, FOXA2-D14, sgNC, NKX2-1 ChIP<br>GSM7662338 LNCaP, FOXA2-D14, sgNKX2-1,NKX2-1 ChIP<br>GSM7662339 LNCaP, FOXA2-D21, sgNC, NKX2-1 ChIP<br>GSM7662340 LNCaP, FOXA2-D21, sgNKX2-1, NKX2-1 ChIP<br>GSM7662341 LNCaP, FOXA2-D28, sgNC, NKX2-1 ChIP<br>GSM7662342 LNCaP, FOXA2-D28, sgNKX2-1, NKX2-1 ChIP<br>GSM7662343 LNCaP, FOXA2-D2, CTCF ChIP<br>GSM7662344 LNCaP, FOXA2-D14, CTCF ChIP<br>GSM7662345 LNCaP, FOXA2-D21, CTCF ChIP<br>GSM7662346 LNCaP, FOXA2-D28, CTCF ChIP<br>GSM7662347 LuCaP35CR, H3K27ac ChIP<br>GSM7662348 LuCaP147CR, H3K27ac ChIP<br>GSM7662349 LuCaP93, H3K27ac ChIP<br>GSM7662350 LuCaP145.2, H3K27ac ChIP |

GSM7662351 NCI-H660, FOXA2 ChIP 1#
GSM7662352 NCI-H660, NKX2-1 ChIP 1#
GSM7662353 LNCaP, FOXA2-D2, H3K27me3 Cut&tag
GSM7662354 LNCaP, FOXA2-D14, H3K27me3 Cut&tag
GSM7662355 LNCaP, FOXA2-D21, H3K27me3 Cut&tag
GSM7662356 LNCaP, FOXA2-D28, H3K27me3 Cut&tag
GSM7662357 LNCaP, FOXA2-D14, FOXA2 ChIP
GSM7662358 LNCaP, FOXA2+NKX2-1-D14, FOXA2 ChIP
GSM7662359 LuNE, sgNC, p300 ChIP
GSM7662360 LuNE, sgNKX2-1, p300 ChIP
GSM7662361 LuNE, sgFOXA2, p300 ChIP
GSM7662362 NCI-H660, FOXA2 ChIP 2#
GSM7662363 NCI-H660, NKX2-1 ChIP 2#
GSM7662364 LuCaP145.2, FOXA2 ChIP
GSM7662365 LuCaP145.2, NKX2-1 ChIP
GSM7662366 LuNE, DMSO, H3K27ac ChIP 1#
GSM7662367 LuNE, CCS1477, H3K27ac ChIP 1#
GSM7662368 LuNE, shCtrl, H3K27ac ChIP 1#
GSM7662369 LuNE, shp300, H3K27ac ChIP 1#
GSM7662370 LuNE, shCBP, H3K27ac ChIP 1#
GSM7662371 LuNE, DMSO, H3K27ac ChIP 2#
GSM7662372 LuNE, CCS1477, H3K27ac ChIP 2#
GSM7662373 LuNE, shCtrl, H3K27ac ChIP 2#
GSM7662374 LuNE, shp300, H3K27ac ChIP 2#
GSM7662375 LuNE, shCBP, H3K27ac ChIP 2#
GSM7662376 NCI-H660, H3K4me1 ChIP 1#
GSM7662377 NCI-H660, H3K4me1 ChIP 2#
GSM7662378 NCI-H660, H3K4me3 ChIP
GSM7662379 LuCaP145.2, H3K4me1 ChIP 1#
GSM7662380 LuCaP145.2, H3K4me1 ChIP 2#
GSM7662381 NCI-H660, H3K27ac ChIP
GSM7662382 LuNE, sgNC, H3K27ac ChIP 1#
GSM7662383 LuNE, sgNKX2-1, H3K27ac ChIP 1#
GSM7662384 LuNE, sgFOXA2, H3K27ac ChIP 1#
GSM7662385 LuNE, sgNC, H3K27ac ChIP 2#
GSM7662386 LuNE, sgNKX2-1, H3K27ac ChIP 2#
GSM7662387 LuNE, sgFOXA2, H3K27ac ChIP 2#
GSM7662388 LuNE, FOXA2 ChIP 1#
GSM7662389 LuNE, FOXA2 ChIP 2#
GSM8061324 LuCaP145.1, H3K27ac ChIP
GSM8061325 LuCaP70CR, H3K27ac ChIP
GSM8061326 LuCaP77CR, H3K27ac ChIP
GSM8061327 LuCaP35CR, H3K27me3 Cut&Tag
GSM8061328 LuCaP93, H3K27me3 Cut&Tag
GSM8061329 LuCaP145.2, H3K27me3 Cut&Tag
GSM8061330 LuCaP147, H3K27me3 Cut&Tag
GSM8061331 LuCaP70CR, H3K27me3 Cut&Tag
GSM8061332 LuCaP145.1, H3K27me3 Cut&Tag
GSM8061333 NCI-H660, H3K27me3 Cut&Tag
GSM8061334 LuCaP77CR, H3K27me3 Cut&Tag
GSM8061335 LuCaP93, NKX2-1 ChIP
GSM8061336 LuCaP145.1, H3K4me1 ChIP
GSM8061337 LuCaP145.1, FOXA2 ChIP
GSM8061338 LuCaP145.1, NKX2-1 ChIP
GSM8373006 LuCaP93, H3K4me1 ChIP

| Genome browser session<br>(e.g. UCSC) | FOXA2, H3K4me1,H3K27ac and NKX2-1 time-course ChIP-seq tracks:<br>https://genome.ucsc.edu/cgi-bin/hgTracks?<br>db=hg19&lastVirtModeType=default&lastVirtModeExtraState=&virtModeType=default&virtMode=0&nonVirtPosition=&position=chr14%3A36970296%2D37004738&hgsid=1892428022_N0ACZGZO2xDNOXfQyjpoKwAA5Zmi |

## Methodology

| Replicates | One or two replicates were performed for ChIP-seq experiments, and the data were pooled for analysis. |

| Sequencing depth | M893 LNCaP, FOXA2-D0, H3K27ac ChIP 50bp single<br>M894 LNCaP, FOXA2-D2, H3K27ac ChIP 50bp single<br>M895 LNCaP, FOXA2-D7, sgNC, H3K27ac ChIP 50bp single<br>M896 LNCaP, FOXA2-D7, sgNKX2-1, H3K27ac ChIP 50bp single<br>M897 LNCaP, FOXA2-D14, sgNC, H3K27ac ChIP 50bp single<br>M898 LNCaP, FOXA2-D14, sgNKX2-1, H3K27ac ChIP 50bp single<br>M899 LNCaP, FOXA2-D21, sgNC, H3K27ac ChIP 50bp single<br>M900 LNCaP, FOXA2-D21, sgNKX2-1, H3K27ac ChIP 50bp single<br>M901 LNCaP, FOXA2-D28, sgNC, H3K27ac ChIP 50bp single<br>M902 LNCaP, FOXA2-D28, sgNKX2-1, H3K27ac ChIP 50bp single |

M903 LNCaP, FOXA2-D0, H3K4me1 ChIP 50bp single
M904 LNCaP, FOXA2-D2, H3K4me1 ChIP 50bp single
M905 LNCaP, FOXA2-D7, sgNC, H3K4me1 ChIP 50bp single
M906 LNCaP, FOXA2-D7, sgNKX2-1, H3K4me1 ChIP 50bp single
M1390 LNCaP,DMSO,H3K27ac ChIP 150bp paired
M1392 LNCaP,CCS1477,H3K27ac ChIP 150bp paired

M907 LNCaP, FOXA2-D14, sgNC, H3K4me1 ChIP 50bp single
M908 LNCaP, FOXA2-D14, sgNKX2-1, H3K4me1 ChIP 50bp single
M909 LNCaP, FOXA2-D21, sgNC, H3K4me1 ChIP 50bp single
M910 LNCaP, FOXA2-D21, sgNKX2-1, H3K4me1 ChIP 50bp single
M911 LNCaP, FOXA2-D28, sgNC, H3K4me1 ChIP 50bp single
M912 LNCaP, FOXA2-D28, sgNKX2-1, H3K4me1 ChIP 50bp single
M913 LNCaP, FOXA2-D0, FOXA2 ChIP 50bp single
M914 LNCaP, FOXA2-D2, FOXA2 ChIP 50bp single
M915 LNCaP, FOXA2-D7, sgNC, FOXA2 ChIP 50bp single
M916 LNCaP, FOXA2-D7, sgNKX2-1, FOXA2 ChIP 50bp single
M917 LNCaP, FOXA2-D14, sgNC, FOXA2 ChIP 50bp single
M918 LNCaP, FOXA2-D14, sgNKX2-1,FOXA2 ChIP 50bp single
M919 LNCaP, FOXA2-D21, sgNC, FOXA2 ChIP 50bp single
M920 LNCaP, FOXA2-D21, sgNKX2-1,FOXA2 ChIP 50bp single
M921 LNCaP, FOXA2-D28, sgNC, FOXA2 ChIP 50bp single
M922 LNCaP, FOXA2-D28, sgNKX2-1, FOXA2 ChIP 50bp single
M923 LNCaP, FOXA2-D7, sgNC, NKX2-1 ChIP 50bp single
M924 LNCaP, FOXA2-D7, sgNKX2-1, NKX2-1 ChIP 50bp single
M925 LNCaP, FOXA2-D14, sgNC, NKX2-1 ChIP 50bp single
M926 LNCaP, FOXA2-D14, sgNKX2-1,NKX2-1 ChIP 50bp single
M927 LNCaP, FOXA2-D21, sgNC, NKX2-1 ChIP 50bp single
M928 LNCaP, FOXA2-D21, sgNKX2-1, NKX2-1 ChIP 50bp single
M929 LNCaP, FOXA2-D28, sgNC, NKX2-1 ChIP 50bp single
M930 LNCaP, FOXA2-D28, sgNKX2-1, NKX2-1 ChIP 50bp single
M1015 LNCaP, FOXA2-D2, CTCF ChIP 50bp paired-end
M1016 LNCaP, FOXA2-D14, CTCF ChIP 50bp paired-end
M1017 LNCaP, FOXA2-D21, CTCF ChIP 50bp paired-end
M1018 LNCaP, FOXA2-D28, CTCF ChIP 50bp paired-end
M994 LuCaP35CR, H3K27ac ChIP 50bp paired-end
M998 LuCaP147CR, H3K27ac ChIP 50bp paired-end
M1026 LuCaP93, H3K27ac ChIP 50bp paired-end
M1027 LuCaP145.2, H3K27ac ChIP 50bp paired-end
M1043 NCI-H660, FOXA2 ChIP 1# 50bp paired-end
M1044 NCI-H660, NKX2-1 ChIP 1# 50bp paired-end
M1047 LNCaP, FOXA2-D2,  H3K27me3 Cut&tag 50bp paired-end
M1048 LNCaP, FOXA2-D14, H3K27me3 Cut&tag 50bp paired-end
M1049 LNCaP, FOXA2-D21, H3K27me3 Cut&tag 50bp paired-end
M1050 LNCaP, FOXA2-D28, H3K27me3 Cut&tag 50bp paired-end
M1205 LNCaP, FOXA2-D14, FOXA2 ChIP 50bp single
M1235 LNCaP, FOXA2+NKX2-1-D14, FOXA2 ChIP 50bp paired-end
M1241 LuNE, sgNC, P300 ChIP 50bp paired-end
M1242 LuNE, sgNKX2-1, P300 ChIP 50bp paired-end
M1243 LuNE, sgFOXA2, P300 ChIP 50bp paired-end
M1135 NCI-H660, FOXA2 ChIP 2# 50bp paired-end
M1136 NCI-H660, NKX2-1 ChIP 2# 50bp paired-end
M1380 LuCaP145.2, FOXA2 ChIP 50bp paired-end
M1152 LuCaP145.2, NKX2-1 ChIP 50bp paired-end
M1331 LuNE, DMSO, H3K27ac ChIP 1# 50bp single
M1332 LuNE, CCS1477, H3K27ac ChIP 1# 50bp single
M1335 LuNE, shCtrl, H3K27ac ChIP 1# 50bp single
M1336 LuNE, shP300, H3K27ac ChIP 1# 50bp single
M1337 LuNE, shCBP, H3K27ac ChIP 1# 50bp single
M1323 LuNE, DMSO, H3K27ac ChIP 2# 50bp paired-end
M1324 LuNE, CCS1477, H3K27ac ChIP 2# 50bp paired-end
M1327 LuNE, shCtrl, H3K27ac ChIP 2# 50bp paired-end
M1328 LuNE, shP300, H3K27ac ChIP 2# 50bp paired-end
M1329 LuNE, shCBP, H3K27ac ChIP 2# 50bp paired-end
M1377 NCI-H660, H3K4me1 ChIP 1# 50bp paired-end
M1378 NCI-H660, H3K4me1 ChIP 2# 50bp paired-end
M1379 NCI-H660, H3K4me3 ChIP 50bp paired-end
M1382 LuCaP145.2, H3K4me1 ChIP 1# 50bp paired-end
M1383 LuCaP145.2, H3K4me1 ChIP 2# 50bp paired-end
M1131 NCI-H660, H3K27ac ChIP 50bp paired-end
M1407 LuNE, sgNC, H3K27ac ChIP 1# 150bp paired-end
M1408 LuNE, sgNKX2-1, H3K27ac ChIP 1# 150bp paired-end
M1409 LuNE, sgFOXA2, H3K27ac  ChIP 1# 150bp paired-end
M1411 LuNE, sgNC, H3K27ac ChIP 2# 150bp paired-end
M1412 LuNE, sgNKX2-1, H3K27ac  ChIP 2# 150bp paired-end
M1413 LuNE, sgFOXA2, H3K27ac  ChIP 2# 150bp paired-end

M1103 LuNE, FOXA2 ChIP 1# 50bp paired-end
M1108 LuNE, FOXA2 ChIP 2# 50bp paired-end
M1507 LuCaP145.1, H3K27ac ChIP 150bp paired-end
M995 LuCaP70CR, H3K27ac ChIP 50bp paired-end
M996 LuCaP77CR, H3K27ac ChIP 50bp paired-end
M1466 LuCaP35CR, H3K27me3 Cut&Tag 150bp paired-end
M1467 LuCaP93, H3K27me3 Cut&Tag 150bp paired-end
M1468 LuCaP145.2, H3K27me3 Cut&Tag 150bp paired-end
M1469 LuCaP147, H3K27me3 Cut&Tag 150bp paired-end
M1496 LuCaP70CR, H3K27me3 Cut&Tag 150bp paired-end
M1497 LuCaP145.1, H3K27me3 Cut&Tag 150bp paired-end
M1498 NCI-H660, H3K27me3 Cut&Tag 150bp paired-end
M1529 LuCaP77CR, H3K27me3 Cut&Tag 150bp paired-end
M1505 LuCaP93, NKX2-1 ChIP 150bp paired-end
M1506 LuCaP145.1, H3K4me1 ChIP 150bp paired-end
M1508 LuCaP145.1, FOXA2 ChIP 150bp paired-end
M1509 LuCaP145.1, NKX2-1 ChIP 150bp paired-end
M1457 LuCaP93, H3K4me1 ChIP 150bp paired-end

| Antibodies | Rabbit recombinant anti-FOXA2  Abcam Cat#ab256493 ChIP/DiMeLo-seq |
|---|---|
| | Rabbit polyclonal anti-TTF-1/NKX2-1 Sigma-Aldrich Cat#07-601 ChIP |
| | Rabbit monoclonal anti- acetyl-histone H3 (Lys27) Cell Signaling Technology Cat#8173S ChIP |
| | Rabbit polyclonal anti-histone H3 (mono methyl K4)  Abcam Cat#ab8895 ChIP |
| | Rabbit polyclonal anti-p300  Bethyl Laboratories Cat# A300-358A-M ChIP |
| | Rabbit (DA1E) mAb IgG XP® Isotype Control  Cell Signaling Technology Cat#66362 ChIP/DiMeLo-seq |
| | Rabbit monoclonal anti-Tri-Methyl-Histone H3 (Lys4)  Cell Signaling Technology Cat#9751 ChIP |
| | Rabbit monoclonal anti-Tri-Methyl-Histone H3 (Lys27) Cell Signaling Technology Cat#9733 Cut&Tag |
| | Rabbit polyconal anti-CTCF Sigma-Aldrich Cat#07-729 ChIP |

| Peak calling parameters | Bowtie2, HOMER and MACS2, all used default parameters. |
|---|---|

| Data quality | Raw sequencing quality was assessed with FastQC/0.11.5, cross-correlation enrichment metrics were checked by deepTools/2.0 plotCorrelation tool. |
|---|---|

| Software | FastQC/0.11.5 |
|---|---|
| | samtools/1.6 |
| | Bowtie2/2.0.5 |
| | HOMER/4.8.3 |
| | MACS/2.0 |
| | samtools/1.6 |
| | Vennerable/3.0 |
| | deepTools/2.0 |

