## [Peer Review File · Nature Genetics]

NKX2-1 drives neuroendocrine transdifferentiation of prostate cancer via epigenetic and 3D chromatin remodeling

Corresponding Author: Professor Jindan Yu

A version of this paper was originally rejected for publication by Nature Genetics, however that decision was reconsidered after appeal by the authors.

Version 0:

Decision Letter:

3rd Aug 2023

Dear Dr Yu,

Thank you for submitting your manuscript entitled "Epigenetic landscape and 4D nucleome orchestrated by lineage-specific transcription factors drive neuroendocrine tumors", for consideration. I regret that we are unable to publish it in Nature Genetics.

As you may know, we decline a substantial proportion of manuscripts without sending them to referees, so that they may be sent elsewhere without delay. Our editorial judgments are based on such considerations as the degree of advance provided, the breadth of potential interest to researchers and timeliness.

In this case, your manuscript has not matched our criteria for further consideration at Nature Genetics, and we think it would find a more suitable outlet in another journal, namely Nature Communications. To discover more about our sister journal and should you wish to have your paper reviewed there, please use the link to the manuscript transfer service provided in the footnote below. Please note that we have not consulted with our editorial colleagues at other journals and that they will make their own independent editorial decision.

Please be assured that this editorial decision does not represent a criticism of the quality of your work, nor are we questioning its value to others working in this area. We hope that you will rapidly receive a more favorable response elsewhere.

Although we cannot offer to publish your manuscript, my colleagues at Nature Communications will send your manuscript out for external review. To transfer your manuscript please use our manuscript transfer portal. You will not have to re-supply manuscript metadata and files, unless you wish to make modifications, but please note that this link can only be used once and remains active until used. For more information, please see our [manuscript transfer FAQ](http://www.nature.com/authors/author_resources/transfer_manuscripts.html?WT.mc_id=EMI_NPG_1511_AUTHORTRANSF&WT.ec_id=AUTHOR) page.

I am sorry that we cannot respond more positively on this occasion.

Sincerely,
Chiara

Chiara Anania, PhD
Associate Editor
Nature Genetics
<https://orcid.org/0000-0003-1549-4157>

Version 1:

Decision Letter:

IMPORTANT: Please note the reference number: NG-A63095R-Z Yu. This number must be quoted whenever you communicate with us regarding this paper.

26th Jan 2024

Dear Dr. Yu,

Thank you for asking us to reconsider our decision on your manuscript "Epigenetic remodeling and 3D chromatin reorganization governed by NKX2-1 drive neuroendocrine prostate cancer". I have now discussed the points of your letter with my colleagues, and we think that you have some valid points. We therefore invite you to revise your manuscript along the lines that you propose.

When preparing a revision, please ensure that it fully complies with our editorial requirements for format and style; details can be found in the Guide to Authors on our website (<http://www.nature.com/ng/>).

Please be sure that your manuscript is accompanied by a separate letter detailing the changes you have made and your response to the points raised. At this stage we will need you to upload:

1) a copy of the manuscript in MS Word .docx format.

2) The Editorial Policy Checklist:

<https://www.nature.com/documents/nr-editorial-policy-checklist.pdf>

3) The Reporting Summary:

(Here you can read about the role of the Reporting Summary in reproducible science:

<https://www.nature.com/news/announcement-towards-greater-reproducibility-for-life-sciences-research-in-nature-1.22062>)

Please use the link below to be taken directly to the site and view and revise your manuscript:

Link Redacted

With kind wishes,
Chiara

Chiara Anania, PhD
Associate Editor
Nature Genetics
<https://orcid.org/0000-0003-1549-4157>

Version 2:

Decision Letter:

19th Mar 2024

Dear Professor Yu,

Your Article, "Epigenetic remodeling and 3D chromatin reorganization governed by NKX2-1 drive neuroendocrine prostate cancer" has now been seen by 3 referees. You will see from their comments below that while they find your work of interest, some important points are raised. We are interested in the possibility of publishing your study in Nature Genetics, but would like to consider your response to these concerns in the form of a revised manuscript before we make a final decision on publication.

To guide the scope of the revisions, the editors discuss the referee reports in detail within the team, including with the chief

editor, with a view to identifying key priorities that should be addressed in revision and sometimes overruling referee requests that are deemed beyond the scope of the current study. In this case, we would like you to address Reviewers' comments in full. Please do not hesitate to get in touch if you would like to discuss this further.

We therefore invite you to revise your manuscript taking into account all reviewer and editor comments. Please highlight all changes in the manuscript text file. At this stage we will need you to upload a copy of the manuscript in MS Word .docx or similar editable format.

*2) If you have not done so already please begin to revise your manuscript so that it conforms to our Article format instructions, available

http://www.nature.com/ng/authors/article_types/index.html here

*3) Include a revised version of any required Reporting Summary: <https://www.nature.com/documents/nr-reporting-summary.pdf>

Please be aware of our <https://www.nature.com/nature-research/editorial-policies/image-integrity> guidelines on digital image standards.

Link Redacted

We hope to receive your revised manuscript within four to eight weeks. If you cannot send it within this time, please let us know.

Nature Genetics is committed to improving transparency in authorship. As part of our efforts in this direction, we are now requesting that all authors identified as 'corresponding author' on published papers create and link their Open Researcher and Contributor Identifier (ORCID) with their account on the Manuscript Tracking System (MTS), prior to acceptance. ORCID helps the scientific community achieve unambiguous attribution of all scholarly contributions. You can create and link your ORCID from the home page of the MTS by clicking on 'Modify my Springer Nature account'. For more information please visit please visit <http://www.springernature.com/orcid>.

Sincerely,
Chiara

Chiara Anania, PhD
Associate Editor
Nature Genetics
<https://orcid.org/0000-0003-1549-4157>

Referee expertise:

Referee #1: genomics, prostate cancer

Referee #2: functional genomics, cancer

Referee #3: cancer epigenetics

Reviewers' Comments:

Reviewer #1:

Remarks to the Author:

In this work, the authors performed Hi-C analysis on PDX models and performed a series of epigenomic and transcriptomic studies using a FOXA2 overexpression system with repeated sampling over 28 days. They documented decreased methylation at sites where FOXA2 binding increases during the 28 day time course and then performed in vivo studies with a model of induced NEPC to assess the impact of p300 inhibition on tumor growth. A significant amount of work is presented over seven figures, and the number of different analyses presented in this study is a strength. The text is clearly written (with minor typos that can be fixed in editing) and the figures are straightforward. My primary concerns are that it is unclear how well this model reflects the mechanism observed in patients, that the main overexpression model is has previously been demonstrated, and that the effect of CCS1477 in these cells is not obviously linked to the NEPC phenotype.

Major issues:

1) The Hopkins paper cited by the authors to justify the claim that NKX2-1 is "highly positive in more than 50% of NE lesions" reported on de novo small cell prostate cancer (not therapy-induced disease) and does not mention NKX2-1. What is the evidence that NKX2-1 overexpression both frequent and a cause of NEPC in mCRPC patients? The authors suggest NKX2-1 is essential for NEPC. Is it really expressed in all NEPC samples? The authors should use published clinical genomics or epigenomics data to strengthen this claim.

2) The authors write "NKX2-1 in PCa [has] not been carefully studied". NKX2-1 was a focus of Baca et al. (ref 23). The authors should acknowledge this point and define how their work extends prior observations.

3) The authors report CCS1477 controls tumor growth of the LuNE NEPC tumor model in SCID mice. This compound targets P300/CBP bromodomain. Although the authors characterize CCS1477 as "in clinical trials in CRPC patients via targeting AR", this compound is not specific to AR as it has been shown (Welti 2021) to target cMYC among other important drivers of AR-independent prostate cancer. Inclusion criteria for NCT04575766, as I read them, would allow NEPC patients to enroll. This therapy is also being tested in haematological malignancies (NCT04068597) and maybe other indications that I'm not aware of. Prior demonstrated efficacy in AR+ models of mCRPC implies the drug's mechanism is not contingent on impacting NE enhancer activity. What evidence supports the hypothesis that the specific impact of this compound on LuNE growth is related to its impact on NE enhancers?

Other issues

1) How is significance for the Aggregate Peak Analysis measured? The authors report significant differences without giving details or quantification of the strength of the differences.

2) The authors report Hi-C rewiring during 28 day exposure to FOXA2 OE. Most correlations here are weak but D21 most strongly correlates with D0 and D14 (0.7, 0.6). D28 is most strongly correlated with D21 (0.4) and has negligible (0.1 to -0.2) correlation with NEPC cell lines. I'm not convinced there is a positive association here.

3) The material presenting methylation analysis convincingly shows an association between FOXA2 binding and decreased methylation. The authors do not present mechanistic evidence of this activity, but cite prior work by others that makes this case. This section appears to have some repetitive material and could be shortened.

4) In the comparisons of gene expression levels, the key results (e.g. Fig 2A) are convincing. However, the methods state differential expression was identified using Deseq2, while the figure legend says a paired t test was applied. Neither refers to multiple test correction. Please clarify these details of the methods.

5) In the abstract, presumably the phrase "attributing to NE" should be "contributing to NE". The introduction contains some minor errors in English that should be corrected.

Reviewer #2:

Remarks to the Author:

In this manuscript, the authors uncover how 3D genome architecture plays a critical role in driving disease phenotypes. The authors utilize patient derived xenograft and isogenic cell line models to study prostate neuroendocrine transdifferentiation. Utilizing 3D chromatin looping assays like HiC, the authors identify mCRPC and NEPC specific looping structures that can distinguish the two disease types. Notably, they find that FOXA2, a pioneer transcription factor drives the transdifferentiation by collaborating with NKX2-1. FOXA2 has been implicated in NEPC previously by others (Han et al, Cancer Cell 2022).

In this study, the authors demonstrate that FOXA2 primes NE enhancers by DNA hypomethylation, leading to activation of NE-genes. Finally, the authors show that NKX2-1 and FOXA2 interact with P300/CBP to epigenetically mark NE enhancers for activation and demonstrated that targeting p300/CBP via specific inhibitor CCS1477 could effectively repress NEPC xenograft tumors.

While the isogenic models provide some evidence for CRPC to NEPC transdifferentiation, some mechanistic questions

remain. Below are specific questions/comments that need to be addressed:

1. Do the authors have any evidence to show that NKX2-1 is a pioneer TF as well?
2. Can the authors provide a panel of PCa cell lines to show expression of NKX2-1 and FOXA2? Do any CRPC lines express these (Particularly DU145 and PC3)?
3. Figure 1j: The CRPC loops have a significant overlap with NE-CREs. How does this compare in the context of a primary PCa sample?
4. Figure S2B: Can the authors also provide immunoblots for FOXA1 and NKX3-1 expression levels?
5. Fig 3c-j: What are the leading edge genes that are defining the two distinct subsets of D0 and D21? And how are these genes impacted upon NKX2-1 or FOXA2 loss?
6. In the rebuttal, the authors pointed out that in some of FOXA2 negative PDX samples, NKX2-1 level still increases. As shown in Fig. 4, NKX2-1 is induced by FOXA2. If this is the case, what is the mechanism of NKX2-1 high expression in FOXA2 negative cells? Is NKX2-1 still important for FOXA2 negative NEPC cells?
7. Furthermore, authors claim that NKX2-1 and FOXA2 interact with each other to facilitate the chromatin looping and upregulate downstream gene expression. If this is the case in FOXA2 positive NEPC, what is the mechanism of action of NKX2-1 in FOXA2 negative PDX? Do FOXA2 negative NEPC cells have distinct 3D chromatin structures as FOXA2 positive NEPC cells?
8. Is NKX2-1's induction by FOXA2 evident at the mRNA level?
9. Regarding statement "FOXA2-only binding sites were marked with H3K4me1 only, indicative of primed enhancers", based on Figure 4e, it doesn't seem to be the case. Are the authors referring to another piece of data in support of this claim?
10. The authors show that NKX2-1 is a promoter binding protein while FOXA2 is an enhancer binding protein.
11. Figure 4g: Can the authors also provide expression of other luminal markers like FOXA1 and NKX2-1, and NEPC markers like ASCL1?
12. The author applied p300/CBP bromodomain inhibitor, CCS1477 to block the NEPC tumor growth. However, several studies showed that p300/CBP bromodomain inhibitors exhibit poor efficacy in AR negative cells (DU145 and PC3, Jonathan Welti et al and Lingyan Jin et al). NEPC is AR negative as well, so the data presented here does not seem consistent with other reports.
13. Figure 7M: Does treatment of NEPC models with CCS1477 induce expression of luminal markers like PSA, AR, etc.? How reversible is the FOXA2/NKX2.1-driven NEPC?
14. What is the NKX2-1 status in DU and PC3 cells? Are these 2 cells also dependent on NKX2-1?

Minor comments:

1. Fig.7a, the mass spec data suggests that SWI/SNF complex also interact with FOXA2. Is there any data showing that targeting SWI/SNF complex could repress NEPC growth?
2. Fig. 7b, the quality of co-IP is poor and needs to be improved.

Reviewer #3:

Remarks to the Author:

The study presents interesting findings on the chromatin architecture differences between neuroendocrine prostate cancer (NEPC) and castration-resistant prostate cancer (CRPC), providing a deep understanding of the mechanisms driving NEPC progression. Through Hi-C analyses of patient-derived xenograft tumors and in vitro experiments on isogenic cells undergoing neuroendocrine transformation (NET), the authors demonstrate significant 3D chromatin re-organization in NEPC samples compared to CRPC. Mechanistically, they identify the neural transcription factor NKX2-1 as a key regulator of NET, orchestrating a hierarchical network of transcription factors and chromatin modifiers. This study expands on recent studies of FOXA2 in NEPC, by showing that NKX2-1 is indispensable for the phenotype driven by FOXA2. Additionally, single-cell multiome analyses reveal intermediate epigenetic and transcriptomic states during luminal-to-NE transformation, providing insights into the evolution of NEPC. The finding of dependency on CBP/p300 and the therapeutic potential of CBP/p300 is very clinically relevant.

In summary, the study presents several novel insights into the role of chromatin dynamics in NEPC progression, filling gaps in understanding the molecular mechanisms underlying this aggressive subtype of prostate cancer. While previous studies have explored the role of FOXA2 in cancer progression, this work integrates chromatin architecture analysis with transcriptional regulation to unveil a comprehensive regulatory network of FOXA2-NKX2-1-CBP/p300 governing NEPC development. However, a few questions need to be addressed to strengthen the conclusions.

1. It is very clear that FOXA2 overexpression induces neurite extension morphology and expression of NE markers. Does FOXA2 overexpression drives the small cell phenotype in xenograft models? If not, please discuss potential reasons.
2. The time course scRNA-seq data provides a very useful resource to understand the evolution from CRPC to NEPC. However, the conclusion related to cell transformation vs. populational switch in line 249-285 is premature. As there is a small population of D0 cells demonstrating low level of AR/ARS., how do we know the increased percentage of AR-/NES+ cells in D14, D21 and D28 is not due to clonal expansion of the pre-existing population? Does this population in D0 cluster together with the AR-/NES+ in D28?
3. The feedback loop of FOXA2 inducing NKX2-1 at transcriptional level, and NKX2-1 interacting with FOXA2 to activates NE regulatory sites is very interesting. Figure 4A showed increased FOXA2 binding near NKX2-1. Is there any evidence from the Hi-C data that these regions form loops with NKX2-1 promoter?
4. In the D28 scRNA-seq, are there FOXA2+/NKX2-1 negative cell population? If yes, any difference of NES compared to FOXA2+/NKX2-1+ cells?

5. Is there FOXA2 expression in the P53/RB1 double KD model? The author claimed that NKX2-1 is as a general regulator of NET that likely also cooperates with other chromatin-pioneering factors. Is there any evidence from the NKX2-1 ChIP-seq data of the potential pioneering factors, e.g., motifs in NKX2-1 binding sites that do not overlap with FOXA2 binding?
6. The finding of CBP/P300 from FOXA2 IP mass spec and the potential therapeutic effect of CBP/p300 inhibitor is intriguing. Figure 7M demonstrated a reduction of SOX2 and NKX2-1 in CCS1477 treated LuCaP145.2 tumors. Do these tumors remain as small cell or demonstrate features of adenocarcinoma?
7. Figure 1A. Please add description of Aggregate Peak Analysis in Figure legend or main text.
8. Figure resolution needs to be improved for many of the panels. Some of the text in Figure 1E-F overlaps; adjustments to alignment are needed.
9. Line 135, please rephrase 'erased' as it is unclear whether the loss of H3K27me3 in NEPC is due to loss of writer or gain of eraser activities.

Version 3:

Decision Letter:

6th Aug 2024

Dear Professor Yu,

Your Article, "Epigenetic remodeling and 3D chromatin reorganization governed by NKX2-1 drive neuroendocrine transdifferentiation of prostate cancer" has now been seen by 3 referees. You will see from their comments below that while they find your work of interest, some important points remain. We are interested in the possibility of publishing your study in Nature Genetics, but would like to consider your response to these concerns in the form of a revised manuscript before we make a final decision on publication.

To guide the scope of the revisions, the editors discuss the referee reports in detail within the team, including with the chief editor, with a view to identifying key priorities that should be addressed in revision and sometimes overruling referee requests that are deemed beyond the scope of the current study. In this case, we ask you to address Reviewers' comments in full. Please do not hesitate to get in touch if you would like to discuss these issues further.

We therefore invite you to revise your manuscript taking into account all reviewer and editor comments. Please highlight all changes in the manuscript text file. At this stage we will need you to upload a copy of the manuscript in MS Word .docx or similar editable format.

*2) If you have not done so already please begin to revise your manuscript so that it conforms to our Article format instructions, available

[here](http://www.nature.com/ng/authors/article_types/index.html).

*3) Include a revised version of any required Reporting Summary: <https://www.nature.com/documents/nr-reporting-summary.pdf>

Please be aware of our [guidelines](https://www.nature.com/nature-research/editorial-policies/image-integrity) on digital image standards.

Link Redacted

We hope to receive your revised manuscript within four to eight weeks. If you cannot send it within this time, please let us know.

Sincerely,
Chiara

Chiara Anania, PhD
Associate Editor
Nature Genetics
<https://orcid.org/0000-0003-1549-4157>

Referee expertise:

Referee #1:

Referee #2:

Referee #3:

Reviewers' Comments:

Reviewer #1:

Remarks to the Author:

The manuscript has been significantly strengthened by the author's careful response to all three reviewer's comments. The major findings of the manuscript are well-supported and overall the interpretation of the results is reasonable.

Major comments:

1) The investigators report that "CCS1477 targets lineage-specific enhancers and genes". Does CCS1477 target distinct collections of lineage-specific enhancers in adeno vs. NEPC cells, or does it act generally on active enhancers? The investigators report that treating LNCaP cells with CCS1477 affects "prostate luminal cell functions, such as cell adhesion, secretion, and lipid metabolic process". These GO terms in the LNCaP case are activities that take place in most cells, and it is unclear how they demonstrate lineage specificity. To demonstrate enrichment for lineage-specific enhancers, in comparison to other enhancers, the investigators should specifically test this claim by defining two sets of enhancers: lineage-specific enhancers, and non-lineage-specific enhancers, and demonstrate that there is enrichment for CCS1477-induced activity in the lineage-specific enhancers compared to a background of all other enhancer sites.

2) The investigators should clarify whether CCS1477 is cytostatic or cytotoxic, as they alternately refer to growth-inhibitory effects and the phrase "strong killing effect".

3) The investigators must provide details in their methods describing how they assessed RNA velocity, particularly in light of recent reports that different published methods for calculating velocity can produce results that strongly disagree (e.g. <https://www.biorxiv.org/content/10.1101/2024.06.25.600667v1>). It would be helpful to apply a method showing that their data have noise properties that are suitable for RNA velocity analysis.

Minor points

1) In the discussion, the word "generable" should be "general".

Reviewer #2:

Remarks to the Author:

In this manuscript, the authors uncover how 3D genome architecture plays a critical role in driving disease phenotypes. The authors utilize patient derived xenograft and isogenic cell line models to study prostate neuroendocrine transdifferentiation. Utilizing 3D chromatin looping assays like HiC, the authors identify mCRPC and NEPC specific looping structures that can distinguish the two disease types. Notably, they find that FOXA2, a pioneer transcription factor drives the transdifferentiation by collaborating with NKX2-1. FOXA2 has been implicated in NEPC previously by others (Han et al, Cancer Cell 2022).

In this study, the authors demonstrate that FOXA2 primes NE enhancers by DNA hypomethylation, leading to activation of NE-genes. Finally, the authors show that NKX2-1 and FOXA2 interact with P300/CBP to epigenetically mark NE enhancers for activation and demonstrated that targeting p300/CBP via specific inhibitor CCS1477 could effectively repress NEPC xenograft tumors.

Here are a few points that the authors should still address (remaining issues):

1. P300/CBP are not the top binding proteins which can interact with FOXA2. What is the rationale to focus on p300/CBP only?
2. The authors should provide evidence that P300 and FOXA2/NKX2-1 can co-occupy the same chromatin region, which is required to regulate NEPC related genes. This is important to identify the interplay between p300/FOXA2/NKX2-1 on chromatin.
3. While it is unclear whether CCS1477 only targets the p300/CBP, it is important to use siRNA or sgRNA to profile the essentiality of p300/CBP in NEPC cells.

Reviewer #3:

Remarks to the Author:

All my prior comments have been addressed satisfactorily.

Version 4:

Decision Letter:

9th Jan 2025

Dear Professor Yu,

Your Article, "Epigenetic remodeling and 3D chromatin reorganization governed by NKX2-1 drive neuroendocrine transdifferentiation of prostate cancer" has now been seen by an additional reviewer, who suggested that to definitively rule out the possibility of cell contamination, you should compare SNP genotypes from the single cell sequencing reads across timepoints and clones. We are interested in the possibility of publishing your study in Nature Genetics, but would like to see this request fully addressed in the form of a revised manuscript before we make a final decision on publication. Please do not hesitate to get in touch if you would like to discuss these issues further.

We therefore invite you to revise your manuscript taking into account all reviewer and editor comments. Please highlight all changes in the manuscript text file. At this stage we will need you to upload a copy of the manuscript in MS Word .docx or similar editable format.

*2) If you have not done so already please begin to revise your manuscript so that it conforms to our Article format instructions, available

http://www.nature.com/ng/authors/article_types/index.html here

*3) Include a revised version of any required Reporting Summary: <https://www.nature.com/documents/nr-reporting-summary.pdf>

Please be aware of our <https://www.nature.com/nature-research/editorial-policies/image-integrity> guidelines on digital image standards.

EXTENDED DATA FIGURES

Link Redacted

We hope to receive your revised manuscript within four to eight weeks. If you cannot send it within this time, please let us know.

Nature Genetics is committed to improving transparency in authorship. As part of our efforts in this direction, we are now requesting that all authors identified as 'corresponding author' on published papers create and link their Open Researcher and Contributor Identifier (ORCID) with their account on the Manuscript Tracking System (MTS), prior to acceptance. ORCID helps the scientific community achieve unambiguous attribution of all scholarly contributions. You can create and link your ORCID from the home page of the MTS by clicking on 'Modify my Springer Nature account'. For more information please visit please visit www.springernature.com/orcid.

Sincerely,
Chiara

Chiara Anania, PhD
Associate Editor
Nature Genetics
<https://orcid.org/0000-0003-1549-4157>

Referee expertise:

Reviewer #5: NEPC resistance, lineage plasticity

Reviewers' Comments:

Reviewer #5 (Remarks to the Author):

I was asked to comment on the clonal variability observed in the LNCaP single cell experiments. Overall, I agree with reviewer 2 on this issue. LNCaP can show some genetic variation and I don't think the differences in copy number profiles necessarily indicate contamination by another cell line or a similar serious issue. One definitive way to rule this out would be for the authors to compare SNP genotypes from the single cell sequencing reads across timepoints and clones. These should be largely similar between LNCaP clones, but would be very different if another cell line had inadvertently been introduced. Regardless, it would be nice to see this result reproduced in another cell line. I understand, though, that this would add additional time, and overall, I don't think it's too surprising that FOXA2 would have these effects, so the burden of proof in my mind is not very high on this point.

Version 5:

Decision Letter:

Our ref: NG-A63095R4

21st Feb 2025

Dear Dr. Yu,

Thank you for submitting your revised manuscript "Epigenetic remodeling and 3D chromatin reorganization governed by NKX2-1 drive neuroendocrine transdifferentiation of prostate cancer" (NG-A63095R4). It has now been seen by the original referees and their comments are below. The reviewers find that the paper has improved in revision, and therefore we'll be happy in principle to publish it in Nature Genetics, pending minor revisions to comply with our editorial and formatting

guidelines.

Sincerely,
Chiara

Chiara Anania, PhD
Associate Editor
Nature Genetics
<https://orcid.org/0000-0003-1549-4157>

Reviewer #5 (Remarks to the Author):

The authors have addressed my suggestions and I am satisfied.

Point-by-Point Rebuttal

Title: Epigenetic remodeling and 3D chromatin reorganization governed by NKX2-1 drive neuroendocrine transdifferentiation of prostate cancer

Ref: NG-A63095R1

Reviewer #1:

Remarks to the Author:

In this work, the authors performed Hi-C analysis on PDX models and performed a series of epigenomic and transcriptomic studies using a FOXA2 overexpression system with repeated sampling over 28 days. They documented decreased methylation at sites where FOXA2 binding increases during the 28 day time course and then performed in vivo studies with a model of induced NEPC to assess the impact of p300 inhibition on tumor growth. A significant amount of work is presented over seven figures, and the number of different analyses presented in this study is a strength. The text is clearly written (with minor typos that can be fixed in editing) and the figures are straightforward. My primary concerns are that it is unclear how well this model reflects the mechanism observed in patients, that the main overexpression model has previously been demonstrated, and that the effect of CCS1477 in these cells is not obviously linked to the NEPC phenotype.

We thank this reviewer for the succinct summary of our study and the overall positive comments. We also appreciate the reviewer's three primary concerns, which we address one by one below.

1. it is unclear how well this model reflects the mechanism observed in patients.

Our model reflects the mechanisms observed in a majority of NEPC patients, supported by the following evidence. First, NKX2-1 and FOXA2 are highly expressed in NEPC compared to CRPC (**Fig.5A**). Further analyses showed that approximately 100% and 75% AR-/NE+ PCa samples express NKX2-1 and FOXA2 transcripts, respectively (**Fig.5B**). IHC staining confirmed that nearly 60% and 38% of NEPC tumors express high levels of NKX2-1 and FOXA2 proteins, respectively, while the detection rates for both proteins in NEPC are at 85% (**Fig.5C**), suggesting that our pathway is relevant to the majority of NEPC PCa patients. Second, cells that have undergone FOXA2-driven NET showed transcriptional and epigenetic programs that are highly similar to NEPC tumors (Revised **Fig.2B** and **Fig.S3C**). Lastly, we also validated our FOXA2/NKX2-1 pathway in additional models, including NCI-H660, RB1/TP53-loss cells (mimicking clinical NEPC driven by genetic aberration), and the 42D cells, an NET model that was derived in vivo upon resistance to AR antagonist enzalutamide (mimicking clinical treatment-induced NEPC) (**Fig.5F-H**).

2. the main overexpression model has previously been demonstrated

We agree with the reviewer that the ability of FOXA2 OE to drive NET has been demonstrated, which, along with the prior knowledge of FOXA2 being overexpressed in a majority of NEPC, is precisely the reason that we exploit the FOXA2-OE model to delineate the essential mechanisms underlying NET of PCa, orchestrated by NKX2-1 up-regulation, FOXA2-induced DNA demethylation, and promoter-enhancer looping. As in response above, we confirmed our major findings in several additional models.

3. The effect of CCS1477 in these cells is not obviously linked to the NEPC phenotype.

We thank the reviewer for this comment and have revised our manuscript with additional data (Fig.S9-10) to demonstrate the link. The efficacy of CCS1477 is due to its ability to inhibit NE lineage-specific enhancers in the LuNE cells. This aligns well with previous notion of its efficacy in CRPC cells where it targets AR and c-Myc, which are key targets of super-enhancers in CRPC. In summary, CCS1477 targets lineage-specific enhancers and genes, which vary in different cell types, and thus the drug is efficacious and specific to the molecular phenotype of various cancer types, from AR+ PCa to NEPC and to hematological malignancies. The details are provided in the response to Major Issue #3 below.

Major issues:

1) The Hopkins paper cited by the authors to justify the claim that NKX2-1 is "highly positive in more than 50% of NE lesions" reported on *de novo* small cell prostate cancer (not therapy-induced disease) and does not mention NKX2-1. What is the evidence that NKX2-1 overexpression is both frequent and a cause of NEPC in mCRPC patients? The authors suggest NKX2-1 is essential for NEPC. Is it really expressed in all NEPC samples? The authors should use published clinical genomics or epigenomics data to strengthen this claim.

We thank the reviewer for raising these questions to clarify an important point of our study. We address each of them one by one below.

- A. We agree that most of the cases in Hopkins' dataset are *de novo* small cell prostate cancer. The authors showed that 23/44 (52.3%) cases are positive for TTF-1, which is a previous name used for NKX2-1. We apologize for the confusion. Another study (PMID: 32739231) examined a total of 30 NEPC cases, including 14 *de novo* and 16 androgen deprivation treatment-induced NEPC, and found that half of the NEPC samples express TTF-1 (i.e., NKX2-1). This relatively lower percentage than ours may be attributed to the fact that more than half of these samples still express AR, despite the presence of some NE features. They also showed that the TTF-1 expression is not significantly different between *de novo* and post-treatment NEPC groups ($P = 0.51$). We found that NKX2-1 is significantly upregulated in NEPC vs. CRPC in 3 large clinical genomic datasets with treatment-induced NEPC (**Fig.5A**). NKX2-1 expression is significantly higher in AR-/NE+ cases compared with others in the GSE126078 data set (**Fig.5B**). IHC of NKX2-1 further confirmed that 85% of AR-/NE+ cases are NKX2-1 positive (**Fig.5C**). These results support that NKX2-1 overexpression is frequent in NEPC.
- B. Our study demonstrates that NKX2-1 overexpression is a cause of NEPC. We showed that NKX2-1 knockdown (KD) abolished FOXA2-driven transcriptional reprogramming and epigenetic remodeling (**Fig.4G-H**), whereas NKX2-1 OE greatly accelerated FOXA2-driven NET, shortening the NET time from 28 days to 14 days (**Fig.5D-E**). Similar essential roles of NKX2-1 in NEPC were also validated in additional models through NKX2-1 OE or KD (**Fig.5F-H**).
- C. Following the reviewer's suggestion, we also examined clinical genomics and epigenomics data that are available publicly or in house. We analyzed enhancer/promoter activity (H3K27ac) and DNA

methylation (5mC) at the NKX2-1 gene region in PDX and clinical PCa samples. We found that H3K27ac is significantly increased at both promoter and enhancer regions of NKX2-1 in NEPC compared with CRPC PDXs (**Revised Fig.S6B**, top), indicating gene activation. DNA methylation is dramatically decreased at CpG islands of the NKX2-1 gene in NEPC PDX and clinical NEPC samples (**Revised Fig.S6B**, bottom; and **Fig.S6C**). In addition, we analyzed NKX2-1 genomic alterations in the published PCa dataset (1253 samples / 1203 patients from 5 studies: MSK, Clin Cancer Res 2020; SU2C/PCF Dream Team, PNAS 2019; Multi-Institute, Nat Med 2016; SU2C/PCF Dream Team, Cell 2015; MCTP, Nature 2012) and found a low frequency of genomic alterations in the NKX2-1 gene. Genomic amplification at NKX2-1 is also not enriched in NEPC samples (**Revised Fig.S6D**). These data suggest that NKX2-1 is up-regulated in NEPC through epigenetic mechanisms, further supporting our conclusions.

2) The authors write "NKX2-1 in PCa [has] not been carefully studied". NKX2-1 was a focus of Baca et al. (ref 23). The authors should acknowledge this point and define how their work extends prior observations.

We thank the reviewer for this comment. We have now discussed this work by Baca et al. in the introduction and revised our sentence to make it more specific in pointing out the gap. Briefly, ectopic overexpression of NKX2-1 has been shown to facilitate ASCL1 in redistributing FOXA1 from luminal- to NE-specific regulatory elements, increasing H3K27 acetylation at these regions and resulting in global transcriptional changes from luminal to NE genes [PMID: 33785741]. However, the mechanism underlying this functionality and the regulation of endogenous NKX2-1 expression during NET have not been carefully studied.

3) The authors report CCS1477 controls tumor growth of the LuNE NEPC tumor model in SCID mice. This compound targets P300/CBP bromodomain. Although the authors characterize CCS1477 as "in clinical trials in CRPC patients via targeting AR", this compound is not specific to AR as it has been shown (Welti 2021) to target cMYC among other important drivers of AR-independent prostate cancer. Inclusion criteria for NCT04575766, as I read them, would allow NEPC patients to enroll. This therapy is also being tested in haematological malignancies (NCT04068597) and maybe other indications that I'm not aware of. Prior demonstrated efficacy in AR+ models of mCRPC implies the drug's mechanism is not contingent on impacting NE enhancer activity. What evidence supports the hypothesis that the specific impact of this compound on LuNE growth is related to its impact on NE enhancers?

We thank the reviewer for this very thoughtful question. The reviewer is correct that CCS1477 not only targets AR but also c-Myc in CRPC cells. In fact, our additional data showed that CCS1477 targets many enhancer-regulated genes that are involved in epithelial function in LNCaP (**Fig.S9D and 9F**). Interestingly, its targets change in LuNE cells as the active enhancers shift from luminal to NE elements. CCS1477 repressed many developmental genes, such as neuron differentiation and embryonic development genes, in LuNE cells (**Revised Fig.7G and Fig.S9E**). Indeed, super-enhancer-associated genes in LNCaP and LuNE cells are both significantly enriched for repression by CCS1477 (**Revised Fig.7H and Fig.S9G**). Therefore, CCS1477 might be efficacious in many cancer types, including hematological malignancies [PMID: 37995682], as mentioned above, by targeting lineage-specific

enhancers and genes, which vary depending on the cancer type. Yet it has limited toxicity in benign/normal prostate cell lines (**Revised Fig.S10D**), likely due to the fact that cancers, like prostate cancer, are known to be enhancer-addicted [PMID: 34937944; 33144576].

Other issues

1) How is significance for the Aggregate Peak Analysis measured? The authors report significant differences without giving details or quantification of the strength of the differences.

We thank the reviewer for pointing this out. We have added the values of APA scores, which reflect loop intensity, in **Revised Fig.1A-B, Fig.2D, and Fig.4F**. We also performed student's *t*-tests for the APA analyses by comparing the APA scores of NEPC PDXs to the APA scores of CRPC PDXs for NEPC-specific loops and CRPC-specific loops. This indicates that the mean of the APA scores is significantly different between NEPC and CRPC in both cases ($p=0.026$ for NEPC-specific and $p=0.022$ for CRPC-specific loops).

2) The authors report Hi-C rewiring during 28-day exposure to FOXA2 OE. Most correlations here are weak, but D21 most strongly correlates with D0 and D14 (0.7, 0.6). D28 is most strongly correlated with D21 (0.4) and has negligible (0.1 to -0.2) correlation with NEPC cell lines. I'm not convinced there is a positive association here.

We thank you for this important observation. To fully address this question, we performed Hi-C analyses of 4 additional PDXs, including 3 CRPC (LuCaP86.2CR, 105CR, 167CR) and 1 NEPC (LuCaP173.1), and re-generated the correlation map with all samples included. Again, we found that D0, D14, and D21 samples clustered together, along with most of the CRPC PDXs, and that D28 cells clustered outside of the CRPC cluster but did not form a tight cluster with NEPC PDXs (**Revised Fig.2C**). In general, NEPC PDXs did not have a good correlation between samples, except for the lines that were derived from the same patient (e.g., 145.1 and 145.2), suggesting huge heterogeneity of NEPC tumors. We have revised our conclusion accordingly and included this point in our discussion.

3) The material presenting methylation analysis convincingly shows an association between FOXA2 binding and decreased methylation. The authors do not present mechanistic evidence of this activity, but cite prior work by others that makes this case. This section appears to have some repetitive material and could be shortened.

Thank you for the comment. As suggested, we have removed some repetitive material and shortened this section in the revised manuscript.

4) In the comparisons of gene expression levels, the key results (e.g. Fig 2A) are convincing. However, the methods state differential expression was identified using Deseq2, while the figure legend says a paired t-test was applied. Neither refers to multiple test corrections. Please clarify these details of the methods.

Thank you for pointing this out. We have added the details of gene expression analyses in the revised methods section. Briefly, as **Fig. 1C-D** data was derived from the public data and did not have raw

RNA-seq counts, which is required for DESeq2, we used FPKM value for each gene and performed differential expression analyses using *t*-tests between the two groups. **Fig.2A** used DESeq2 with the Likelihood Ratio Test, considering the time-course nature of the data, whereas **Fig.5A** GSE126078 and Beltran2016 dataset and **Fig.7E-F** used DESeq2 with the default Wald test in a pairwise manner. DESeq2 has built-in multiple-test corrections for all tests. **Fig.5A** GSE74685 (microarray) dataset used Bioconductor limma package.

5) In the abstract, presumably the phrase "attributing to NE" should be "contributing to NE". The introduction contains some minor errors in English that should be corrected.

Thank you. We have corrected this and checked other errors in the revised manuscript.

Reviewer #2:

Remarks to the Author:

In this manuscript, the authors uncover how 3D genome architecture plays a critical role in driving disease phenotypes. The authors utilize patient derived xenograft and isogenic cell line models to study prostate neuroendocrine transdifferentiation. Utilizing 3D chromatin looping assays like HiC, the authors identify mCRPC and NEPC specific looping structures that can distinguish the two disease types. Notably, they find that FOXA2, a pioneer transcription factor drives the transdifferentiation by collaborating with NKX2-1. FOXA2 has been implicated in NEPC previously by others (Han et al, Cancer Cell 2022).

In this study, the authors demonstrate that FOXA2 primes NE enhancers by DNA hypomethylation, leading to activation of NE-genes. Finally, the authors show that NKX2-1 and FOXA2 interact with P300/CBP to epigenetically mark NE enhancers for activation and demonstrated that targeting p300/CBP via specific inhibitor CCS1477 could effectively repress NEPC xenograft tumors.

While the isogenic models provide some evidence for CRPC to NEPC transdifferentiation, some mechanistic questions remain. Below are specific questions/comments that need to be addressed:

We thank the reviewer for their overall enthusiasm about our study.

1. Do the authors have any evidence to show that NKX2-1 is a pioneer TF as well?

Thank you for the great questions. Most pioneer factors are known to bind lineage-specific enhancers. However, NKX2-1 showed many strong binding events at the promoters (with H3K4me3), whereas FOXA2 only binds at enhancers (**Fig.4E**). To directly address this question, we performed OE of NKX2-1 and/or FOXA2 in LNCaP cells and conducted ChIP-seq. We found that NKX2-1 alone is unable to bind the NE enhancers (C3-C4) that are inaccessible at D14 (**Fig.3A**), while FOXA2, as a pioneer factor, is able to bind (**figure on the right**, first column of FOXA2 or NKX2-1 ChIP-seq), suggesting NKX2-1 is not a pioneer TF.

2. Can the authors provide a panel of PCa cell lines to show expression of NKX2-1 and FOXA2? Do any CRPC lines express these (Particularly DU145 and PC3)?

Following this suggestion, we analyzed FOXA2, NKX2-1 mRNA and protein levels (**figure on the right**) in 2 normal prostate cell lines: PrEC and RWPE-1; one benign prostatic hyperplasia cell line: BPH-1; 6 CRPC cell lines: LNCaP, C4-2B, VCaP, 22Rv1, PC-3, DU145; and one NEPC cell line: NCI-H660. The data show that AR-negative CRPC cell line DU145 and NEPC cell line NCI-H660 express NKX2-1, whereas RWPE-1, PC-3, and NCI-H660 cells express high levels of FOXA2.

3. Figure 1j: The CRPC loops have a significant overlap with NE-CREs. How does this compare in the context of a primary PCa sample?

We thank the reviewer for this critical comment. It is important to note that the number of NE-CREs ($n=14,985$) is 3 times greater than Ad-CREs ($n=4,338$), which will, by chance, lead to a higher percentage of overlaps with any loop sets. Using the Permutation test, we showed that there are significantly more NEPC-specific loops overlap with NE-CREs, and CRPC-specific overlap with Ad-CREs, than with control chromatin regions of equal size (P value < 0.001). By contrast, there are no significance difference between NEPC-specific loops with Ad-CREs (P value=0.98) and CRPC-specific loops with NE-CREs (P value=0.87) than with control regions. We have now labeled the p values on the plots.

4. Figure S2B: Can the authors also provide immunoblots for FOXA1 and NKX3-1 expression levels?

Following this suggestion, we have added FOXA1 and NKX3-1 in the immunoblots (**revised Fig.S2B**). Similar to luminal markers AR and HOXB13, FOXA1 and NKX3-1 are also decreased during NET.

5. Fig 3c-j: What are the leading-edge genes that are defining the two distinct subsets of D0 and D21? And how are these genes impacted upon NKX2-1 or FOXA2 loss?

We thank the reviewer for this thoughtful question. When generating the UMAP, we used the top 3000 highly variable genes. We found that the leading-edge genes that are differentially expressed in LuNE cells upon FOXA2 KD or NKX2-1 KD are strongly enriched for repression (**Figure on the right**). Further, these repressed genes are functionally enriched in neuron development processes, such as neuron projection development and axon guidance (**Figure above**), being consistent with the function of NKX2-1 and FOXA2 in modulating the NE lineage state.

6. In the rebuttal, the authors pointed out that in some of FOXA2-negative PDX samples, NKX2-1 level still increases. As shown in Fig. 4, NKX2-1 is induced by FOXA2. If this is the case, what is the mechanism of NKX2-1 high expression in FOXA2 negative cells? Is NKX2-1 still important for FOXA2-negative NEPC cells?

This reviewer is correct that NKX2-1 can be highly expressed in PDX (**Revised Fig.S7A**) or clinical samples that are negative for FOXA2 (**Fig.5B-C**), likely due to the presence of upstream inducers of NKX2-1 other than FOXA2. For instance, ASCL1 strongly binds at promoter and enhancer regions of NKX2-1 in LuCaP93, and the expression of NKX2-1 is positively correlated with ASCL1 in PCa samples (**Revised Fig.S7B-C**). Further, ASCL1 KD in LuCaP93 organoids significantly reduced the expression of NKX2-1 at both protein and mRNA levels (**Revised Fig.S7D**).

We believe NKX2-1 is still important for FOXA2-negative NEPC cells. Baca et al. have previously shown that ectopic NKX2-1 could facilitate the ability of ASCL1 to reprogram FOXA1 to NE enhancers and induce NET in LNCaP cells that are negative for FOXA2 [PMID: 33785741]. In the LuCaP93 model, we found substantial NKX2-1 and ASCL1 co-occupancy on chromatin, and NKX2-1 KD dramatically inhibited LuCaP93 organoids growth (**Revised Fig.S7E-F**).

7. Furthermore, the authors claim that NKX2-1 and FOXA2 interact with each other to facilitate the chromatin looping and upregulate downstream gene expression. If this is the case in FOXA2-positive NEPC, what is the mechanism of action of NKX2-1 in FOXA2-negative PDX? Do FOXA2-negative NEPC cells have distinct 3D chromatin structures as FOXA2-positive NEPC cells?

We thank the reviewer for this thoughtful question. We believe NKX2-1, as a neural transcription factor, is able to cooperate with pioneer factors other than FOXA2, such as ASCL1. Indeed, the ASCL1 motif is significantly enriched in NKX2-1 and/or FOXA2-binding sites in NCI-H660 and LuCaP145.1 cells that express all three proteins (**Revised Fig.S5G**). As described in response to point #6 above, we found a significant overlap between ASCL1 and NKX2-1 binding sites in FOXA2-negative LuCaP93 (**Revised Fig.S7E**). Indeed, NKX2-1 cooperation with ASCL1 has already been reported by Baca et al. [PMID: 33785741].

Regarding the 3D chromatin structure, we found that all NEPC PDX models are separated from the CRPC cluster (**Revised Fig.2C**). To directly address whether FOXA2-negative NEPC cells have distinct 3D chromatin structures as FOXA2-positive cells, we performed chromatin compartment analysis based on the top 25% most variable PC1 values in NKX2-1-positive NEPC models. We found that FOXA2+ NEPC cells (LuCaP145.1, 145.2 and NCI-H660) clustered in one group, while FOXA2-NEPC cells (LuCaP93 and 173.1) are loosely clustered (**Figure above**), indicating potential differences. However, the sample size is too small to conclude a real difference between the two clusters other than high heterogeneity across NEPC samples derived from different patients.

8. Is NKX2-1's induction by FOXA2 evident at the mRNA level?

The reviewer is correct. RNA-seq showed that NKX2.1 is strongly induced by FOXA2 OE, starting at D14, in LNCaP cells (**Fig.4A**). To confirm the RNA-seq data, we now performed qRT-PCR of NKX2-1 in LNCaP-FOXA2 time-course cells and stable LuNE cells. The mRNA of NKX2-1 is increased ~160 fold in later time points (D21, D28) and LuNE cells compared with D2 cells (**Revised Fig.S5A**).

9. Regarding statement “FOXA2-only binding sites were marked with H3K4me1 only, indicative of primed enhancers”, based on Figure 4e, it doesn't seem to be the case. Are the authors referring to another piece of data in support of this claim?

We thank the reviewer for pointing this out. The issue was two folds. First, H3K4me1 enrichment was overall much weaker than H3K27ac, which we now adjusted by using different scale bars to make the strongest signal of the 3 histone markers similar in the heatmap (**Revised Fig.4E**). Second, FOXA2 enrichment at FOXA2-only sites is itself not strong, supporting the requirement of NKX2-1 co-occupancy to further enhance FOXA2 binding and H3K4me1. Thus, FOXA2-only sites are often weakly occupied by FOXA2 and slightly primed, with no H3K27ac at all (**example in Figure above, red highlighted**). We have revised our sentence to better reflect this.

10. The authors show that NKX2-1 is a promoter-binding protein while FOXA2 is an enhancer-binding protein.

The reviewer is correct. Our data showed that NKX2-1 has a lot of binding sites at promoters, marked by H3K4me3, and that FOXA2 is almost exclusively binding to enhancers (marked by H3K4me1 and no H3K4me3) (**Revised Fig.4E and Fig.S5E**). Pie Chart analyses showed that over 40% of NKX2-1-only sites are at promoters (**Revised Fig.S5F**), similar to previously reported promoter-dominating transcription factors such as ERG[PMID:20478527], whereas only ~15% of FOXA2-only sites are located at promoters, typical of a pioneer factor (**Revised Fig.S5F**). Likewise, **Fig.4H** showed much stronger NKX2-1 enrichment at promoter regions (C1 and C2). However, we do recognize NKX2-1 binding events at enhancer elements, likely due to recruitment by FOXA2. We have softened our conclusion in the revised manuscript to indicate that NKX2-1 by itself is a predominantly promoter-binding protein but can also bind to enhancers likely via recruitments by pioneering factors.

11. Figure 4g: Can the authors also provide expression of other luminal markers like FOXA1 and NKX2-1, and NEPC markers like ASCL1?

Following this suggestion, we have added luminal markers FOXA1 and NKX3-1 in **revised Fig.4G**. As ASCL1 is not detectable in our model, we added another NE marker, INSM1, in the revised figure.

12. The author applied p300/CBP bromodomain inhibitor, CCS1477 to block the NEPC tumor growth. However, several studies showed that p300/CBP bromodomain inhibitors exhibit poor efficacy in AR negative cells (DU145 and PC3, Jonathan Welti et al and Lingyan Jin et al). NEPC is AR negative as well, so the data presented here does not seem consistent with other reports.

We thank you for this great question. To address this, we tested CCS1477 in various PCa models. Our data indeed show that 100nM CCS1477 killed over ~95% and 90% of LNCaP and LuNE cells, respectively (**Revised Fig.S10B-C**). CCS1477 is also effective in additional NEPC cell models, NCI-H660 and LuCaP145.2 organoids (**Revised Fig.S10E**). Critically, CCS1477 did not show toxic effect on the growth of normal or benign prostate cell lines RWPE-1 and BPH-1 (**Revised Fig.S10D**). We believe that p300 inhibitors such as CCS1477 target lineage-specific enhancers to inhibit cancer cells that are addicted to enhancers, such as CRPC and NEPC [PMID: 34937944; 33144576]. For details, please check our response to major comment #3 of Reviewer #1. It is possible that Du145 and PC3, which are not typical NEPC models and have not been reported to have addiction to enhancers, have less sensitivity to p300 inhibitors. Welti et al. showed that CRPC cells with AR and variant overexpression are more sensitive to CCS1477 than LNCaP, which could be caused by a higher dependency on AR enhancers, being consistent with our data.

13. Figure 7M: Does treatment of NEPC models with CCS1477 induce expression of luminal markers like PSA, AR, etc.? How reversible is the FOXA2/NKX2.1-driven NEPC?

This is a great question. To address this, we first treated LuNE cells with 500nM of CCS1477 for up to 60 hours and observed that the expression of NKX2-1 gradually decreased. However, the expression of luminal markers, such as AR and PSA, was not recovered by CCS1477 (**Revised Fig.S10I**). Second, we performed H&E and IHC of AR and PSA in LuCaP145.2 tumors treated with vehicle or CCS1477. CCS1477 was not able to revert small-cell carcinoma histology of LuCaP145.2 to adenocarcinoma, nor the expression of luminal markers: AR and PSA (**Revised Fig.S10K**). Taken together, these data suggest that the chromatin/epigenetic state of the fully transformed LuNE cells is stable and cannot be reverted by p300/CBP inhibitor CCS1477. However, it is possible at some earlier stage of NET, p300i might be able to prevent or revert NET, which, however, is beyond the scope of the current study. We discussed this important point in the revised manuscript.

14. What is the NKX2-1 status in DU and PC3 cells?
Are these 2 cells also dependent on NKX2-1?

DU145, but not PC-3, expresses NKX2-1 (for a complete cell line panel, please see our response to major point #2 above). NKX2-1 KD moderately decreased DU145 cell colony formation (figure above).

Minor comments:

1. Fig. 7a, the mass spec data suggests that SWI/SNF complex also interact with FOXA2. Is there any data showing that targeting SWI/SNF complex could repress NEPC growth?

To address this question, we treated LuNE cells with SMARCA2/4 ATPase inhibitor (BRM014) and SMARCA2/4 degrader (ACB11) and found that they repressed NEPC cell growth (**Figure below**). However, further investigation of SMARCA2/4 is beyond the scope of this present study.

2. Fig. 7b, the quality of co-IP is poor and needs to be improved.

Thanks for pointing out this issue. We have repeated the experiments and have improved Co-IP data in revised Fig.7b.

Reviewer #3:

Remarks to the Author:

The study presents interesting findings on the chromatin architecture differences between neuroendocrine prostate cancer (NEPC) and castration-resistant prostate cancer (CRPC), providing a deep understanding of the mechanisms driving NEPC progression. Through Hi-C analyses of patient-derived xenograft tumors and in vitro experiments on isogenic cells undergoing neuroendocrine transformation (NET), the authors demonstrate significant 3D chromatin re-organization in NEPC samples compared to CRPC. Mechanistically, they identify the neural transcription factor NKX2-1 as a key regulator of NET, orchestrating a hierarchical network of transcription factors and chromatin modifiers. This study expands on recent studies of FOXA2 in NEPC, by showing that NKX2-1 is indispensable for the phenotype driven by FOXA2. Additionally, single-cell multiome analyses reveal intermediate epigenetic and transcriptomic states during luminal-to-NE transformation, providing insights into the evolution of NEPC. The finding of dependency on CBP/p300 and the therapeutic potential of CBP/p300 is very clinically relevant.

In summary, the study presents several novel insights into the role of chromatin dynamics in NEPC progression, filling gaps in understanding the molecular mechanisms underlying this aggressive subtype

of prostate cancer. While previous studies have explored the role of FOXA2 in cancer progression, this work integrates chromatin architecture analysis with transcriptional regulation to unveil a comprehensive regulatory network of FOXA2-NKX2-1-CBP/p300 governing NEPC development. However, a few questions need to be addressed to strengthen the conclusions.

We thank the reviewer for a succinct, high-level summary of our work, recognizing the novelty and impact of our study on the field.

1. It is very clear that FOXA2 overexpression induces neurite extension morphology and expression of NE markers. Does FOXA2 overexpression drive the small cell phenotype in xenograft models? If not, please discuss potential reasons.

Thanks for the great question. To fully address this question, we inoculated control (plv) and FOXA2-OE LNCaP cells into Nude-SCID mice to generate xenograft tumors (**newly added Fig.S2E-G**). Being consistent with the NE-transformed nature, FOXA2-OE LNCaP cells grew xenograft tumors rapidly, much faster than the control cells. Further, H&E staining showed heterogeneity in FOXA2-OE tumors. Morphologically, most (~95%) still manifest as adenocarcinoma, while ~5% exhibit small cell-like features. Molecularly, 76.9% of FOXA2-positive tumors are also SYP positive, suggesting most FOXA2+ tumors harbor some NEPC features, although they do not show as pure small cell carcinoma. There are two potential reasons for our observation. First, the tumors were not monitored long enough (only 5 weeks). Consistent with this hypothesis, a recent study (PMID: 37995683) using PARCB driven NEPC model shows that most tumors are adenocarcinoma and squamous cell carcinoma at an early time point (4-6 weeks) and that the small cell carcinoma feature of tumors only begins to show at transition time (8-10 weeks) and further increases at a late time point (>10 weeks). Second, for a complete small-cell carcinoma morphology in vivo, additional genetic alterations, such as TP53 and RB1 loss, may be required.

2. The time course scRNA-seq data provides a very useful resource to understand the evolution from CRPC to NEPC. However, the conclusion related to cell transformation vs. populational switch in line 249-285 is premature. As there is a small population of D0 cells demonstrating low level of AR/ARS., how do we know the increased percentage of AR-/NES+ cells in D14, D21 and D28 is not due to clonal expansion of the pre-existing population? Does this population in D0 cluster together with the AR-/NES+ in D28?

We thank the reviewer for this very provocative question that has tremendously helped us to reach a more definitive conclusion on the matter. We agree with the reviewer that a small population of D0 cells have low AR and low (AR-/NES-) and a few cells of low AR and intermediate NES (AR-/NES+).

Interestingly, they are still grouped with the luminal cluster, similar to the AR+/ARS+ cells (**Figure above**). To further investigate the clonality and how they change over NET, we inferred clonality of each cell in scRNA-seq data based on gene copy

number variation (CNV) by CopyKAT. We identified two clones: clone 1 (D0/luminal-dominant) and clone 2 (D21/NE-dominant). Over the time course, the number of clone 1 cells decreased while the number of clone 2 cells increased (Revised Fig. S4A-B). Critically, despite their clonality, D0 cells are high-ARS/low-NES and D14 cells have intermediate ARS and low NES, supporting transformation. Interestingly, most D21 clone 1 cells contained some levels of ARS and are NES-negative, whereas D21 clone 2 cells became low-ARS/high-NES (Fig. S4C). Very similar results were found from clonality analyses of scATAC-seq data (Fig. S4D-F). Thus, cells of both clonality types went through transformation from high-ARS/low-NES, to intermediate-ARS/low-NES, to low-ARS/high-NES from D0 to D21 with clone 2 being fully transformed and dominating at D21, supporting a mixed model of cell transformation and clonal selection.

3. The feedback loop of FOXA2 inducing NKX2-1 at transcriptional level, and NKX2-1 interacting with FOXA2 to activates NE regulatory sites is very interesting. Figure 4A showed increased FOXA2 binding near NKX2-1. Is there any evidence from the Hi-C data that these regions form loops with NKX2-1 promoter?

We thank this reviewer for the great question. Indeed, there are a number of FOXA2-bound enhancers interacting with NKX2-1 promoter in this region, being the most apparent in NEPC PDXs and NCI-H660 cell line (Revised Fig.S5D), but also emerging in D21 and D28 cells (Revised Fig.S5C).

4. In the D28 scRNA-seq, are there FOXA2+/NKX2-1 negative cell population? If yes, any difference of NES compared to FOXA2+/NKX2-1+ cells?

This is a great question. However, our ability to address the question is constrained by the inherent limitation of scRNA-seq technology in detecting lowly expressed genes due to the compromised sequencing depth for each cell when thousands of cells are sequenced simultaneously. NKX2-1 expression is a lowly expressed gene, even though its expression increases from 0.6 to 100 RPKM over the time course. The scRNA-seq technology also has a challenge in detecting ectopic FOXA2, which is 2-kb away from ectopic poly-A tail in the construct and scRNA-seq only captures the 3' end of the genes. In an attempt to address the question, we re-analyzed D21 scRNA-seq with additional sequencing, reaching 34,125 read pairs/cell, which is still much less than bulk RNA-seq (often of 200,000 read pairs). After using an imputation method (Please see Methods for details), we detected FOXA2 in 44.02% of cells (1705/3873), out of these, NKX2-1 was detected in 7.9% of the cells. We compared the differences of NES as suggested by the reviewer in the cell populations of distinct FOXA2/NKX2-1 status and found that the AR signature score is significantly lower and the NE signature score is significantly higher in F+/N+ cells compared with F+/N- and other cells (right Figure), supporting the roles of FOXA2 and NKX2-1 in cooperatively regulating NET.

5. Is there FOXA2 expression in the P53/RB1 double KD model? The author claimed that NKX2-1 is as a general

regulator of NET that likely also cooperates with other chromatin-pioneering factors. Is there any evidence from the NKX2-1 ChIP-seq data of the potential pioneering factors, e.g., motifs in NKX2-1 binding sites that do not overlap with FOXA2 binding?

Thanks for the questions. We indeed observed that the expression of FOXA2 is induced in LNCaP cells with RB1/P53 double KD for 4 weeks (**Revised Fig.S2D**). We analyzed potential transcription factor motifs that are enriched in NKX2-1-only sites in LuCaP145.1 and NCI-H660 and found significant enrichment of pro-neuronal transcription factor ASCL1 motif in NKX2-1-only sites (**Revised Fig.S5G**). This data indicates that NKX2-1 may cooperate with ASCL1 to regulate NET in the FOXA2-negative model, which we discussed in detail in response to Reviewer #2 major point #6.

6. The finding of CBP/P300 from FOXA2 IP mass spec and the potential therapeutic effect of CBP/p300 inhibitor is intriguing. Figure 7M demonstrated a reduction of SOX2 and NKX2-1 in CCS1477-treated LuCaP145.2 tumors. Do these tumors remain as small cell or demonstrate features of adenocarcinoma? Thanks for the great questions. The CCS1477-treated tumors remained as small cell carcinoma without the induction of adenocarcinoma markers, such as AR and PSA (**Revised Fig.S10K**), although NE lineage genes, including SOX2 and NKX2-1, are suppressed (Please check our response to Reviewer #2 major comment #13 for more discussions).

7. Figure 1A. Please add description of Aggregate Peak Analysis in Figure legend or main text. Thanks for pointing this out. We have added details for APA analysis in the revised methods section.

8. Figure resolution needs to be improved for many of the panels. Some of the text in Figure 1E-F overlaps; adjustments to alignment are needed. Thanks for the comments. Figure resolution was compromised as we made the PDF figures manageable. We have now updated with high-resolution Hi-C contact maps and heatmaps in the revised figures and improved Fig.1E-F.

9. Line 135, please rephrase 'erased' as it is unclear whether the loss of H3K27me3 in NEPC is due to loss of writer or gain of eraser activities. Thanks for pointing out the issue. We have revised the term as diminished in the updated manuscript.

Point-by-Point Rebuttal

Title: Epigenetic remodeling and 3D chromatin reorganization governed by NKX2-1 drive neuroendocrine transdifferentiation of prostate cancer

Ref: NG-A63095R2

Reviewer #1:

Remarks to the Author:

The manuscript has been significantly strengthened by the author's careful response to all three reviewer's comments. The major findings of the manuscript are well-supported and overall the interpretation of the results is reasonable.

Response: We thank you for the very positive comments on the high quality of our revised manuscript.

Major comments:

1) The investigators report that "CCS1477 targets lineage-specific enhancers and genes". Does CCS1477 target distinct collections of lineage-specific enhancers in adeno vs. NEPC cells, or does it act generally on active enhancers? The investigators report that treating LNCaP cells with CCS1477 affects "prostate luminal cell functions, such as cell adhesion, secretion, and lipid metabolic process". These GO terms in the LNCaP case are activities that take place in most cells, and it is unclear how they demonstrate lineage specificity. To demonstrate enrichment for lineage-specific enhancers, in comparison to other enhancers, the investigators should specifically test this claim by defining two sets of enhancers: lineage-specific enhancers, and non-lineage-specific enhancers, and demonstrate that there is enrichment for CCS1477-induced activity in the lineage-specific enhancers compared to a background of all other enhancer sites.

Response: we thank you for this critical question. To address this question, we performed H3K27ac ChIP-seq, the marker of active enhancers, in LNCaP and LuNE cells treated with vehicle control or 250nM of CCS1477 (Figure on the right). Importantly, we found that 1) distinct sets of enhancers are active in the two cell lines - luminal enhancers (C3, C4) are active in LNCaP whereas NE enhancers (C5, C6) are active in LuNE cells; and 2) CCS1477 targets these lineage-specific enhancers as well as shared enhancers (C1, C2), which however still maintain some basal activities. Therefore, we conclude that CCS1477 targets all active enhancers in a cell. However, because the active enhancers in LuNE cells are distinct from the active enhancers in LNCaP, CCS1477 treatment of a particular cell selectively targets lineage-specific enhancers of that cell/lineage over enhancer elements of other lineages.

As lineage-specific enhancers or super-enhancers (SE) are essential for the transcription of proto-oncogenes and lineage-specific transcription factors [PMID:23582323; 23582322], CCS1477 may be a potent drug for both prostate adenocarcinoma and NEPC, which is consistent with our preclinical data (Figure 7I-M). To verify this mechanism of action for CCS1477, we first performed SE analyses and identified 394 SEs in LuNE cells (Revised Figure S9C). Critically, many SE-associated genes in LuNE cells are embryo/neuron-specific transcription factors, such as HOXA1, HOXB2, and ZIC5 [PMID: 28860158; 24094100; 15136147], and proto-oncogenes, such as SOX2 and NFIB, that have been implicated in NEPC and small-cell lung cancer [PMID: 28059768; 28059767; 27374332; 21764851]. By contrast, genes associated with SEs in LNCaP cells are prostatic transcription factors, such as HOXB13, FOXA1 and GATA2, and oncogenic transcription factors, such as CCND1 and MYC (Revised Figure S9E). Next, we examined how they respond to CCS1477 treatment. SE-associated genes in LuNE cells are highly significantly enriched for suppression by CCS1477 (Figure 7H), accompanied by a remarkable loss of H3K27ac (Revised Figure S9D). Similarly, CCS1477 led to a substantial loss of H3K27ac at SEs in LNCaP cells (completely different from the SEs in LuNE) with concordant repression of associated genes (Revised Figure S9F and L). Therefore, the ability of CCS1477 to target lineage-specific enhancers underscores its therapeutic effects. Moreover, recent studies report that SEs are more sensitive to p300 inhibition than typical enhancers [PMID: 33765415; 34772733; 30773298], further supporting its selectivity. We have discussed these in our manuscript. We thank the reviewer for this provocative question, which has substantially improved the clarity of our manuscript and further increased our understanding of the mechanism of action of CCS1477.

To address the side question regarding the GO terms, cell adhesion, secretion, and lipid metabolic process, not being representative of the luminal function of LNCaP cells, we performed Hallmark and GSEA analyses of CCS1477-repressed genes in LNCaP and found that androgen response genes are indeed significantly repressed by CCS1477 (Revised Figure S9K). Our data is consistent with the finding in a recent study that the transcription of androgen response genes was significantly inhibited by CCS1477 in 22Rv1 cells [PMID: 33431496].

2) The investigators should clarify whether CCS1477 is cytostatic or cytotoxic, as they alternately refer to growth-inhibitory effects and the phrase "strong killing effect".

Response: we thank you for pointing out this ambiguity in our verbiage. To address this and also improve our understanding of how CCS1477 works, we performed cell cycle analyses and found that CCS1477 treatment drastically induced cell cycle arrest at G1 and S phase in LNCaP and LuNE, respectively (Figure on the right), which is consistent with previous reports [PMID: 38766099; 37995682]. Consistently, WB showed a clear increase of p21 expression following CCS1477 treatment, whereas there was no detectable expression of apoptotic markers (Revised Figure S10D). We thus conclude that CCS1477 is cytostatic and have revised our manuscript accordingly.

3) The investigators must provide details in their methods describing how they assessed RNA velocity, particularly in light of recent reports that different published methods for calculating velocity can produce results that strongly disagree (e.g.

<https://www.biorxiv.org/content/10.1101/2024.06.25.600667v1>). It would be helpful to apply a method showing that their data have noise properties that are suitable for RNA velocity analysis.

Response: we thank the reviewer for this comment. We have updated the details of RNA velocity analysis in Methods. Briefly, Velocyto (0.17.17) was used to generate spliced and unspliced counts for all cells. A subset of high-quality cells was kept for following analysis. Genes included in the analysis were filtered by detection levels and the top 3000 highly variable genes were selected and normalized. The “constant velocity” assumption was used when calculating velocity estimation. RNA velocities were projected into the UMAP which was generated in Seurat.

The reviewer is correct that the agreement of different methods for calculating velocity could be low especially when the samples are heterogenous, and the quality of single-cell data is low. To address your concern, we repeated the analysis using a different method, scVelo. Using both dynamic and steady-state modes in scVelo, we found that both modes generated similar results where there is a tendency of KLK3-high cells to become KLK3-low cells (**Figure below**), consistent with the conclusion made by the Velocyto method in revised **Fig.3G**. We would like to point out that the biorxiv paper the reviewer has kindly provided indeed mentioned that different RNA velocity analysis methods do agree well for certain cell types, specifically those with well-defined lineages, which is precisely our case.

In addition, our clonality analysis in **revised Fig.S4** suggests that cells of both clonality types went through transformation from ARS+/NES- to ARS-intermediate/NES-, to ARS-/NES+ from D0 to D21, further supporting that the cells transformed from KLK3-high to KLK3-low cells.

Minor points

1) In the discussion, the word "generable" should be "general".

Response: Thanks for noting this. We have revised it in the manuscript.

Reviewer #2:

Remarks to the Author:

In this manuscript, the authors uncover how 3D genome architecture plays a critical role in driving disease phenotypes. The authors utilize patient derived xenograft and isogenic cell line models to study prostate neuroendocrine transdifferentiation. Utilizing 3D chromatin looping assays like HiC, the authors identify mCRPC and NEPC specific looping structures that can distinguish the two disease types. Notably, they find that FOXA2, a pioneer transcription factor drives the transdifferentiation by collaborating with NKX2-1. FOXA2 has been implicated in NEPC previously by others (Han et al, Cancer Cell 2022).

In this study, the authors demonstrate that FOXA2 primes NE enhancers by DNA hypomethylation, leading to activation of NE-genes. Finally, the authors show that NKX2-1 and FOXA2 interact with P300/CBP to epigenetically mark NE enhancers for activation and demonstrated that targeting p300/CBP via specific inhibitor CCS1477 could effectively repress NEPC xenograft tumors.

Here are a few points that the authors should still address (remaining issues):

1. P300/CBP are not the top binding proteins which can interact with FOXA2. What is the rationale to focus on p300/CBP only?

Response: we thank the reviewer for this question and have revised our manuscript to clarify this point. Briefly, there are two reasons for us to focus on p300/CBP, rather than other interacting proteins. First, the focus of our study is to investigate epigenetic mechanisms underlying luminal-to-NE transformation. While the interaction between FOXA2 and NKX2-1 explains NE enhancer priming. Ultimately, these enhancers need to get activated in order to express the associated NE genes. We thus examined H3K27ac and focused on p300/CBP that catalyze H3K27ac. Second, a major motivation for us to investigate the molecular mechanisms underlying luminal-to-NE transformation is to develop a cure for NEPC. We are therefore looking for something druggable. While working on our previous *Nature Genetics* paper [PMID: 35468964], where we studied H3K27ac in HOXB13-loss cells, we became interested in its inhibitors and became aware of CCS1477. Although the critical functions of p300/CBP and its inhibition by CCS1477 have been reported in CRPC [PMID: 33431496], they have not been carefully studied in NEPC. In fact, some studies using DU145 and PC-3 cells have resulted in the misleading conclusion that p300/CBP inhibitors do not work for NEPC, as discussed in the previous rebuttal, further necessitating this study.

2. The authors should provide evidence that P300 and FOXA2/NKX2-1 can co-occupy the same chromatin region, which is required to regulate NEPC related genes. This is important to identify the interplay between p300/FOXA2/NKX2-1 on chromatin.

Response: we fully agree with this reviewer that it is critical to show that p300 co-occupies the same chromatin regions as FOXA2 and NKX2-1 and is required for NEPC-related gene expression. Looking

at various FOXA2-binding sites, ChIP-seq of p300 in LuNE cells confirmed that it co-occupies at D21 and D28 FOXA2-binding sites, with concordant enrichment of H3K27ac (**Fig.7D, left 4 panels**). We further showed that knockdown of p300 or CCS1477 treatment eliminated H3K27ac enrichment at these sites (**Fig.7D, right two panels**) and abolished FOXA2/NKX2-1-regulated genes in a way very similar to knockdown of FOXA2 or NKX2-1 themselves (**Fig.7E**). Moreover, we examined their occupancy at the 6 ATAC-seq clusters and found that, in LuNE cells, FOXA2, NKX2-1, and p300 co-occupy NE enhancers (C3, C4) as well as the shared enhancers (C1, C2) (**Revised Figure S9A**).

3. While it is unclear whether CCS1477 only targets the p300/CBP, it is important to use siRNA or sgRNA to profile the essentiality of p300/CBP in NEPC cells.

Response: To address this question, we performed RNA-seq of LuNE cells subjected to either CCS1477 treatment or p300/CBP knockdown. Integrated analysis showed that the majority of CCS1477-regulated genes are concordantly regulated by p300/CBP knockdown, especially their double knockdown (**Revised Figure S9J**), supporting that CCS1477 selectively targets p300/CBP. This is consistent with the previous report that CCS1477 was developed to selectively target specific conserved bromodomain of p300 and CBP and shows excellent selectivity against other bromodomain-containing proteins such as BRD4 [PMID: 33431496].

Reviewer #3:

Remarks to the Author:

All my prior comments have been addressed satisfactorily.

Response: Thank you!

Point-by-Point Rebuttal

Title: Epigenetic remodeling and 3D chromatin reorganization governed by NKX2-1 drive neuroendocrine transdifferentiation of prostate cancer

Ref: NG-A63095R3

Reviewer #5 (Remarks to the Author):

I was asked to comment on the clonal variability observed in the LNCaP single cell experiments. Overall, I agree with reviewer 2 on this issue. LNCaP can show some genetic variation and I don't think the differences in copy number profiles necessarily indicate contamination by another cell line or a similar serious issue. One definitive way to rule this out would be for the authors to compare SNP genotypes from the single cell sequencing reads across timepoints and clones. These should be largely similar between LNCaP clones, but would be very different if another cell line had inadvertently been introduced. Regardless, it would be nice to see this result reproduced in another cell line. I understand, though, that this would add additional time, and overall, I don't think it's too surprising that FOXA2 would have these effects, so the burden of proof in my mind is not very high on this point.

Response: We thank you for your expert opinion on LNCaP cell line and its subclonal diversity.

We thank you for your suggestion to examine SNP genotypes at single-cell level. To address this, we performed scNanoRNA-seq [PMID: 37433798], a novel long-read-based single-cell RNA sequencing using Nanopore Technology to analyze the D0 (start point) and D28 (endpoint) LNCaP+FOXA2 cells. This innovative scNanoRNA-seq sequences the full-length mRNAs and/or pre-mRNAs, thus providing a deeper and broader genome coverage and a much higher likelihood of capturing single-nucleotide variants (SNVs), the majority of which occur in intronic and intergenic regions, with only ~10% at 5'UTR, 3'UTR, and exon regions [PMID: 31735626, 36997428]. As short-read scRNA-seq, such as ours, often uses a 3' gene expression kit from 10x Genomics, which only captures ~100bp of 3'UTR regions of expressed genes, SNV calling from short-read scRNA-seq data has been very challenging, with results greatly confounded by individual gene expression levels and limited by poor coverage [PMID: 30459309, 37592035].

Comparing scNanoRNA-seq data of D0 and D28 cells, we identified 1093 SNVs with variant alleles present in at least 5% of high-quality cells of both timepoints (**Methods**). Clustering analyses based on variant allele frequencies (VAFs) of these 1093 SNVs revealed two clone types (clone1 and clone2) (**Revised Fig. S4D**). Most D0 cells were clone1, with only 69 cells being clone2 type, while most D28 cells were clone2 type, with only 11 clone1 cells (**Revised Fig. S4E**). UMAP analyses of gene expression separated the D0 and D28 cells into two clusters enriched for AR signature (ARS) and NE signature (NES), respectively (**Revised Fig. S4E-F**). Critically, both clone1 and clone 2 D0 cells showed a much higher ARS score and lower NES score than the clones of D28 cells (**Revised Fig. S4G**), suggesting that cells of both clonality types went through initial NE transformation with clone2 becoming dominant in D28,

supporting a mixed model of cell transformation and clonal selection during FOXA2-driven NET.

To determine whether the two clone types are isogenic, we analyzed germline mutations, defined as SNVs that have read coverage (i.e., covered) in more than 50% of all cells of a particular timepoint and were mutated, supported by at least 1 mutant read, in more than 95% of these covered cells. We found 585 and 357 germline mutations in D0 and D28 cells, respectively. Critically, a significant 139 germline mutations ($p < 0.001$, Fisher's exact test) were shared between D0 and D28 cells (**Revised Fig.S4H**). In addition, the rest also showed an overall trend of consistency, despite many of them being detected in only one but not the other timepoint, wherein they had zero (appearing as NA) or insufficient read coverage to capture the mutant allele (appearing as WT). Altogether, these results strongly support that D0 and D28 cells were subclones of LNCaP cells.

Point-by-Point Rebuttal

Title: Epigenetic remodeling and 3D chromatin reorganization governed by NKX2-1 drive neuroendocrine transdifferentiation of prostate cancer

Ref: NG-A63095R4

Reviewer #5:

Remarks to the Author:

The authors have addressed my suggestions and I am satisfied

Response: We thank the reviewer for all the meaningful comments and suggestions.